# Allosteric conformational ensembles have unlimited capacity for integrating information

**John W Biddle[1†‡], Rosa Martinez-Corral[1†], Felix Wong[2,3], Jeremy Gunawardena[1]***

[1]Department of Systems Biology, Harvard Medical School, Boston, United States; [2]Institute for Medical Engineering and Science, Department of Biological Engineering, Massachusetts Institute of Technology, Cambridge, United States; [3]Infectious Disease and Microbiome Program, Broad Institute of MIT and Harvard, Cambridge, United States

**Abstract** Integration of binding information by macromolecular entities is fundamental to cellular functionality. Recent work has shown that such integration cannot be explained by pairwise cooperativities, in which binding is modulated by binding at another site. Higher-order cooperativities (HOCs), in which binding is collectively modulated by multiple other binding events, appear to be necessary but an appropriate mechanism has been lacking. We show here that HOCs arise through allostery, in which effective cooperativity emerges indirectly from an ensemble of dynamically interchanging conformations. Conformational ensembles play important roles in many cellular processes but their integrative capabilities remain poorly understood. We show that sufficiently complex ensembles can implement any form of information integration achievable without energy expenditure, including all patterns of HOCs. Our results provide a rigorous biophysical foundation for analysing the integration of binding information through allostery. We discuss the implications for eukaryotic gene regulation, where complex conformational dynamics accompanies widespread information integration.

**\*For correspondence:**
jeremy_gunawardena@hms.
harvard.edu

[†]These authors contributed equally to this work

**Present address:** [‡]Holy Cross College, Notre Dame, United States

**Competing interests:** The authors declare that no competing interests exist.

## Introduction

Cells receive information in different ways, of which molecular binding is the most diverse and widespread. Binding events influence downstream biological functions. In the biophysical treatment that we present here, biological functions, such as the output of a gene or the oxygen-carrying capacity of haemoglobin, are quantified as averages over the probabilities of microscopic states. We will be concerned with how binding events collectively determine these probability distributions and will refer to this process as the integration of binding information.

The most proximal form of such integration is pairwise cooperativity, in which binding at one site modulates binding at another site. This can arise through direct interaction, where one binding event creates a molecular surface, which either stabilises or destabilises the other binding event. This situation is illustrated in *Figure 1A*, which shows the binding of ligand to sites on a target molecule. (In considering the target of binding, we use 'molecule' for simplicity to denote any molecular entity, from a single polypeptide to a macromolecular aggregate such as an oligomer or complex with multiple components.) We use the notation $K_{i,S}$ for the association constant—on-rate divided by off-rate, with dimensions of (concentration)$^{-1}$—where $i$ denotes the binding site and $S$ denotes the set of sites which are already bound. This notation was introduced in previous work (*Estrada et al., 2016*) and is explained further in the Materials and methods. It allows binding to be analysed while keeping track of the context in which binding occurs, which is essential for making sense of how binding information is integrated.

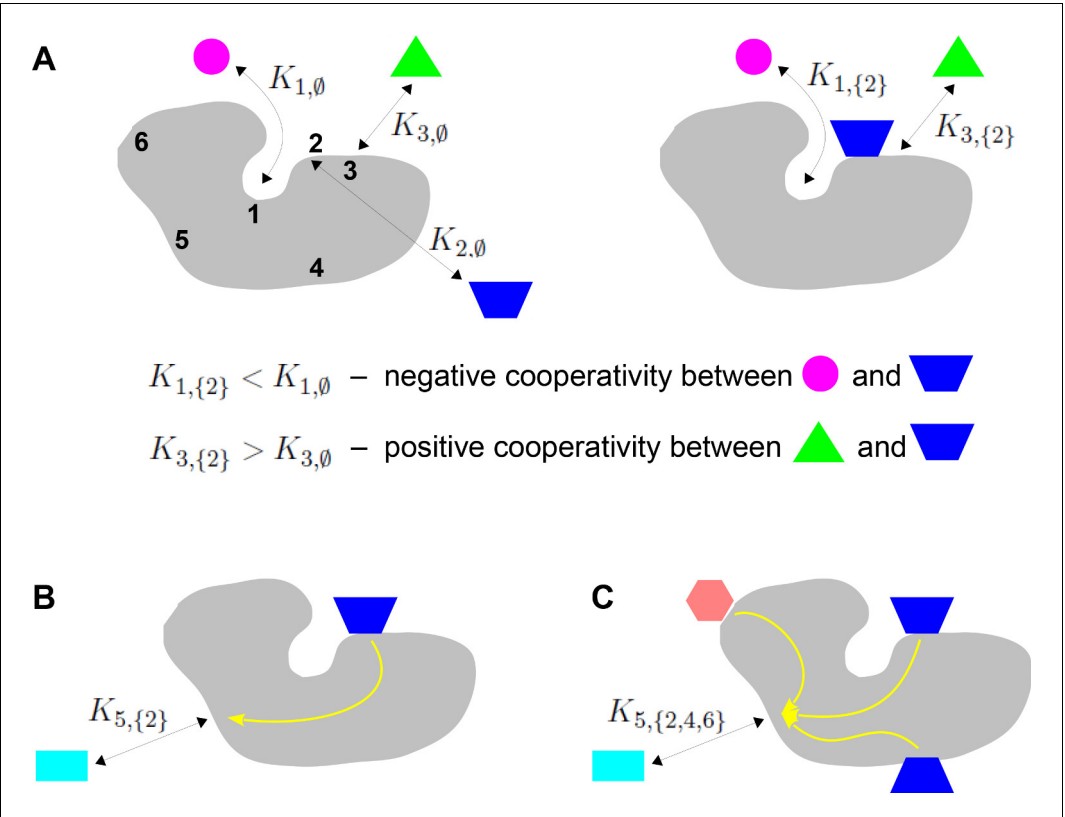

**Figure 1.** Binding cooperativity. (**A**) Pairwise cooperativity by direct interaction on a target molecule (grey). As discussed in the text, the target could be any molecular entity. Left: target molecule with no ligands bound; numbers $1, \cdots, 6$ denote the binding sites. Right: target molecule after binding of blue ligand to site 2. (**B**) Indirect long-distance pairwise cooperativity, which can arise 'effectively' through allostery. (**C**) Higher-order cooperativity, in which multiple bound sites, 2, 4 and 6, affect binding at site 5.

Oxygen binding to haemoglobin is a classical example of integration of binding information, for which Linus Pauling gave the first biophysical definition of cooperativity (*Pauling, 1935*). At a time when the mechanistic details of haemoglobin were largely unknown, Pauling assumed that cooperativity arose from direct interactions between the four haem groups. He defined the pairwise cooperativity for binding to site $i$, given that site $j$ is already bound, as the fold change in the association constant compared to when site $j$ is not bound. In other words, the pairwise cooperativity is given by $K_{i,\{j\}}/K_{i,\emptyset}$, where $\emptyset$ denotes the empty set. (Pauling considered non-pairwise effects but deemed them unnecessary to account for the available data.) It is conventional to say that the cooperativity is 'positive' if this ratio is greater than 1 and 'negative' if this ratio is less than 1; the sites are said to be 'independent' if the cooperativity is exactly 1, in which case binding to site $j$ has no influence on binding to site $i$. This terminology reflects the underlying free energy (*Equation 1*). Association constants and cooperativities may be thought of as an alternative way of describing the free-energy landscape, as we will explain in more detail in the Results. *Figure 1A* depicts the situation in which there is negative cooperativity for binding to site 1 and positive cooperativity for binding to site 3, given that site 2 is bound.

Studies of feedback inhibition in metabolic pathways revealed that information to modulate binding could also be conveyed over long distances on a target molecule, beyond the reach of direct interactions (*Changeux, 1961*; *Gerhart, 2014*; *Figure 1B*). Monod and Jacob coined the term 'allostery' for this form of indirect cooperativity (*Monod and Jacob, 1961*). Monod, Wyman and Changeux (MWC) and, independently, Koshland, Némethy and Filmer (KNF) put forward equilibrium thermodynamic models, which showed how effective cooperativity could arise from the interplay between ligand binding and conformational change (*Koshland et al., 1966*; *Monod et al., 1965*). In

the two-conformation MWC model (*Figure 2B*), there is no 'intrinsic' cooperativity—the binding sites are independent in each conformation—and 'effective' cooperativity arises as an emergent property of the dynamically interchanging ensemble of conformations.

In these studies, the effective cooperativity between sites was not quantitatively determined. Instead, the presence of cooperativity was inferred from the shape of the binding function, which is the average fraction of bound sites, or fractional saturation, as a function of ligand concentration (*Figure 2A*). The famous MWC formula is an expression for this binding function (*Monod et al., 1965*). If the sites are effectively independent, the binding function has a hyperbolic shape, similar to that of a Michaelis–Menten curve. A sigmoidal curve, which flattens first and then rises more steeply, indicates positive cooperativity, while a curve which rises steeply first and then flattens indicates negative cooperativity. Surprisingly, despite decades of study, the effective cooperativity of allostery is still largely assessed in this way, through the shape of the binding function, which is sometimes quantified in terms of a sensitivity or Hill coefficient. However, the shape of the binding function, and any associated Hill coefficient, are measures which aggregate over conformations and

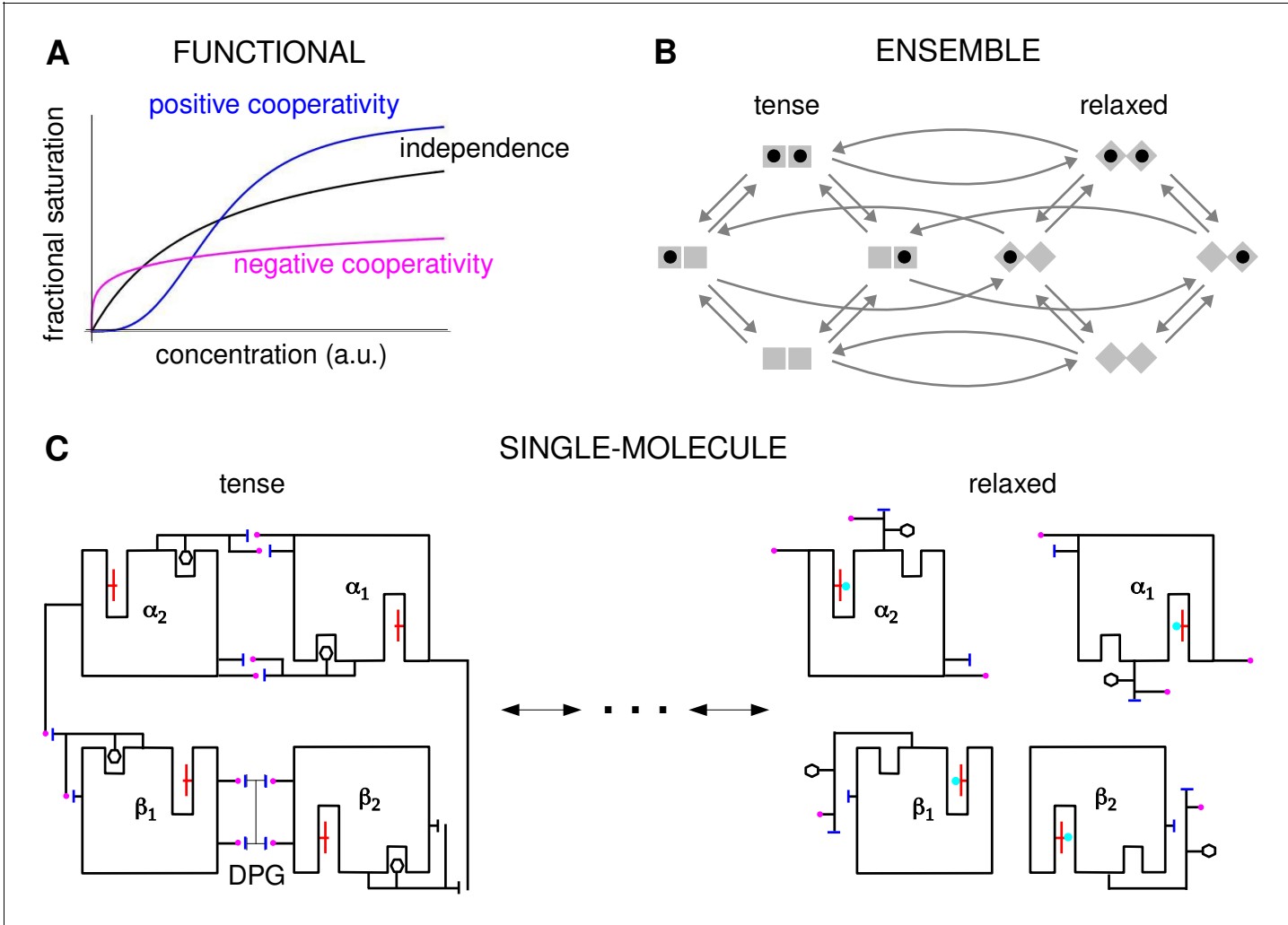

**Figure 2.** Cooperativity and allostery from three perspectives. (**A**) Plots of the binding function, whose shape reflects the interactions between binding sites, as described in the text. (**B**) The Monod, Wyman and Changeux (MWC) model with a population of dimers in two quaternary conformations, with each monomer having one binding site and ligand binding shown by a solid black disc. The two monomers are considered to be distinguishable, leading to four microstates. Directed arrows show transitions between microstates. This picture anticipates the graph-theoretic representation used later in this paper. (**C**) Schematic of the end points of the allosteric pathway between the tense, fully deoxygenated and the relaxed, fully oxygenated conformations of a single haemoglobin tetramer, $\alpha_1\alpha_2\beta_1\beta_2$, showing the tertiary and quaternary changes, based on Figure 4 of *Perutz, 1970*. Haem group (red); oxygen (cyan disc); salt bridge (positive, magenta disc; negative, blue bar); DPG is 2–3-diphosphoglycerate.

binding states, and they give little insight into how binding information is being integrated. To put it another way, the underlying free-energy landscape cannot be inferred from the shape of the binding function: as we will see below, different free-energy landscapes can give rise to indistinguishable binding functions. One of the contributions of this paper is to show how effective cooperativities can be quantified, providing thereby a set of parameters which collectively describe the allosteric free-energy landscape and placing allosteric information integration on a similar biophysical foundation to that provided by Pauling for direct interactions between two sites.

The MWC and KNF models are phenomenological: effective cooperativity arises as an emergent property of a conformational ensemble. This leaves open the question of how information is propagated between distant binding sites across a single molecule. This question was particularly relevant to haemoglobin, for which it had become clear that the haem groups were sufficiently far apart that direct interactions were implausible. Perutz's X-ray crystallography studies of haemoglobin revealed a pathway of structural transitions during cooperative oxygen binding which linked one conformation to another (*Figure 2C*), thereby relating the single-molecule viewpoint to the ensemble viewpoint (*Perutz, 1970*). These pioneering studies provided important justification for key aspects of the MWC model, which has endured as one of the most successful mathematical models in biology (*Changeux, 2013*; *Marzen et al., 2013*).

Allostery was initially thought to be limited to certain symmetric protein oligomers like haemoglobin and to involve only a few, usually two, conformations. But Cooper and Dryden's theoretical demonstration that information could be conveyed by fluctuations around a dominant conformation anticipated the emergence of a more dynamical perspective (*Cooper and Dryden, 1984*; *Henzler-Wildman and Kern, 2007*). At the single-molecule level, it has been found that binding information can be conveyed over long distances by complex atomic networks, of which Perutz's linear pathway (*Figure 2C*) is only a simple example (*Schueler-Furman and Wodak, 2016*; *Kornev and Taylor, 2015*; *Knoverek et al., 2019*; *Wodak et al., 2019*). These atomic networks may in turn underpin complex ensembles of conformations in many kinds of target molecules and allosteric regulation is now seen to be common to most cellular processes (*Nussinov et al., 2013*; *Changeux and Christopoulos, 2016*; *Motlagh et al., 2014*; *Lorimer et al., 2018*; *Wodak et al., 2019*; *Ganser et al., 2019*). The unexpected finding of widespread intrinsic disorder in proteins has been particularly influential in prompting a reassessment of the classical structure-function relationship, with conformations which may only be fleetingly present providing plasticity of binding to many partners (*Wrabl et al., 2011*; *Wright and Dyson, 2015*; *Berlow et al., 2018*).

However, while ensembles have grown greatly in complexity from MWC's two conformations and new theoretical frameworks for studying them have been introduced (*Wodak et al., 2019*), the quantitative analysis of information integration has barely changed beyond pairwise cooperativity. In the present paper, we will be particularly concerned with higher-order cooperativities (HOCs) in which multiple binding events collectively modulate another binding site (*Figure 1C*). Such higher-order effects can be quantified by association constants, $K_{i,S}$, where the set $S$ has more than one bound site. The size of $S$, denoted by $\#(S)$, is the order of cooperativity, so that pairwise cooperativity may be considered as HOC of order 1. For the example in *Figure 1C*, the ratio, $K_{5,\{2,4,6\}}/K_{5,\emptyset}$, defines the non-dimensional HOC of order 3 for binding to site 5, given that sites 2, 4 and 6 are already bound. The notation used here is essential to express such higher-order concepts.

Higher-order effects have been discussed in previous studies (*Dodd et al., 2004*; *Peeters et al., 2013*; *Martini, 2017*; *Gruber and Horovitz, 2018*) and treated systematically in the mutant-cycle strategy developed in *Horovitz and Fersht, 1990* and recently reviewed (*Carter, 2017*). The latter approach relies on perturbing residues or modules to unravel networks of energetic couplings within a macromolecule. It focusses on the single-molecule scale in contrast to the ensemble scale of the present paper (*Figure 2*). Mutant-cycle studies have confirmed the presence of substantial higher-order interactions underlying information propagation in proteins (*Jain and Ranganathan, 2004*; *Sadovsky and Yifrach, 2007*; *Carter et al., 2017*). The two approaches may be seen as different ways of analysing the free-energy landscape, as we explain in the Results.

HOCs were introduced in *Estrada et al., 2016*, where it was shown that experimental data on the sharpness of gene expression could not be accounted for purely in terms of pairwise cooperativities (*Park et al., 2019a*). In this context, the target molecule is the chromatin structure containing the relevant transcription factor (TF) binding sites and the analogue of the binding function is the steady-state probability of RNA polymerase being recruited, considered as a function of TF

concentration (*Estrada et al., 2016*; *Park et al., 2019a*). The Hunchback gene considered in *Estrada et al., 2016*, *Park et al., 2019a*, which is thought to have six binding sites for the TF Bicoid, requires HOCs up to order 5 to account for the data, under the assumption that the regulatory machinery is operating without energy expenditure at thermodynamic equilibrium. An important problem emerging from this previous work, and one of the starting points for the present paper, is to identify a molecular mechanism capable of implementing such HOCs.

In the present paper, we show that allosteric conformational ensembles can implement any pattern of effective HOCs. Accordingly, they can implement any form of information integration that is achievable at thermodynamic equilibrium. We work at the ensemble level (*Figure 2B*) using a graph-based representation of Markov processes developed previously (below). We introduce a systematic method of 'coarse graining', which is likely to be broadly useful for other studies. This allows us to define the effective HOCs arising from any allosteric ensemble, no matter how complex. These effective HOCs provide a quantitative language in which the integrative capabilities of any ensemble can be specified. We show, in particular, that allosteric ensembles can account for the experimental data on Hunchback mentioned above, which was the problem that prompted the present study. It is straightforward to determine the binding function from the effective HOCs, and we derive a generalised MWC formula for an arbitrary ensemble, which recovers the functional perspective. Our results subsume and generalise previous findings and clarify issues which have been present since the concept of allostery was introduced. Our graph-based approach further enables general theorems to be rigorously proved for any ensemble (below), in contrast to calculation of specific models which has been the norm up to now.

Our analysis raises questions about how effective HOCs are implemented at the level of single molecules, similar to those answered by Perutz for haemoglobin and the MWC model (*Figure 2C*). This important problem lies outside the scope of the present paper and requires different methods (*Wodak et al., 2019*), such as the mutant-cycle approach mentioned above (*Carter, 2017*). Our analysis is also restricted to ensembles which are at thermodynamic equilibrium without expenditure of energy, as is generally assumed in studies of allostery. Energy expenditure may be present in maintaining a conformational ensemble, for example, through post-translational modification, but the significance of this has not been widely appreciated in the literature. Thermodynamic equilibrium sets fundamental physical limits on information processing in the form of 'Hopfield barriers' (*Estrada et al., 2016*; *Biddle et al., 2019*; *Wong and Gunawardena, 2020*). Energy expenditure can bypass these barriers and substantially enhance equilibrium capabilities. However, the study of non-equilibrium systems is more challenging and we must defer analysis of this interesting problem to subsequent work (Discussion).

The integration of binding information through cooperativities leads to the integration of biological functions. Haemoglobin offers a vivid example of how allostery implements this relationship. This one target molecule integrates two distinct functions, of taking up oxygen in the lungs and delivering oxygen to the tissues, by having two distinct conformations, each adapted to one of the functions, and dynamically interchanging between them. In the lungs, with a higher oxygen partial pressure, binding cooperativity causes the relaxed conformation to be dominant in the molecular population, which thereby takes up oxygen; in the tissues, with a lower oxygen pressure, binding cooperativity causes the tense conformation to be dominant in the population, which thereby gives up oxygen. Evolution may have used this integrative strategy more widely than just to transport oxygen, and we review in the Discussion some of the evidence for an analogy between functional integration by haemoglobin and by gene regulation.

## Results

### Construction of the allostery graph

Our approach uses the linear framework for timescale separation (*Gunawardena, 2012*), details of which are provided in the 'Materials and methods' along with further references. We briefly outline the approach here.

In the linear framework, a suitable biochemical system is described by a finite directed graph with labelled edges. In our context, graph vertices represent microstates of the target molecule and graph edges represent transitions between microstates, for which the edge labels are the

instantaneous transition rates. A linear framework graph specifies a finite-state, continuous-time Markov process, and any reasonable such Markov process can be described by such a graph. We will be concerned with the probabilities of microstates at steady state. These probabilities can be interpreted in two ways, which reflect the ensemble and single-molecule viewpoints of *Figure 2*. From the ensemble perspective, the probability is the proportion of target molecules which are in the specified microstate, once the molecular population has reached steady state, considered in the limit of an infinite population. From the single-molecule perspective, the probability is the proportion of time spent in the specified microstate, in the limit of infinite time. The equivalence of these definitions comes from the ergodic theorem for Markov processes (*Stroock, 2014*). These different interpretations may be helpful when dealing with different biological contexts: a population of haemoglobin molecules may be considered from the ensemble viewpoint, while an individual gene may be considered from the single-molecule viewpoint. As far as the determination of probabilities is concerned, the two viewpoints are equivalent.

The graph representation may also be seen as a discrete approximation of a continuous energy landscape, as in *Figure 3*, in which the target molecule is moving deterministically on a high-dimensional landscape in response to a potential, while being buffeted stochastically through interactions with the surrounding thermal bath (*Frauenfelder et al., 1991*). In mathematics, this approximation goes back to the work of Wentzell and Freidlin on large deviation theory for stochastic differential equations in the low noise limit (*Ventsel' and Freidlin, 1970*; *Freidlin and Wentzell, 2012*). It has been exploited more recently to sample energy landscapes in chemical physics (*Wales, 2006*) and in the form of Markov State Models arising from molecular dynamics simulations (*Noé and Fischer, 2008*; *Sengupta and Strodel, 2018*). In this approximation, the vertices correspond to the minima of the free energy up to some energy cut-off, the edges correspond to appropriate limiting barrier crossings and the labels correspond to transition rates over the barrier.

The linear framework graph, or the accompanying Markov process, describes the time-dependent behaviour of the system. Our concern in the present paper is with systems which have reached a steady state of thermodynamic equilibrium, so that detailed balance, or microscopic reversibility, is satisfied. The assumption of thermodynamic equilibrium has been standard since allostery was introduced (*Koshland et al., 1966*; *Monod et al., 1965*) but has significant implications, as pointed out in the Introduction, and we will return to this issue in the Discussion. At thermodynamic equilibrium, we can dispense with dynamical information and work with what we call 'equilibrium graphs' (*Figure 3*). These are also directed graphs with labelled edges but the edge labels no longer contain dynamical information in the form of rates but rather ratios of forward to reverse rates. These ratios are determined by the minima of the free-energy landscape, with the equilibrium label on the edge from vertex $i$ to vertex $j$ being given by the formula in *Figure 3* . Free energy is often expressed relative to a reference level, as we will do below, so it will be convenient to write the equilibrium label from $i$ to $j$ as

$$\exp\left(-\frac{\Delta\Phi_j - \Delta\Phi_i}{k_B T}\right), \tag{1}$$

where $\Delta\Phi_u$ is the relative free-energy of vertex $u$, $k_B$ is Boltzmann's constant and $T$ is the absolute temperature (*Figure 3*). Note that if the edge in question involves components from outside the graph itself, such as a ligand which binds to $i$ to yield $j$, then the chemical potential of the ligand will contribute to the free energy. This contribution will manifest itself in the presence of a ligand concentration term in the edge label, as seen in *Figure 4*. The equilibrium edge labels are the only parameters needed at thermodynamic equilibrium and the free energies of the vertices can be recovered from them, up to an additive constant. From now on, in the main text, when we say 'graph', we will mean 'equilibrium graph'.

We explain such graphs using our main example. *Figure 4* shows the graph, $A$, for an allosteric ensemble, with multiple conformations $c_1, \cdots, c_N$ and multiple sites, $1, \cdots, n$, for binding of a single ligand ($n = 3$ in the example). The graph vertices represent abstract conformations with patterns of ligand binding, denoted $(c_k, S)$, where the index $k$ designates the conformation with $1 \leq k \leq N$, and $S \subseteq \{1, \cdots, n\}$ is the subset of bound sites. Directed edges represent transitions arising either from binding without change of conformation ('vertical' edges), $(c_k, S) \rightarrow (c_k, S \cup \{i\})$ where $i \notin S$, which occur for all conformations $c_k$, or from conformational change without binding ('horizontal' edges),

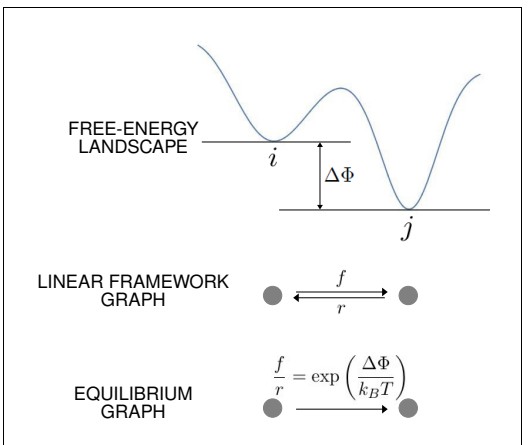

**Figure 3.** The free-energy landscape and corresponding graphs. From the top, a hypothetical one-dimensional free-energy landscape, showing two graph vertices, $i$ and $j$, as local minima of the free energy; the corresponding linear framework graph showing the edges between $i$ and $j$ with respective transition rates; the corresponding equilibrium graph whose edge label is the ratio of the transition rates, which is determined by the free-energy difference between the vertices (**Equation 1**).

$(c_k, S) \to (c_j, S)$ where $k \neq j$, which occur for all binding subsets $S$. Edges are shown in only one direction for clarity—when binding or unbinding is present, we use the direction of binding—but edges are always reversible, in accordance with thermodynamic equilibrium. Ignoring labels and thinking only in terms of vertices and edges, or 'structure', $A$ has a product form: the vertical sub-graphs, $A^{c_k}$, consisting of those vertices with conformation $c_k$ and all edges between them, all have the same structure and the horizontal sub-graphs, $A_S$, consisting of those vertices with binding subset $S$ and all edges between them, also all have the same structure (**Figure 4**). Structurally speaking, we can think of $A$ as the graph product (**Ahsendorf et al., 2014**) of the vertical subgraph $A^{c_1}$ and the horizontal subgraph $A_\emptyset$ (**Figure 4**).

In an allostery graph, 'conformation' is meant abstractly as any state for which binding association constants can be defined. It does not imply any particular atomic configuration of a target molecule nor make any commitments as to how the pattern of binding changes.

The product-form structure of the allostery graph reflects the 'conformational selection' viewpoint of MWC, in which conformations exist prior to ligand binding, rather than the 'induced fit' viewpoint of KNF, in which binding can induce new conformations. Considerable evidence now exists for conformational selection, in the form of transient, rarely populated conformations which exist prior to binding (**Tzeng and Kalodimos, 2011**). Induced fit may be incorporated within our graph-based approach by treating new conformations as always present but at extremely low probability. One of the original justifications for induced fit was that it enabled negative cooperativities, in contrast to conformational selection (**Koshland and Hamadani, 2002**), but we will show below that induced fit is not necessary for this and that negative HOCs arise naturally in our approach. Accordingly, the product-form structure of our allostery graphs is both convenient and powerful.

The edge labels are the non-dimensional ratios of the forward transition rate to the reverse transition rate; accordingly, the label for the reverse edge is the reciprocal of the label for the forward edge (Materials and methods). Labels may include the influence of components outside the graph, such as a binding ligand. For instance, the label for the binding edge $(c_k, S) \to (c_k, S \cup \{i\})$ is $xK_{c_k,i,S}$, where $x$ is the ligand concentration and $K_{c_k,i,S}$ is the association constant (**Figure 1A**), with dimensions of (concentration)$^{-1}$, as described in the Introduction. Horizontal edge labels are not individually annotated and need only be specified for the horizontal subgraph of empty conformations, $A_\emptyset$, since all other labels are determined by detailed balance (Materials and methods).

The graph structure allows HOCs between binding events to be calculated, as suggested in the Introduction. We will define this first for the 'intrinsic' HOCs which arise in a given conformation and explain in the next section how 'effective' HOCs are defined for the ensemble. In conformation $c_k$, the intrinsic HOC for binding to site $i$, given that the sites in $S$ are already bound, denoted $\omega_{c_k,i,S}$, is defined by normalising the corresponding association constant to that for binding to site $i$ when nothing else is bound (**Estrada et al., 2016**),

$$\omega_{c_k,i,S} = \frac{K_{c_k,i,S}}{K_{c_k,i,\emptyset}} . \tag{2}$$

HOCs are non-dimensional quantities. If $S$ has only a single site, say $S = \{j\}$, then the intrinsic HOC of order 1, $\omega_{c_k,i,\{j\}}$, is the classical pairwise cooperativity between sites $i$ and $j$. There is positive or negative intrinsic HOC if $\omega_{c_k,i,S} > 1$ or $\omega_{c_k,i,S} < 1$, respectively, and independence if $\omega_{c_k,i,S} = 1$ (**Figure 1A**).

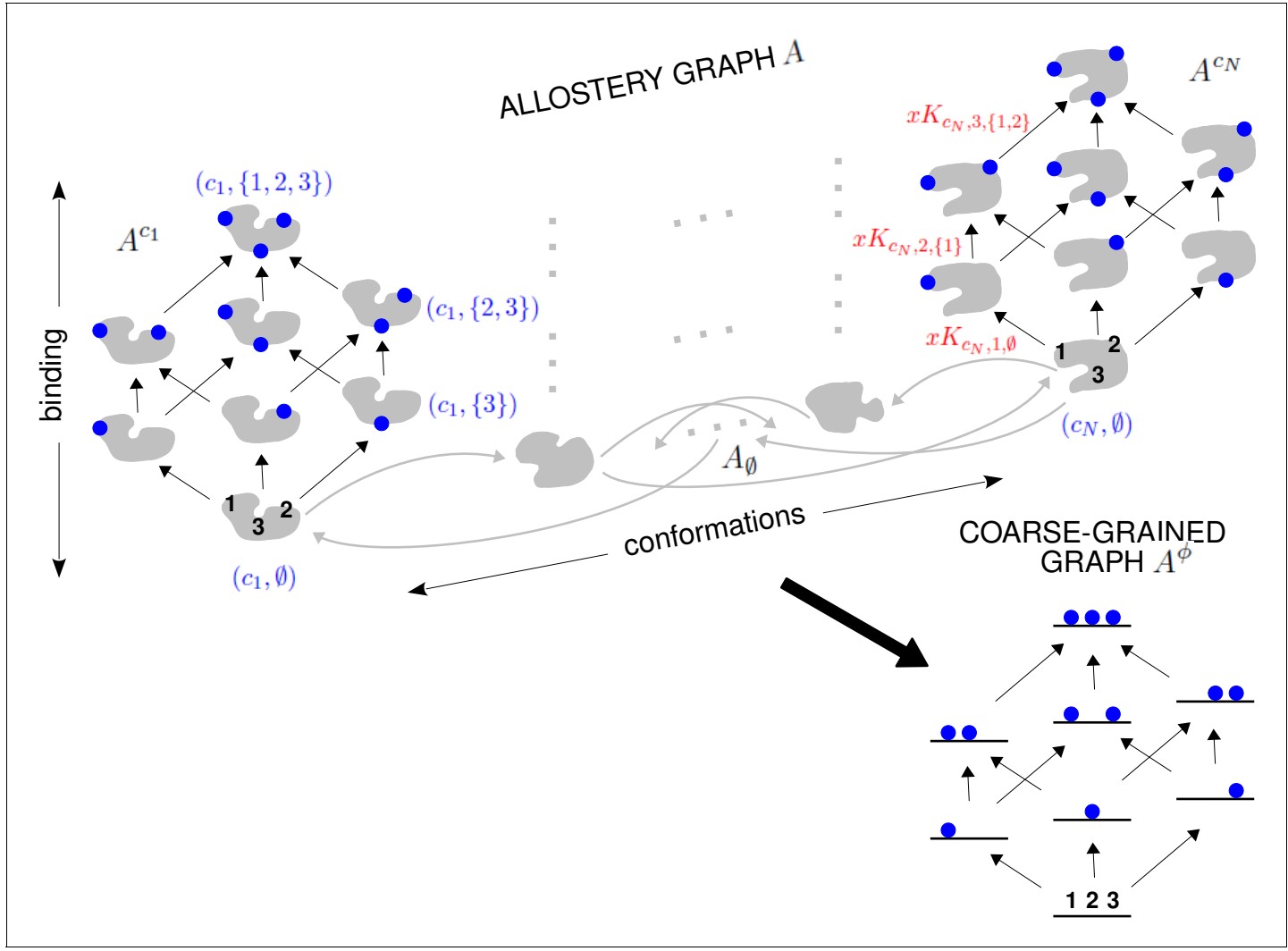

**Figure 4.** The allostery graph and coarse graining. A hypothetical allostery graph $A$ (top) with three binding sites for a single ligand (blue discs) and conformations, $c_1, \cdots, c_N$, shown as distinct grey shapes. Binding edges ('vertical' in the text) are black and edges for conformational transitions ('horizontal') are grey. Similar binding and conformational edges occur at each vertex but are suppressed for clarity. Note that edges are shown in only one direction but are always reversible. All vertical subgraphs, $A^{c_k}$, have the same structure, as seen for the vertical subgraphs, $A^{c_1}$ (left) and $A^{c_N}$ (right), and all horizontal subgraphs, $A_S$, also have the same structure, shown schematically for the horizontal subgraph of empty conformation, $A_\emptyset$, at the base. Example notation is given for vertices (blue font) and edge labels (red font), with $x$ denoting ligand concentration and sites numbered as shown for vertices $(c_1, \emptyset)$ and $(c_N, \emptyset)$. The coarse-graining procedure coalesces each horizontal subgraph, $A_S$, into a new vertex and yields the coarse-grained graph, $A^\phi$ (bottom right), which has the same structure as $A^{c_k}$ for any $k$. Further details in the text and the Materials and methods.

For any graph $G$, the steady-state probabilities of the vertices can be calculated from the edge labels. For each vertex, $v$, in $G$, the probability, $\mathrm{Pr}_v(G)$, is proportional to the quantity, $\mu_v(G)$, obtained by multiplying the edge labels along any directed path of edges from a fixed reference vertex to $v$. It is a consequence of detailed balance that $\mu_v(G)$ does not depend on the choice of path in $G$. This implies algebraic relationships among the edge labels. These can be fully determined from $G$ and independent sets of parameters can be chosen (Materials and methods). For the allostery graph, a convenient choice vertically is those association constants $K_{c_k,i,S}$ with $i$ less than all the sites in $S$, denoted $i<S$; horizontal choices are discussed in the Materials and methods but are not needed for the main text.

Since probabilities must add up to 1, it follows that

$$\mathrm{Pr}_v(G) = \frac{\mu_v(G)}{\sum_{u \in G} \mu_u(G)}. \tag{3}$$

*Equation 3* yields the same result as equilibrium statistical mechanics, with the denominator being the partition function for the thermodynamic grand canonical ensemble. Equilibrium statistical mechanics typically focusses only on vertices and uses their free energies as the fundamental parameters. Directed graphs of the form considered here were previously used in *Hill, 1966* and *Schnakenberg, 1976* to study systems away from thermodynamic equilibrium, where the graph edges become essential to represent entropy production (*Wong and Gunawardena, 2020*). We find that the graph remains just as useful at thermodynamic equilibrium because binding and unbinding are the fundamental mechanisms through which information is integrated and these mechanisms must be represented by graph edges. Indeed, as the next section shows, graphs are invaluable for formulating higher-order concepts.

Our specification of an allostery graph allows for arbitrary conformational complexity and arbitrary interacting ligands (we consider only one ligand here for simplicity), with the independent association constants in each conformation being arbitrary and with arbitrary changes in these parameters between conformations. Moreover, the abstract nature of 'conformation', as described above, permits substantial generality. Allostery graphs can be formulated to encompass the two conformations of MWC (*Marzen et al., 2013*), nested models (*Robert et al., 1987*), the fluctuations of *Cooper and Dryden, 1984* and more recent views of dynamical allostery (*Tzeng and Kalodimos, 2011*), the multiple domains of the Ensemble Allosteric Model developed by Hilser and colleagues (*Hilser et al., 2012*) and applied also to intrinsically disordered proteins (*Motlagh et al., 2012*), other ensemble models (*LeVine and Weinstein, 2015*; *Tsai and Nussinov, 2014*) and Markov State Models arising from molecular dynamics simulations (*Noé and Fischer, 2008*).

## Relationships between higher-order measures

As mentioned in the Introduction, a systematic approach to higher-order effects using mutant-cycle analysis was developed in *Horovitz and Fersht, 1990* and *Horovitz and Fersht, 1992* and widely used subsequently (*Carter, 2017*). The HOCs presented above were introduced in our previous work (*Estrada et al., 2016*), and the present paper is concerned not with HOCs per se, but with effective HOCs that arise from an allosteric ensemble, as will be described below. Nevertheless, it may still be helpful to explain the relationship between our HOCs and the higher-order couplings arising from mutant-cycle analysis. We are grateful to an anonymous reviewer for making this point to us. The material which follows may be of particular interest to those familiar with the relevant literature but is not required for the main results of the paper.

Both HOCs and higher-order couplings can be seen as different ways of analysing the underlying free-energy landscape. Both approaches make essential use of directed graphs to organise this landscape. *Figure 5A* shows the labelled equilibrium graph for ligand binding to three sites in a single conformation, while *Figure 5B* shows a directed graph of the kind used in *Horovitz and Fersht, 1990* for defining higher-order couplings for perturbations to three sites. The latter graphs are sometimes called 'boxes' (*Horovitz and Fersht, 1990*). We use 'sites' here for either individual residues or the modules described in *Carter, 2017*. Perturbations are typically mutations, such as replacement of an asparagine residue by alanine. The choice of replacement can make a difference to the results, but this is not usually depicted in graph representations like *Figure 5B*. The directed edges have rather different interpretations in the two examples in *Figure 5*: for the equilibrium graph in *Figure 5A,* a directed edge represents the biochemical process of ligand binding; for the coupling graph in *Figure 5B,* a directed edge represents an experimental perturbation. In both cases, the vertices have an associated free energy, denoted $\Delta\Phi_S$, where $S \subseteq \{1, \cdots, n\}$ is either the subset of bound sites in the equilibrium graph (*Figure 5A*) or the subset of perturbed sites in the coupling graph (*Figure 5B*). The $\Delta$ notation is conventionally used in the literature to signify a free-energy difference (*Equation 1*) or free energy relative to a chosen zero level. A frequent choice of zero is the free energy of empty binding or of the unperturbed state, in which case $\Delta\Phi_\emptyset = 0$, but we have not assumed this here. Note that the free energies of the equilibrium graph have a contribution from the ligand, which manifests itself in the dependence of the edge labels on the ligand concentration, $x$, while the free energies of the coupling graph do not. Despite this difference, the free energies provide in both cases the fundamental independent thermodynamic parameters, of which there are $2^n - 1$ for $n$ sites, in terms of which both HOCs and higher-order couplings can be rigorously defined.

The definition is easiest for HOCs. *Equation 1* tells us that the edge label, $xK_{i,S}$, is given by

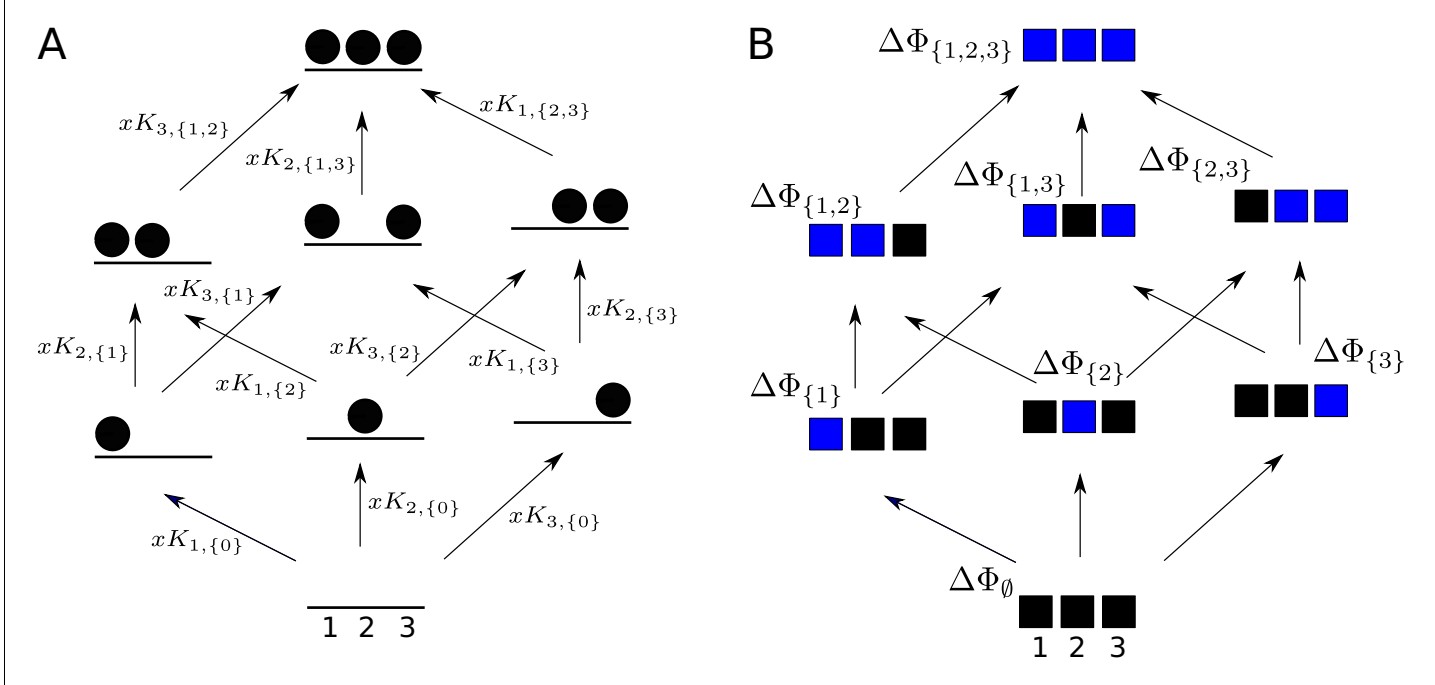

**Figure 5.** Graphs for defining higher-order measures. (A) Equilibrium graph, similar to those in *Figure 4*, for binding of a ligand to three sites on a single conformation, ordered as shown at the base, and annotated with edge labels. The single conformation has been omitted from subscripts for clarity. (B) Directed graph used to define higher-order couplings, for a macromolecule with three sites or modules (solid squares), ordered as shown at the base, with perturbations indicated by blue colour in place of black. Vertices are annotated with the corresponding free energy.

$$xK_{i,S} = \exp\left(-\frac{\Delta\Phi_{S\cup\{i\}} - \Delta\Phi_S}{k_B T}\right). \tag{4}$$

We omit the single conformation from subscripts for clarity. It follows from *Equation 2* that HOCs can be written in terms of free energies as follows:

$$\omega_{i,S} = \exp\left(-\frac{(\Delta\Phi_{S\cup\{i\}} - \Delta\Phi_S) - (\Delta\Phi_{\{i\}} - \Delta\Phi_\emptyset)}{k_B T}\right). \tag{5}$$

HOCs are non-dimensional quantities associated to graph edges. As noted above, there are algebraic relationships among them arising from detailed balance at thermodynamic equilibrium. An independent set of parameters is formed by restricting to those for which $i<S$, of which there are $2^n - n - 1$. Taken together with the $n$ 'bare' association constants for initial ligand binding, $K_{i,\emptyset}$, they form a complete set of $2^n - 1$ independent parameters for the free-energy landscape. It follows from *Equations 4 and 5* that these parameters can be used to recover the fundamental free energies, so that the two sets of parameters are mathematically equivalent.

Mutant-cycle studies often refer to both *Horovitz and Fersht, 1990* and *Horovitz and Fersht, 1992*, which present apparently different measures of higher-order coupling. The second of these papers introduces what we will refer to as the 'residual free energy' of a vertex and denote $\Delta\phi_S$. This is the free energy remaining at vertex $S$ after accounting for the contributions from all proper subsets of $S$. The residual free energy may be concisely defined recursively, starting from $\Delta\phi_\emptyset = \Delta\Phi_\emptyset$, by

$$\Delta\phi_S = \Delta\Phi_S - \left(\sum_{X\subset S}\Delta\phi_X\right). \tag{6}$$

We see from *Equation 6* that $\Delta\phi_{\{i\}} = \Delta\Phi_{\{i\}} - \Delta\Phi_\emptyset$ and that $\Delta\phi_{\{i,j\}} = \Delta\Phi_{\{i,j\}} - (\Delta\Phi_{\{i\}} + \Delta\Phi_{\{j\}}) + \Delta\Phi_\emptyset$. $\Delta\phi_S$ may be calculated directly from $\Delta\Phi_X$ but, as the previous example suggests, overlapping contributions of the actual free energies must be cancelled out (*Horovitz and Fersht, 1992*, *Equation 4*),

$$\Delta\phi_S = \sum_{0 \leq k \leq \#(S)} (-1)^{\#(S)-k} \left( \sum_{Y \subseteq S, \#(Y)=k} \Delta\Phi_Y \right).$$ (7)

To see why *Equation 7* is a consequence of *Equation 6*, note first that *Equation 7* gives the correct result for $S = \emptyset$. It may then be recursively checked by assuming it holds for $X \subset S$ and substituting into *Equation 6* to check that it holds for $S$. Each subset $Y \subset S$ contributes a term $\pm\Delta\Phi_Y$ arising from $\Delta\phi_X$ for each $X$ that satisfies $Y \subseteq X \subset S$. The sign of $\Delta\Phi_Y$ coming from *Equation 7* is $(-1)^{\#(X)-\#(Y)}$. These terms almost completely cancel each other out because, letting $p = \#(S) - \#(Y)$,

$$\sum_{Y \subseteq X \subset S} (-1)^{\#(X)-\#(Y)} = \sum_{V \subset S \backslash Y} (-1)^{\#(V)} = \sum_{0 \leq j < p} \binom{p}{j}(-1)^j = (-1)^{p+1}.$$

Taking into account the additional sign coming from *Equation 6*, we recover *Equation 7* for $S$. This proves recursively that *Equation 7* is the solution of *Equation 6* in terms of free energies.

We can go further to show how $\Delta\phi_S$ is expressed in terms of HOCs. For this, we must assume that $q = \#(S) > 1$. When $q = 1$, ligand binding contributes to $\Delta\phi_S$, but when $q > 1$ that is no longer the case, as we will see. Choose any site $i \in S$. The summation in *Equation 7* involves $2^q$ terms $\Delta\Phi_Y$. It can be reorganised into a sum of $2^{q-1}$ terms of the form $\pm(\Delta\Phi_{Z \cup \{i\}} - \Delta\Phi_Z)$, where $Z \subseteq S \backslash \{i\}$. The sign of these terms is given by the sign of $\Delta\Phi_{Z \cup \{i\}}$ coming from *Equation 7* and is therefore $(-1)^{\#(S)-\#(Z)-1}$. It is easy to see that, because $q > 1$, there must be equal numbers of $+1$ and $-1$ signs. It follows from *Equation 4* that

$$\exp\left(-\frac{\Delta\phi_S}{k_B T}\right) = \prod_{Z \subseteq S \backslash \{i\}} (x K_{i,Z})^{(-1)^{\#(S)-\#(Z)-1}},$$

where the double exponent just means that the right-hand side is a ratio in which those terms for which $\#(S) - \#(Z)$ is odd go in the numerator and those terms for which $\#(S) - \#(Z)$ is even go in the denominator. Using *Equation 2*, we can rewrite $K_{i,Z}$ as $K_{i,\emptyset}\omega_{i,Z}$. Since there are equal numbers of each sign, we can cancel each occurrence of $xK_{i,\emptyset}$ between numerator and denominator to yield a formula for residual free energies in terms of HOCs when $\#(S) > 1$:

$$\exp\left(-\frac{\Delta\phi_S}{k_B T}\right) = \prod_{Z \subseteq S \backslash \{i\}} (\omega_{i,Z})^{(-1)^{\#(S)-\#(Z)-1}}.$$ (8)

The choice of $i \in S$ in *Equation 8* is arbitrary. As an illustration of *Equation 8*, recalling from *Equation 5* that $\omega_{i,\emptyset} = 1$, we see that

$$\exp\left(-\frac{\Delta\phi_{\{i_1,i_2\}}}{k_B T}\right) = \omega_{i_1,\{i_2\}}, \quad \exp\left(-\frac{\Delta\phi_{\{i_1,i_2,i_3\}}}{k_B T}\right) = \frac{\omega_{i_1,\{i_2,i_3\}}}{\omega_{i_1,\{i_2\}}\omega_{i_1,\{i_3\}}}.$$ (9)

*Equations 8 and 9* show how the residual free energy is built up from binding at any given site to the hierarchy of subsets of the remaining sites.

Residual free energies can be thought of as a measure of collective synergy between sites (*Horovitz and Fersht, 1992*). They are associated to graph vertices and constitute $2^n - 1$ independent parameters, with no algebraic relationships between them. It follows from *Equations 6 and 7* that they are mathematically equivalent to the fundamental free energies. Residual free energies have also been independently described for other purposes in Equation 4 of *Martini, 2017*.

The higher-order couplings introduced in *Horovitz and Fersht, 1990* appear at first sight to be quite different from the residual free energies introduced in *Horovitz and Fersht, 1992*. The couplings are described by examples for low orders, as are typically encountered in practice (*Horovitz and Fersht, 1990*). We provide a general definition here by introducing a slightly more complex version. A coupling is associated to a pair, consisting of, first, a vertex, $Z \subseteq \{1, \cdots, n\}$, and, second, an ordered sequence of distinct sites, $(i_1, \cdots, i_k)$, none of which are in $Z$, so that $Z \cap \{i_1, \cdots, i_k\} = \emptyset$. The vertex $Z$ should be thought of as an 'offset' within the coupling graph and the sites, $i_1, \cdots, i_k$ as specifying an ordered sequence of perturbations undertaken around $Z$. Higher-

order couplings are conventionally used in the literature only for $Z = \emptyset$, but this more complex version is needed for the definition in *Equation 11* below. Associated to such a pair $Z, (i_1, \cdots, i_k)$ is a $k$th order coupling, which we will denote by $\Delta^k \gamma_{Z,(i_1, \cdots, i_k)}$. We start by defining the first-order coupling, $\Delta^1 \gamma_{Z,(i_1)}$, for any $Z$ satisfying the restriction above, in terms of the free energy,

$$\Delta^1 \gamma_{Z,(i_1)} = \Delta \Phi_{Z \cup \{i_1\}} - \Delta \Phi_Z. \tag{10}$$

With that in hand, we can define for $k \geq 2$, again for any $Z$ satisfying the restriction

$$\Delta^k \gamma_{Z,(i_1, \cdots, i_k)} = \Delta^{k-1} \gamma_{Z \cup \{i_k\}, (i_1, \cdots, i_{k-1})} - \Delta^{k-1} \gamma_{Z,(i_1, \cdots, i_{k-1})}, \tag{11}$$

where it is clear that $Z \cup \{i_k\}$ must be disjoint from $\{i_1, \cdots, i_{k-1}\}$, so that the right-hand side of *Equation 11* is recursively well defined. Unravelling *Equations 11 and 10*, we see that

$$\Delta^2 \gamma_{Z,(i_1, i_2)} = \Delta^1 \gamma_{Z \cup \{i_2\}, (i_1)} - \Delta^1 \gamma_{Z,(i_1)} = \Delta \Phi_{Z \cup \{i_1, i_2\}} - \Delta \Phi_{Z \cup \{i_2\}} - (\Delta \Phi_{Z \cup \{i_1\}} - \Delta \Phi_Z), \tag{12}$$

which corresponds when $Z = \emptyset$ to Equation 1 of *Horovitz and Fersht, 1990*. With some more work, it can be seen that *Equation 11* reproduces the $k = 3$ and $k = 4$ examples in *Horovitz and Fersht, 1990*. *Equation 12* expresses the intuition behind higher-order coupling, that it measures the effect of a perturbation relative to the unperturbed state, hierarchically for a sequence of perturbations.

It can be seen quite easily from *Equations 5 and 12* that

$$\exp\left(-\frac{\Delta^2 \gamma_{Z,(i_1, i_2)}}{k_B T}\right) = \frac{\omega_{i_1, Z \cup \{i_2\}}}{\omega_{i_1, Z}}. \tag{13}$$

We note from *Equation 13* that 'order' is counted differently between HOCs and conventional higher-order couplings: when $Z = \emptyset$, *Equation 13* relates a higher-order coupling with $k = 2$ to a HOC of order 1. Substituting *Equation 13* into *Equation 11* and continuing the recursion, we find that

$$\exp\left(-\frac{\Delta^3 \gamma_{Z,(i_1, i_2, i_3)}}{k_B T}\right) = \frac{\omega_{i_1, Z \cup \{i_2, i_3\}}}{\omega_{i_1, Z \cup \{i_2\}} \omega_{i_1, Z \cup \{i_3\}}},$$

at which point the similarity with *Equation 9* becomes evident and the pattern emerges. It can be shown by direct substitution in *Equation 11* that the following general formula holds, which expresses higher-order couplings in terms of HOCs for any $k \geq 2$:

$$\exp\left(-\frac{\Delta^k \gamma_{Z,(i_1, \cdots, i_k)}}{k_B T}\right) = \prod_{X \subseteq \{i_2, \cdots, i_k\}} (\omega_{i_1, Z \cup X})^{(-1)^{k-1-\#(X)}}. \tag{14}$$

Comparing *Equation 14* with *Equation 8* we see that, despite their very different definitions in *Equations 11 and 6*, conventional higher-order couplings are the same as residual free energies. Indeed, for $k \geq 1$,

$$\Delta^k \gamma_{\emptyset,(i_1, \cdots, i_k)} = \Delta \phi_{\{i_1, \cdots, i_k\}}. \tag{15}$$

*Equation 15* may seem strange because a higher-order coupling is defined in terms of an ordered sequence of perturbations, $(i_1, \cdots, i_k)$, while a residual free energy depends only on the subset of sites, $\{i_1, \cdots, i_k\}$, without respect to the order of sites. It is a consequence of detailed balance at thermodynamic equilibrium that the order in which the perturbations are undertaken does not matter. For example, it is clear from *Equation 12* that $\Delta^2 \gamma_{\emptyset,(i_1, i_2)} = \Delta^2 \gamma_{\emptyset,(i_2, i_1)}$. More generally, if $\rho$ is any permutation of the perturbed sites, so that $\rho$ is a bijective function, $\rho : \{i_1, \cdots, i_k\} \to \{i_1, \cdots, i_k\}$, then it can be shown that

$$\Delta^k \gamma_{Z,(i_1, \cdots, i_k)} = \Delta^k \gamma_{Z,(\rho(i_1), \cdots, \rho(i_k))}. \tag{16}$$

Note that *Equation 16* follows from *Equation 15* when $Z = \emptyset$. This property of invariance under

permutation is referred to as 'symmetry' in *Horovitz and Fersht, 1990* and is similar to the algebraic relations which give rise to the independent HOCs, $\omega_{i,S}$ with $i<S$, as described previously.

The equality between the higher-order couplings introduced in *Horovitz and Fersht, 1990* and the residual free energies introduced in *Horovitz and Fersht, 1992*, as described in *Equation 15*, is presumably well known to those in the field. It seems to be implicitly assumed in *Horovitz and Fersht, 1992*, but we have not found a clear statement of it in the literature. It would be difficult to formulate one in the absence of a general definition of higher-order coupling, as we have given in *Equation 11*. The formulas above may therefore be of some value in offering a rigorous treatment.

Each of the measures we have discussed, HOCs, residual free energies and higher-order couplings, offers a different way of analysing the free-energy landscape using the graphs in *Figure 5*. HOCs are associated to graph edges; residual free energies are associated to graph vertices; and higher-order couplings are associated to sequences of sites, at least when symmetries are ignored. As we have seen above, the three measures are mathematically equivalent. However, they are useful for different purposes. HOCs tell us about the integration of binding information; residual free energies capture the collective synergy between sets of sites; and higher-order couplings show how these same synergies can be extracted from a sequence of experimental perturbations. One advantage of HOCs is that they are non-dimensional quantities in terms of which it is straightforward to calculate the other measures. By doing so, we were able to show rigorously that higher-order couplings are also residual free energies (*Equation 15*).

Having explained how various higher-order measures are related to each other, we return to the question of how effective cooperativity arises from allosteric ensembles with multiple conformations. For this problem, HOCs are much easier to use than either residual free energies or higher-order couplings. With *Equations 8 and 14* now available, effective residual free energies or effective higher-order couplings may be calculated from the effective HOCs that we construct below, but we will not exploit this capability in the present paper.

## Coarse graining yields effective HOCs

As MWC showed, even if there is no intrinsic cooperativity in any conformation, an effective cooperativity can arise from the ensemble. This is usually detected in the shape of the binding function (*Figure 2A*). Here, we introduce a method of coarse graining through which effective cooperativities can be rigorously defined. We illustrate this for the allostery graph, $A$, and explain the general coarse-graining method in the Materials and methods. For allostery, the idea is to treat the horizontal subgraphs, $A_S$, as the vertices of a new coarse-grained graph, $A^\phi$, (*Figure 4*, bottom right). There is an edge between two vertices in $A^\phi$, if, and only if, there is an edge in $A$ between the corresponding horizontal subgraphs. It is not hard to see that $A^\phi$ is identical in structure to any of the vertical subgraphs $A^{c_k}$. We can think of $A^\phi$ as if it represents a single effective conformation to which ligand is binding, and we can index each vertex of $A^\phi$ by the corresponding subset of bound sites, $S$. The key point, as explained in detail in the Materials and methods, is that it is possible to assign labels to the edges in $A^\phi$ so that

$$\mathrm{Pr}_S(A^\phi) = \sum_{k=1}^{N} \mathrm{Pr}_{(c_k,S)}(A), \qquad (17)$$

with $A^\phi$ being at thermodynamic equilibrium under these label assignments. According to *Equation 17*, the probability of being in a coarse-grained vertex of $A^\phi$ is identical to the overall probability of being in any of the corresponding vertices of $A$. This is exactly the property a coarse graining should satisfy at steady state. It is not difficult to see why a procedure like this should work. The coarse-graining formula in *Equation 17* tells us the expected probability distribution on the coarse-grained graph, $A^\phi$. *Equation 3* can then be used to back out the equilibrium labels on the edges of $A^\phi$ which give rise to this probability distribution. We provide a more direct way of achieving the same result in *Equation 40*. This assignment of labels to $A^\phi$ is the only way to ensure *Equation 17* at equilibrium, so that the coarse graining is both systematic and unique. The Materials and methods gives a more careful treatment for coarse graining any linear framework graph, which may not itself be at thermodynamic equilibrium.

Our coarse-graining procedure offers a general method for calculating how effective behaviour emerges, at thermodynamic equilibrium, from a more detailed underlying mechanism. This procedure is likely to be broadly useful for other studies. We note that it applies only to the steady state. It does not provide a coarse graining of the underlying dynamics, which is a much harder problem.

Because $A^\phi$ resembles the graph for ligand binding at a single conformation, we can calculate HOCs for $A^\phi$—equivalently, effective HOCs for $A$—just as we did above, by normalising the effective association constants. Once the dust of calculation has settled (Materials and methods), we find that $A$ has effective association constants and effective HOCs:

$$K_{i,S}^\phi = \frac{\langle K_{c_k,i,S} \cdot \mu_S(A^{c_k}) \rangle}{\langle \mu_S(A^{c_k}) \rangle} \quad \text{and} \quad \omega_{i,S}^\phi = \frac{\langle K_{c_k,i,S} \cdot \mu_S(A^{c_k}) \rangle}{\langle K_{c_k,i,\emptyset} \rangle \langle \mu_S(A^{c_k}) \rangle}. \tag{18}$$

The quantity $\mu_S(A^{c_k})$ is calculated by multiplying labels over paths, as above, within the vertical subgraph $A^{c_k}$. The terms within angle brackets, of the form $\langle X(c_k) \rangle$, where $X(c_k)$ is some function over conformations $c_k$, denote averages over the steady-state probability distribution of the horizontal subgraph: $\langle X(c_k) \rangle = \sum_{1 \leq k \leq N} X(c_k) \mathrm{Pr}_{c_k}(A_\emptyset)$. The right-hand formula in *Equation 18* for the effective HOCs has a suggestive structure: it is an average of a product divided by the product of the averages. The effective parameters in *Equation 18* provide a biophysical language in which the integrative capabilities of any ensemble can be rigorously specified.

## Effective HOCs for MWC-like ensembles

The functional viewpoint is readily recovered from the ensemble. A generalised MWC formula can be given in terms of effective HOCs, from which the classical two-conformation MWC formula is easily derived (Materials and methods). Some expected properties of effective HOCs are also easily checked (Materials and methods). First, $\omega_{i,S}^\phi$ is independent of ligand concentration, $x$. Second, there is no effective HOC for binding to an empty conformation, so that $\omega_{i,\emptyset}^\phi = 1$. Third, if there is only one conformation $c_1$, then the effective HOC reduces to the intrinsic HOC, so that $\omega_{i,S}^\phi = \omega_{c_1,i,S}$.

More illuminating are the effective HOCs for the MWC model. We consider any conformational ensemble which is MWC-like: there is no intrinsic HOC in any conformation, so that $\omega_{c_k,i,S} = 1$ and $K_{c_k,i,S} = K_{c_k,i,\emptyset}$; and the bare association constants are identical at all sites, so that we can set $K_{c_k,i,\emptyset} = K_{c_k}$. There may, however, be any number of conformations, not just the two conformations of the classical MWC model. It then follows that $\omega_{i,S}^\phi$ depends only on the size of $S$, so that we can write $\omega_{i,S}^\phi$ as $\omega_s^\phi$, where $s = \#(S)$ is the order of cooperativity. *Equation 18* then simplifies to (Materials and methods)

$$\omega_s^\phi = \frac{\langle (K_{c_k})^{s+1} \rangle}{\langle K_{c_k} \rangle \langle (K_{c_k})^s \rangle}. \tag{19}$$

We see that, although there is no intrinsic HOC in any conformation, effective HOC of each order arises from the moments of $K_{c_k}$ over the probability distribution on $A_\emptyset$. In particular, *Equation 19* shows that the effective pairwise cooperativity is $\omega_1^\phi = \langle (K_{c_k})^2 \rangle / \langle K_{c_k} \rangle^2$.

In studies of G-protein coupled receptor (GPCR) allostery, Ehlert relates 'empirical' to 'ultimate' levels of explanation by a procedure similar to our coarse graining, but with only two conformations, and calculates a 'cooperativity constant' which is the same as $\omega_1^\phi$ (*Ehlert, 2016*). Gruber and Horovitz calculate 'successive ligand binding constants' for the two-conformation MWC model which are the same as effective association constants, $K_s^\phi$, (*Gruber and Horovitz, 2018*) (Materials and methods). To our knowledge, these are the only other calculations of effective allosteric quantities. We note that *Equation 19* applies to all HOCs, not just pairwise, and to any MWC-like ensemble, not just those with two conformations.

The classical MWC model yields only positive cooperativity (*Koshland and Hamadani, 2002*; *Monod et al., 1965*), as measured in the functional perspective (*Figure 2A*). We find that MWC-like ensembles yield positive effective HOCs of all orders. Strikingly, these effective HOCs increase with increasing order of cooperativity: provided $K_{c_k}$ is not constant over conformations (Materials and methods),

$$1 < \omega_1^\phi < \omega_2^\phi < \cdots < \omega_{n-1}^\phi. \tag{20}$$

This shows that ensembles with independent and identical sites, including the two-conformation MWC model, can effectively implement high orders and high levels of positive cooperativity. *Equation 20* is very informative, and we return to it in the Discussion.

It is often suggested that negative cooperativity requires a different kind of ensemble to those considered here, such as one allowing KNF-style induced fit (*Koshland and Hamadani, 2002*). However, if two sites are independent but not identical, so that $K_{c_k,1,\emptyset} \neq K_{c_k,2,\emptyset}$, then, with just two conformations, the effective pairwise cooperativity can become negative. Indeed, $\omega_{1,\{2\}}^\phi < 1$, if, and only if, the values of the association constants are not in the same relative order in the two conformations (Materials and methods). Negative effective cooperativity can arise from non-identical sites and does not need a special kind of ensemble.

## Integrative flexibility of ensembles

*Equation 18* shows that effective HOCs of any order can arise for a conformational ensemble but does not reveal what values they can attain. Can they vary arbitrarily? The question can be rigorously posed as follows. Suppose that we are considering $n$ binding sites and that numbers $\beta_i > 0$, for $1 \leq i \leq n$, and $\alpha_{i,S} > 0$, for $i < S$, are chosen at will. Does there exist a conformational ensemble such that the bare effective association constants satisfy $K_{i,\emptyset}^\phi = \beta_i$, and the independent effective HOCs satisfy $\omega_{i,S}^\phi = \alpha_{i,S}$?

To address this question, we assume that there is no intrinsic HOC, so as not to introduce cryptically what we want to generate. It follows that the sites cannot be identical, for otherwise *Equation 20* shows that integrative flexibility is impossible. Accordingly, the bare association constants, $K_{c_k,i,\emptyset}$ for $1 \leq i \leq n$, can be treated as $n$ free parameters in each conformation $c_k$. If there are $N$ conformations in the ensemble, then there are $N-1$ free parameters coming from the horizontal edges (Materials and methods). Dimensional considerations imply that the effective HOCs cannot take arbitrary values if $n(N-1) < 2^n - 1$. Conversely, we prove the following flexibility theorem: any pattern of values can be realised by an allosteric ensemble with no intrinsic cooperativity, to any required degree of accuracy, provided there are enough conformations with the right probability distribution and the right patterns of bare association constants.

To see why this is possible, we outline the argument here and give rigorous details in Theorem 1 in the Materials and methods. Other arguments may of course be possible and the details presented here should not be thought of as the only way for the results to hold. We will use an allostery graph $A$ whose conformations are indexed by subsets $T \subseteq \{1, \cdots, n\}$ and denoted $c_T$. Both binding subsets and conformations will then be indexed by subsets of $\{1, \cdots, n\}$. To avoid confusion, we will use $S$ to label binding subsets and $T$ to label conformations, so that a vertex of $A$ will be $(c_T, S)$. The allostery graph for the case $n = 2$ is shown in *Figure 6*. We will focus on the horizontal subgraph of empty conformations, $A_\emptyset$, because that is what is needed for calculating effective HOCs using *Equation 18*. We will take the reference vertex of $A_\emptyset$ to be $c_\emptyset$. Recall from what was explained previously that the product of the equilibrium labels along any path in $A_\emptyset$ from the reference vertex to the vertex $c_T$ is the quantity $\mu_{c_T}(A_\emptyset)$, in terms of which the steady-state probabilities of $A_\emptyset$ are given by *Equation 3*. Let $\lambda_T = \mu_{c_T}(A_\emptyset)$. These quantities are $2^n - 1$ free parameters whose values we are going to assign. They are more convenient for our purposes than an independent set of equilibrium labels for $A_\emptyset$. By *Equation 3*,

$$\text{Pr}_{c_T}(A_\emptyset) = \frac{\lambda_T}{\sum_{X \subseteq \{1, \cdots, n\}} \lambda_X}. \tag{21}$$

The other free parameters that we need are $n$ quantities, $\kappa_1, \cdots, \kappa_n > 0$, to which we will subsequently assign values, in terms of which we will define the intrinsic association constants. We will assume that the sites are independent in each conformation, so that all intrinsic HOCs of $A$ are 1. It follows that $K_{c_T,i,S} = K_{c_T,i,\emptyset}$. We then set $K_{c_T,i,\emptyset} = \kappa_i$ if $i \in T$, and $K_{c_T,i,\emptyset} = \varepsilon \kappa_i$ if $i \notin T$. Here, $\varepsilon$ is a small positive quantity which can be chosen to determine the degree of accuracy to which the $\beta_i$ and $\alpha_{i,S}$ are approximated. In the calculations which follow, we will only be interested in terms which do not

involve $\varepsilon$ as a factor. Because the sites are independent in each conformation, it follows that, in the vertical subgraph, $A^{c_T}$, at any conformation $c_T$, $\mu_S(A^{c_T}) = (\prod_{i \in S} \kappa_i) x^{\#(S)}$, whenever $S \subseteq T$. However, if $S \nsubseteq T$, then $\mu_S(A^{c_T})$ acquires factors of $\varepsilon$ and so $\mu_S(A^{c_T}) \approx 0$, where $\approx$ means simply that the related quantities become equal as $\varepsilon$ becomes very small. In this case, for our purposes, $\mu_S(A^{c_T})$ is negligible whenever $S \nsubseteq T$. *Figure 6* illustrates how this plays out in the allostery graph for $n = 2$.

To calculate the effective association constants, the left-hand formula in *Equation 18* shows that we must evaluate the averages $\langle K_{c_T, i, S} \cdot \mu_S(A^{c_T}) \rangle$ and $\langle \mu_S(A^{c_T}) \rangle$. Using *Equation 21*,

$$\langle \mu_S(A^{c_T}) \rangle = \sum_T \mu_S(A^{c_T}) \left( \frac{\lambda_T}{\sum_X \lambda_X} \right).$$

The only terms in the sum which do not involve $\varepsilon$ as a factor are those $T$ for which $S \subseteq T$. Furthermore, the definition of $\mu_S(A^{c_T})$ given above shows that these terms do not depend on $T$. Similarly, using *Equation 21* again,

$$\langle K_{c_T, i, S} \cdot \mu_S(A^{c_T}) \rangle = \sum_T K_{c_T, i, S} \cdot \mu_S(A^{c_T}) \left( \frac{\lambda_T}{\sum_X \lambda_X} \right)$$

and the only terms in the sum which do not involve $\varepsilon$ as a factor are those for which $S \cup \{i\} \subseteq T$. These terms also do not depend on $T$. It follows from *Equation 18* that

$$K_{i,S}^{\phi} = \frac{\langle K_{c_T, i, S} \cdot \mu_S(A^{c_T}) \rangle}{\langle \mu_S(A^{c_T}) \rangle} \approx \kappa_i \left( \frac{\sum_{S \cup \{i\} \subseteq T} \lambda_T}{\sum_{S \subseteq T} \lambda_T} \right), \tag{22}$$

where we have ignored all terms involving $\varepsilon$ as a factor.

*Equation 22* tells us two things. First, that the effective association constants are approximately proportional to the corresponding $\kappa$'s. Hence, if the proportionality constants, which depend only on the $\lambda_T$, are determined, we can choose the $\kappa_i$ so as to make the bare effective association constants $K_{i,\emptyset}^{\phi}$ approximately equal to $\beta_i$. Second, *Equation 22* tells us that the effective HOCs, $\omega_{i,S}^{\phi}$, are independent of the $\kappa_i$ and depend only on the $\lambda_T$,

$$\omega_{i,S}^{\phi} = \frac{K_{i,S}^{\phi}}{K_{i,\emptyset}^{\phi}} \approx \frac{\left( \sum_{\emptyset \subseteq T} \lambda_T \right) \left( \sum_{S \cup \{i\} \subseteq T} \lambda_T \right)}{\left( \sum_{\{i\} \subseteq T} \lambda_T \right) \left( \sum_{S \subseteq T} \lambda_T \right)}. \tag{23}$$

It remains for us to assign values to the $\lambda_T$ so that the effective HOCs become approximately equal to the $\alpha$'s.

To do this, assume that, for the conformation $c_T$, the subset $T$ is written as $T = \{i_1, \cdots i_k\}$, where the indices are in increasing order, $i_1 < i_2 < \cdots < i_k$. Because of this ordering, the quantities $\alpha_{i_j, \{i_{j+1}, \cdots, i_k\}}$ are given to us by hypothesis. Hence, we can define

$$\lambda_T = \alpha_{i_1, \{i_2, \cdots, i_k\}} \alpha_{i_2, \{i_3, \cdots, i_k\}} \cdots \alpha_{i_{k-1}, \{i_k\}} \delta^k. \tag{24}$$

Here, $\delta$ is another small positive quantity, similar to $\varepsilon$, which can be chosen to set the degree of accuracy to which the $\beta$'s and $\alpha$'s are approximated. As with $\varepsilon$, we will treat as negligible terms in which $\delta$ is a factor. *Figure 6* illustrates *Equation 24* for the case $n = 2$.

It can be seen from *Equation 24* that $\sum_{X \subseteq T} \lambda_T = \lambda_X (1 + U)$, where $U$ has a factor of $\delta$ and is therefore negligible as $\delta$ becomes very small, $U \approx 0$. It then follows from *Equation 23* that

$$\omega_{i,S}^{\phi} = \frac{(1 + U)\lambda_{S \cup \{i\}}(1 + U)}{\delta(1 + U)\lambda_S(1 + U)}, \tag{25}$$

where we have used $U$ as a generic symbol for quantities which are negligible as $\delta$ becomes very small. By *Equation 24*, $\lambda_{S \cup \{i\}} = \alpha_{i,S} \delta \lambda_S$, so that

$$\omega_{i,S}^{\phi} \approx \alpha_{i,S}. \tag{26}$$

This establishes part of what is required. For the other part, we can return to *Equation 22* and set

$$\kappa_i = \beta_i \left( \frac{\sum_{\{i\} \subseteq T} \lambda_T}{\sum_{\emptyset \subseteq T} \lambda_T} \right),$$

from which it follows from *Equation 22* that

$$K_{i,\emptyset}^{\phi} \approx \beta_i. \tag{27}$$

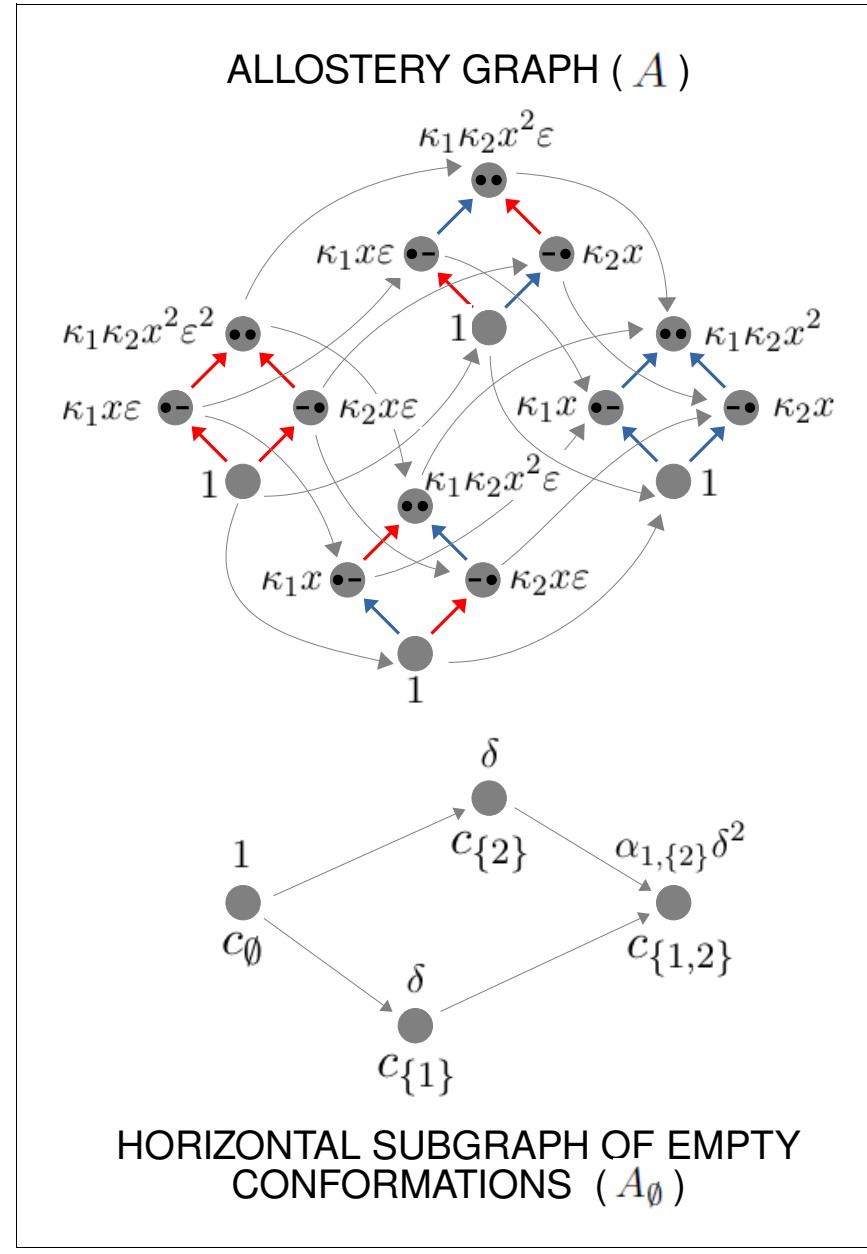

**Figure 6.** Example allostery graph for the flexibility theorem. There are $n = 2$ sites and $2^2 = 4$ conformations, giving a 16-vertex allostery graph (top). Vertices indicate a bound site with a solid black dot and an unbound site with a black dash. Sites are indexed in increasing order, $1, 2$, from left to right. The red vertical binding edges carry a factor of $\varepsilon$ in their equilibrium labels; the blue vertical binding edges do not, as specified in the text and *Equation 60*. The vertices of the allostery graph are annotated with the values of $\mu_S(A^{c_T})$, as specified in the text and *Equation 61*. The horizontal subgraph of empty conformations is shown at the bottom, with conformations indexed below each vertex by subsets of $\{1, 2\}$ and annotated above each vertex with the corresponding value of $\lambda_T$, as specified by *Equation 24*.

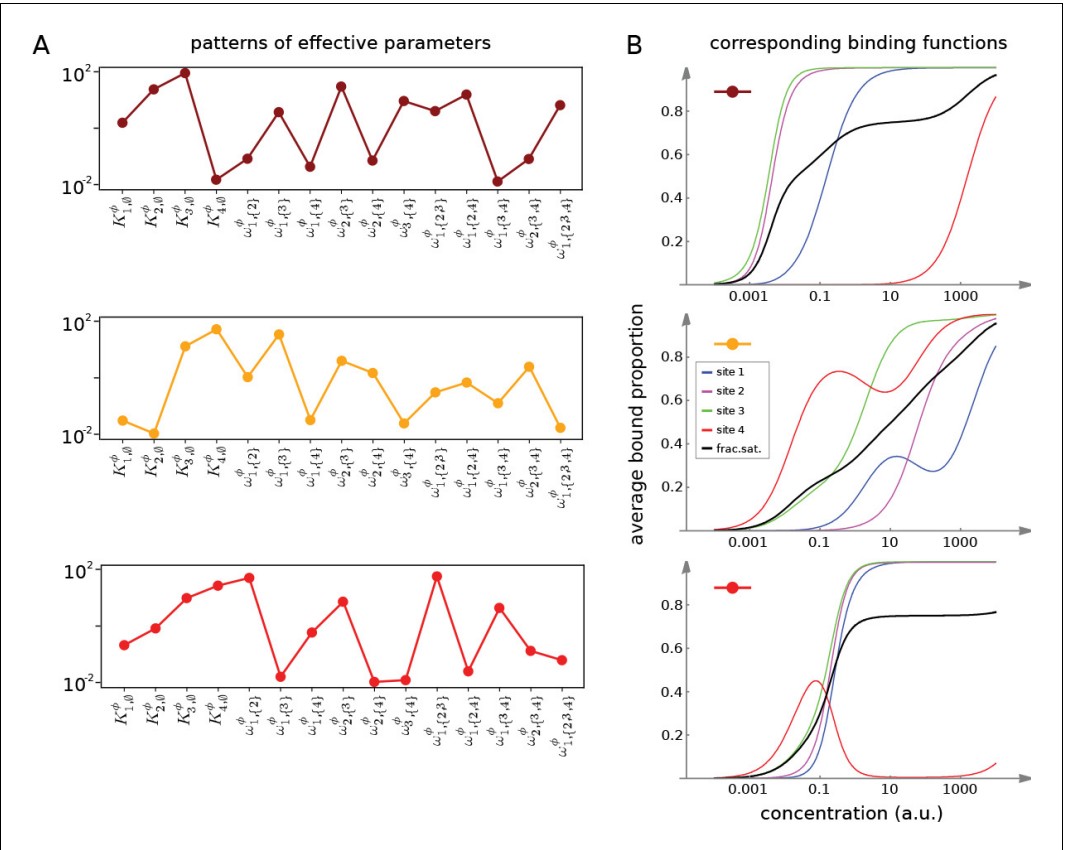

**Figure 7.** Integrative flexibility of allostery I. (**A**) Three choices of effective bare association constants, $K_{i,\emptyset}^{\phi}$, in arbitrary units of (concentration)$^{-1}$, and independent effective higher-order cooperativities , $\omega_{i,S}^{\phi}$, for $i<S$, in non-dimensional units, for ligand binding to four sites. Each example is coded by a colour (maroon, orange, red) and exhibits a different pattern of positive and negative effective HOCs. (**B**) Corresponding plots of average bound proportion at each site (colour coded as in middle inset) and the overall binding function, or fractional saturation, (black), calculated directly from the effective parameters. Note that the latter is always increasing; see the text for more details.

*Equations 26 and 27* show that the effective association constants and effective HOCs can take arbitrary positive values to any desired degree of accuracy, as determined by ε and δ. This establishes the flexibility theorem. The Materials and methods provides a more careful treatment in Theorem 1, which rigorously establishes the approximation as ε and δ become very small.

*Figures 7* and *8* together illustrate the flexibility theorem. *Figure 7A* shows three arbitrarily chosen patterns of effective parameters for a target molecule with four ligand binding sites. *Figure 7B* shows the corresponding overall binding functions (black curves) together with the individual site-specific binding functions (coloured curves). As a matter of thermodynamics, the overall binding function is always an increasing function of ligand concentration. In contrast, the site-specific binding functions may increase or decrease depending on the combinations of positive and negative effective HOCs in *Figure 7A*, and thereby show more clearly the complexity arising from those different combinations. The implementation of the effective parameters by an allosteric ensemble, as specified by the flexibility theorem, is illustrated in *Figure 8*. *Figure 8A* shows the allosteric ensemble for $n = 4$ sites as a product graph with 16 binding patterns and 16 conformations. *Figure 8B* shows the intrinsic association constants in each conformation coming from the proof of the flexibility theorem, to an accuracy of 0.01. *Figure 8C* confirms that this allosteric ensemble exactly reproduces the overall binding functions in *Figure 7B*.

In respect of the dimensional argument made previously, the allostery graph used in the proof above has $2^n - 1$ free parameters for $A_\emptyset$ and the $\kappa_1, \cdots, \kappa_n$ are a further $n$ free parameters, giving

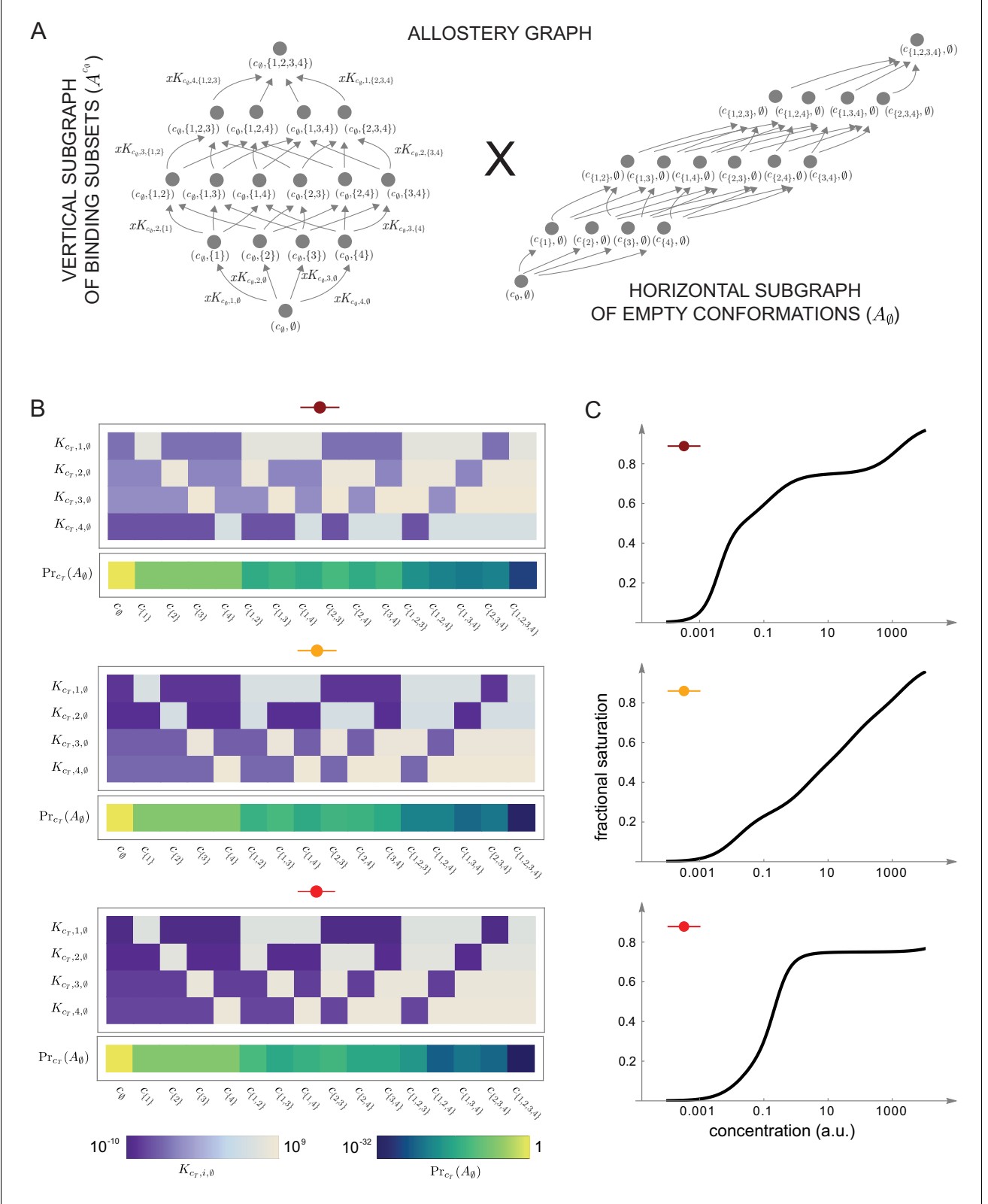

**Figure 8.** Integrative flexibility of allostery II. (**A**) The allostery graph, $A$, which implements the choices of effective higher-order cooperativities (HOCs) in *Figure 7*, shown as the product of the vertical subgraph of binding patterns at conformation $c_\emptyset$, $A^{c_\emptyset}$, and the horizontal subgraph of empty conformations, $A_\emptyset$. As required in the proof of the flexibility theorem, both conformations and binding subsets are indexed by subsets of $\{1, \cdots, n\}$, where $n$ is the number of binding sites. Since $n = 4$ for the effective HOCs in *Figure 7*, there are 16 binding subsets and 16 conformations,
*Figure 8 continued on next page*

*Figure 8 continued*

$c_\emptyset, \cdots, c_{\{1,2,3,4\}}$. (B) Intrinsic bare association constants, $K_{c_T,i,\emptyset}$, in each conformation, in arbitrary units of (concentration)$^{-1}$, and the probability distribution on the subgraph of empty conformations, $A_\emptyset$, for the allostery graph in (A), giving the three choices of effective parameters in *Figure 7A* to an accuracy of 0.01 (Materials and methods), colour coded on a log scale as shown in the respective legends below. (C) Overall binding functions for the three parameterised ensembles in (B) (black curves), overlaid on the overall binding functions from *Figure 7B* (red curves), which were calculated from the effective parameters. The match is too close for the red curves to be visible. Numerical values are given in the Materials and methods. Calculations were undertaken in a Mathematica notebook, available on request.

$2^n - 1 + n$ free parameters in total. This is more than the minimal required number of $2^n - 1$ but not by much. It remains an interesting open question whether a conformational ensemble can be constructed, perhaps with more free parameters, which gives the effective HOCs exactly, rather than only approximately. One consequence of the definitions of $K_{c_T,i,\emptyset}$ and of $\lambda_T$ in *Equation 24* is that the parameters of the allosteric ensemble become exponentially small, as is evident for the examples in *Figure 8B*. Another interesting question is whether alternative constructions can be found which do not exhibit such a broad range of parameter values. Irrespective of these questions, the proof given above confirms that there is no fundamental biophysical limitation to achieving any pattern of values to any desired degree of accuracy. Accordingly, a central result of the present paper is that sufficiently complex allosteric ensembles can implement any form of information integration that is achievable without energy expenditure.

## Allosteric ensembles for Hill functions

As mentioned in the Introduction, the starting point for the present paper was to account for experimental data on gene expression. Studies in *Drosophila* have shown that the Hunchback gene, in response to the maternal TF Bicoid, is sharply expressed in a way that is well fitted, after appropriate normalisation, to a Hill function, $\mathcal{H}_h(x) = x^h/(1 + x^h)$. This sharp expression underlies the initial patterning of anterior-posterior stripes in the early Drosophila embryo. Estimated values for the Hill coefficient, $h$, vary depending on the experimental construct and time of measurement but are typically in the range $4 \le h \le 8$ during early nuclear cycle 14 (*Tran et al., 2018*). The relevant promoter is believed to have $n = 6$ Bicoid binding sites, and the mechanistic basis for the sharpness is the subject of considerable interest. We showed in previous work that, if the promoter was assumed to have six Bicoid binding sites and to be operating at thermodynamic equilibrium, then the highest Hill coefficient that could be achieved of $h = 6$, at the so-called Hopfield barrier, required HOCs for Bicoid binding of order up to 5 (*Estrada et al., 2016*). In particular, pairwise cooperativities, which had previously been invoked to account for the sharpness (*Gregor et al., 2007*), are not sufficient to explain the data. Left open by this previous work was a molecular mechanism which could create the high-order HOCs required for Hill functions. We have seen above that allosteric ensembles can create any pattern of HOCs, so it is natural to ask if there are allosteric ensembles which yield good approximations to Hill functions.

We implemented a numerical optimisation algorithm to find binding functions which approximated Hill functions (Materials and methods). Hill functions are naturally normalised so that $\mathcal{H}_h(1) = 0.5$, so we followed the procedure introduced previously (*Estrada et al., 2016*) of normalising concentration to its value at half-maximum: if the normalised binding function is denoted $f(x)$, then $f(1) = 0.5$. *Figure 9* shows results for an allosteric ensemble with four conformations for ligand binding to six sites. The ensemble has no intrinsic cooperativity in any conformation, so that $K_{c_k,i,S} = K_{c_k,i,\emptyset}$ for any binding subset $S \subseteq \{1, \cdots, 6\}$, while the bare association constants, $K_{c_k,i,\emptyset}$, differ between the conformations (*Figure 9B*). This gives $4 \times 6 = 24$ free parameters together with an additional three free parameters for the independent equilibrium labels on the horizontal subgraph $A_\emptyset$ (*Figure 9A*). We limited the parameter ranges so that the $K_{c_k,i,\emptyset}$ were in the range $[10^{-4}, 10^4]$ and the equilibrium labels of $A_\emptyset$ were in the range $[10^{-6}, 10^6]$. With these settings, it was not difficult to find normalised binding functions which are very well fitted by the Hill function, $\mathcal{H}_h(x)$, for Hill coefficients $h = 4$, 5 and 6 (*Figure 9D*).

We were able to find multiple sets of parameters which yielded excellent fits; *Figure 9* shows two representative examples for each Hill coefficient. It is evident that very different numerical ensembles (*Figure 9B*) can give almost identical binding functions (*Figure 9D*). This reinforces the point made

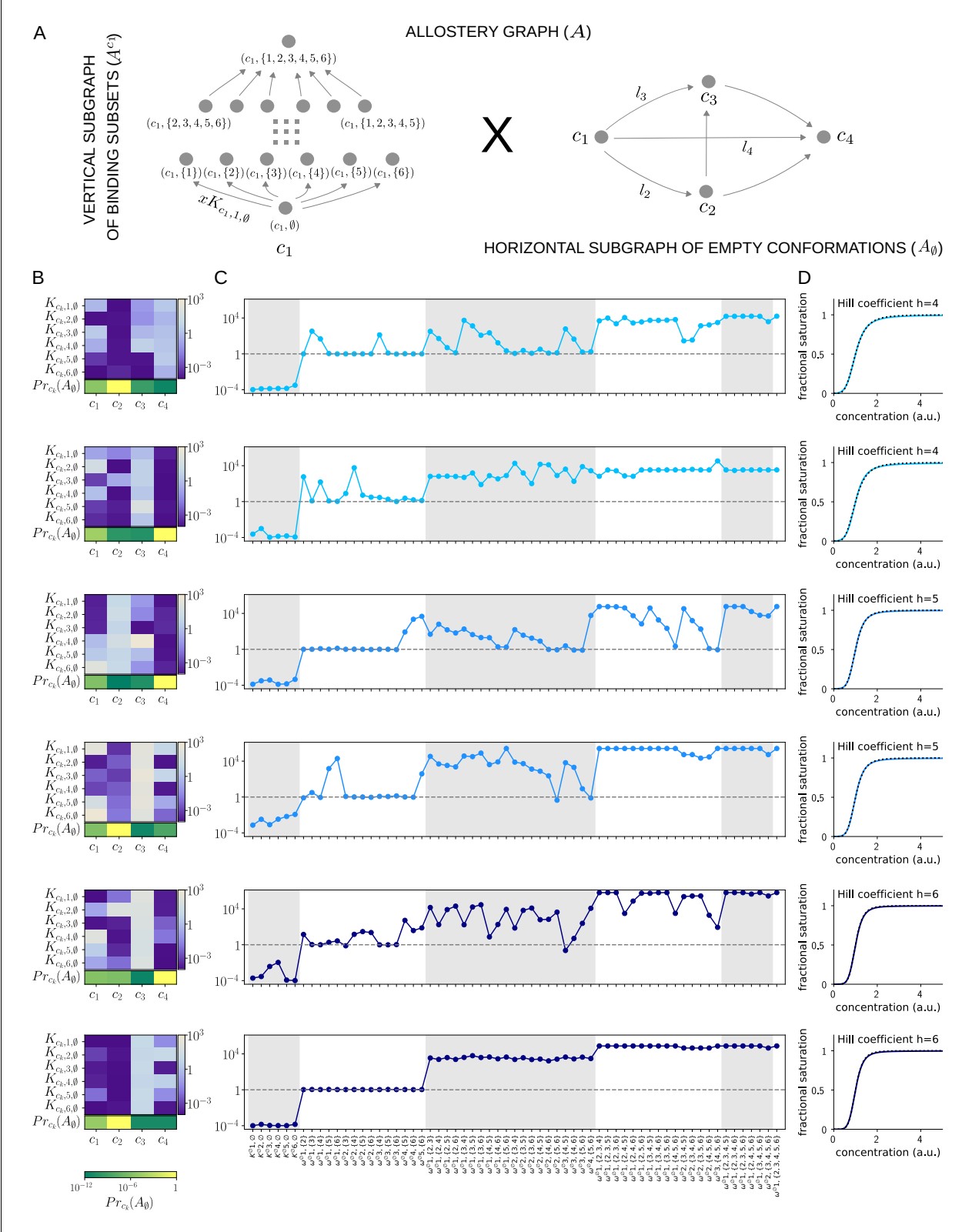

**Figure 9.** Allosteric ensembles for Hill functions. (A) Allostery graph, $A$, for representing Hill functions with six binding sites and four conformations, shown as the product of the vertical subgraph $A^{c_1}$ of binding subsets and the horizontal subgraph $A_\emptyset$ of empty conformations. Some vertices are annotated and some edges are labelled; the edge labels, $l_2, l_3$ and $l_4$, on the horizontal subgraph are the independent labels coming from a spanning tree used in the algorithm described in the text. (B) Intrinsic bare association constants in each conformation, in arbitrary units of (concentration)$^{-1}$

*Figure 9 continued on next page*

*Figure 9 continued*

colour coded in the vertical bar on the right, and the probability distribution on the subgraph of empty conformations, colour coded in the horizontal bar at the bottom. (**C**) Corresponding effective association constants in arbitrary units of (concentration)$^{-1}$ and the non-dimensional independent effective higher-order cooperativities arising from the ensemble. (**D**) Corresponding binding functions (blue curves) overlaid on the Hill function $\mathcal{H}_h(x)$ (black dashes) with the indicated Hill coefficient, $h$. Two sets of parameter values are shown, with the same shade of blue, for each Hill coefficient $h = 4$, 5 and 6.

in the Introduction that the binding function, or some associated measure of its shape, such as a Hill coefficient, are aggregate measures which give little insight into how binding information is being integrated. For this, the patterns of effective parameters provide more detailed information. As can be seen from *Figure 9C*, effective HOCs of all orders up to 5 are needed for each Hill function, as suggested previously (*Estrada et al., 2016*), with predominantly positive effective HOCs, $\omega_{i,S}^{\phi} > 1$, and varying amounts of independence, $\omega_{i,S}^{\phi} = 1$.

It is interesting to ask what role the size of the ensemble plays in approximating Hill functions. We cannot give a definitive answer but can make some observations. We were able to approximate $\mathcal{H}_6$ with a two-conformation ensemble with six sites but only with much wider parametric ranges. It was also more difficult in terms of optimisation time to find a good fit, and we did not find multiple fits. This suggests that the larger the ensemble the easier it is to approximate Hill functions with limited parameter ranges. It is also conceivable that the size of the ensemble may have to increase with the number of binding sites to retain control over the parametric ranges. We must leave such issues to subsequent work. While our results are numerical, and therefore limited to the ensemble we have analysed, it seems clear that allosteric ensembles provide a molecular mechanism that can closely approximate Hill functions with the required high orders of effective cooperativity, thereby providing a solution to our original question. Since Hill functions are widely used to fit data, the potential for an underlying allosteric mechanism may be broadly useful.

## Discussion

Jacques Monod famously described allostery as 'the second secret of life' (*Ullmann, 2011*). It is only relatively recently, however, that the prescience of his remark has been appreciated and the wealth of conformational ensembles present in most cellular processes has been revealed (*Changeux and Christopoulos, 2016*; *Motlagh et al., 2014*; *Nussinov et al., 2013*).

The present paper seeks to expand the existing allosteric perspective by providing a biophysical foundation for information integration by conformational ensembles. *Equation 48* and *Equation 49* in the Materials and methods (*Equation 18* above) provide for the first time a rigorous definition of effective, higher-order quantities—the association constants, $K_{i,S}^{\phi}$, and cooperativities, $\omega_{i,S}^{\phi}$,—arising from any ensemble. Since our methods are equivalent to those of equilibrium statistical mechanics (Material and methods), these definitions correctly aggregate the free-energy contributions which emerge in the ensemble from ligand binding to a conformation, intrinsic cooperativity within a conformation and conformational change. As noted above, our results encompass recent work on effective properties of the classical, two-conformation MWC ensemble—for pairwise cooperativity (*Ehlert, 2016*) and higher-order association constants (*Gruber and Horovitz, 2018*)—but they hold more generally for ensembles of arbitrary complexity with any number of conformations, including those with intrinsic cooperativities.

The effective quantities introduced here provide a language in which the integrative capabilities of an ensemble can be rigorously expressed. To begin with, the overall binding function can be determined in terms of the effective quantities through a generalised MWC formula (Materials and methods), thereby recovering the functional viewpoint (*Figure 2A*) from the ensemble viewpoint (*Figure 2B*). This generalised MWC formula reduces to the usual MWC formula for the classical two-conformation MWC model (*Equation 55*). We also clarify issues which had been difficult to understand in the absence of a quantitative definition of effective quantities. We find that the classical MWC model exhibits effective HOCs of any order and that these are always positive. In other words, binding always encourages further binding. Moreover, these effective HOCs increase strictly with increasing order (*Equation 20*), so that the more sites which are bound, the greater the

encouragement to further binding. We see that HOC has always been present, even for oxygen binding to haemoglobin, albeit unrecognised for lack of an appropriate quantitative definition. *Equation 20* confirms in a more precise way the long-standing realisation from the functional perspective that the MWC model exhibits only positive cooperativity; at the same time it succinctly expresses the rigidity and limitations of this model.

It is often stated in the allostery literature that negative cooperativity requires induced fit, in which binding induces conformations which are not present prior to binding. This view goes back to Koshland, who pointed to the emergence of negative cooperativity in the KNF model of allostery, which allows induced fit, and contrasted that to the positive cooperativity of the MWC model, which assumes conformational selection (*Koshland and Hamadani, 2002*). Our language of effective quantities permits a more discriminating analysis. It confirms, as just pointed out, that the classical MWC model exhibits only positive effective HOCs but also shows that induced fit is not required for negative effective HOC, which can arise just as readily from conformational selection (Materials and methods). What is required is not a different kind of ensemble but, rather, binding sites that are not identical.

Our main result, on the flexibility of conformational ensembles, shows that positive and negative HOCs of any value can occur in any pattern whatsoever, provided that the conformational ensemble is sufficiently complex, with enough conformations (*Figure 8*). Since the effective quantities provide a complete parameterisation of an ensemble at thermodynamic equilibrium, we see that conformational ensembles can implement any form of information integration that is achievable without external sources of energy. In particular, allosteric ensembles can be found whose binding functions closely approximate Hill functions (*Figure 9*), thereby answering the question which prompted this study, as to how such functions might arise in gene regulation.

Eukaryotic gene regulation is one of the most complex forms of cellular information processing (*Wong and Gunawardena, 2020*). Information from the binding of multiple TFs at many sites, often widely distributed across the genome in distal enhancer sequences, must be integrated to determine whether, and in what manner, a gene is expressed. The results of the present paper offer a way to think further about how such integration takes place (*Tsai and Nussinov, 2011*). We focus on gene regulation, but our results may also be useful for analysing other mechanisms of information integration, such as GPCRs (*Thal et al., 2018*).

As pointed out in the Introduction, haemoglobin solves the problem of integrating two quite different physiological functions—picking up oxygen in the lungs and delivering oxygen to the tissues—by having two conformations, each adapted to one of these functions, and dynamically interconverting between them (*Figure 10A*). The effective cooperativity of oxygen binding ensures that the appropriate conformation dominates the ensemble in the distinct contexts of the lungs, where oxygen is abundant, and the tissues, where oxygen is scarce, so that oxygen is transferred from the former to the latter.

Genes have to be regulated to achieve yet more elaborate forms of integration, with the same gene being expressed differently in different contexts. Such pleiotropy is particularly evident in developmental genes (*Bolt and Duboule, 2020*) but usually occurs in distinct cells within the developing organism. The same gene is present in these cells, but it may be difficult to know whether the corresponding regulatory machineries are also the same. More directly suitable examples for the present discussion arise in individual cells exposed to distinct stimuli (*Molina et al., 2013*; *Kalo et al., 2015*; *Lin et al., 2015*), which may be particularly the case for neurons or cells of the immune system (*Marco et al., 2020*; *Smale et al., 2013*).

Depending on the input pattern of TFs present in a given cellular context (*Figure 10B*, left), a gene may be expressed in a certain way, as a distribution of splice isoforms, each with an overall level of mRNA expression and a pattern of stochastic bursting (*Lammers et al., 2020*; *Figure 10B*, right). A different input pattern of TFs may elicit a different mRNA output. Our results suggest that one way in which these different input-output relationships could be integrated in the workings of a single gene is through allostery of the overall regulatory machinery. An allosteric analogy in gene regulation was previously made by *Mirny, 2010*, building upon observations of indirect cooperativity between TFs that were mediated by nucleosomes (*Miller and Widom, 2003*). In the allosteric analogy, TF binding to DNA takes place in one of two conformations—nucleosome present or absent—which dynamically interchange, leading to the classical MWC model. Here, we build upon Mirny's idea to suggest that not only indirect cooperativity but also, more broadly, information integration

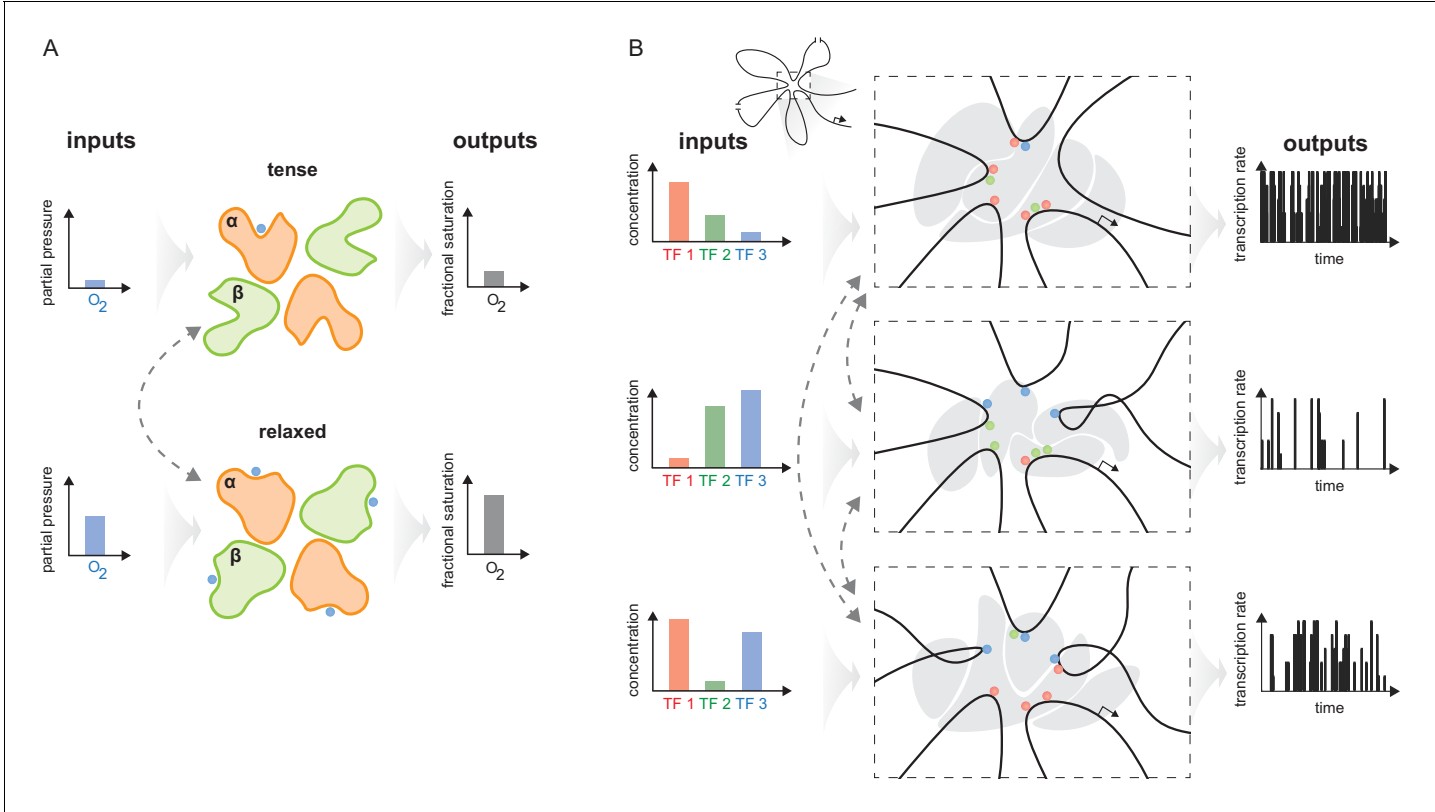

**Figure 10.** The haemoglobin analogy in gene regulation. (**A**) The two conformations of haemoglobin are each adapted to one of the two input-output functions which haemoglobin integrates to solve the oxygen transport problem. These conformations dynamically interchange in the ensemble (grey dashed arrows). (**B**) The gene regulatory machinery couples input patterns of transcription factors (TFs) (left) to output patterns of stochastic expression of mRNA splice isoforms (right, showing bursting patterns of one isoform). Our results suggest that a sufficiently complex conformational ensemble, built out of chromatin, TFs, co-regulators and phase-separated condensates (centre, grey shapes in three distinct conformations), could integrate these functions at a single gene in an analogous way to haemoglobin. Chromatin is represented by the thick black curve, whose looped arrangement around the promoter is shown schematically (top).

may be accounted for by the conformational dynamics of the gene regulatory machinery. The latter comprises not just individual nucleosomes but whatever other molecular entities are implicated in conveying information from TF binding sites to RNA polymerase and the transcriptional machinery (*Figure 10B*, centre), as discussed below. If this hypothesis is correct, then the flexibility result tells us that the overall regulatory conformational ensemble must exhibit sufficient complexity to implement the integration of binding information.

Studies of individual regulatory components have revealed many levels of conformational complexity. DNA itself exhibits conformational changes in respect of TF binding (*Kim et al., 2013*). Nucleosomes are moved or evicted to alter chromatin conformation and DNA accessibility (*Mirny, 2010*; *Voss and Hager, 2014*). TFs, in particular, show high levels of intrinsic disorder compared to other classes of proteins (*Liu et al., 2006*), especially in their activation domains, and these disordered regions exhibit dynamic multivalent interactions characteristic of higher-order effects (*Chong et al., 2018*; *Clark et al., 2018*). Hub TFs like p53 exhibit high levels of conformational flexibility in the context of specific DNA binding (*Demir et al., 2017*). Transcriptional co-regulators, which do not directly bind DNA but are recruited there by TFs, exhibit substantial conformational complexity: CBP/p300 has multiple intrinsically disordered regions which facilitate higher-order cooperative interactions (*Dyson and Wright, 2016*), while the Mediator complex exhibits quite remarkable conformational changes upon binding to TFs (*Allen and Taatjes, 2015*). Transcription initiation sub-complexes such as TFIID, which help assemble the transcriptional machinery, show conformational plasticity (*Nogales et al., 2017*), while the C-terminal domain of RNA Pol II, which is repetitive and intrinsically disordered, shows surprising local structural heterogeneity (*Portz et al.,*

*2017*). The significance of RNA conformational dynamics during transcription is becoming clearer (*Ganser et al., 2019*). Finally, transcription may also be regulated within larger-scale entities, such as transcription factories (*Edelman and Fraser, 2012*), phase-separated condensates (*Sabari et al., 2018*) and topological domains (*Benabdallah and Bickmore, 2015*). The role of such entities remains a matter of debate (*Mir et al., 2019*), but they may play a significant role in conveying information over long genomic distances between distal enhancers and target promoters (*Furlong and Levine, 2018*). From the perspective taken here, in view of their size and extent, they may exhibit conformational dynamics on longer timescales.

These various findings have emerged largely independently of each other. They indicate the presence of many conformations of components of the gene regulatory machinery, with these components dynamically interchanging on varying timescales. The collective effect of these coupled dynamics is difficult to predict but we can hazard some guesses. It has been suggested, for example, that multi-protein complexes like Mediator couple the conformational repertoires of their component proteins into complex allosteric networks for processing information (*Lewis, 2010*). From an ensemble viewpoint, if component $X$ has $m$ conformations and component $Y$ has $n$ conformations, we might naively expect that the coupling of $X$ and $Y$ in a complex yields roughly $mn$ conformations. Following this multiplicative logic for the many components involved in eukaryotic gene regulation, from DNA itself to condensates and domains, suggests that the gene regulatory machinery has enormous conformational capacity with a deep hierarchy of timescales.

In making the analogy to haemoglobin, it is the conformational dynamics which implements the transfer of information from upstream TF inputs to downstream gene output. In any given cellular context, as determined by the input pattern of TFs, we may expect one, or perhaps a few, overall regulatory conformations which are well-adapted to generate the required mRNA output and these conformations will be the most frequently observed. The ensemble may exhibit complex patterns of positive and negative effective HOCs among the input TFs which will characterise the required output. In the light of our flexibility theorem, the occurrence of such HOCs, which appear to be necessary to account for data on gene regulation (*Park et al., 2019a*), may be seen as evidence for conformational complexity. When the cellular context changes, different conformations, adapted to produce the output required in the new context, may be present most often—although careful inspection may show them to have been more fleetingly present previously, as would be expected under conformational selection. More broadly, the complexity of the regulatory conformational ensemble and its dynamics reflects the complexity of functional integration which the gene has to undertake.

Furlong and Levine have suggested a 'hub and condensate' model for the overall gene regulatory machinery, which brings together aspects of earlier models to account for how remote enhancers communicate with target promoters (*Furlong and Levine, 2018*). The allosteric perspective taken here emphasises the significance of conformational dynamics for the functional integration undertaken by such 'hubs'.

Testing these ideas on the scale of the regulatory machinery presents a daunting challenge, but recent developments point the way towards approaching them, including advances in cryo-EM (*Lewis and Costa, 2020*), single-molecule microscopy (*Li et al., 2019*; *Bacic et al., 2020*), NMR (*Shi et al., 2020*), synthetic biology (*Park et al., 2019b*) and the measurement of higher-order quantities (*Gruber and Horovitz, 2018*). Before experiments can be formulated, an appropriate conceptual picture needs to be described and that is what we have tried to formulate here. We now know a great deal about the molecular components involved in gene regulation, but the question of how these components collectively give rise to function has been harder to grasp. The allosteric analogy to haemoglobin, upon which we have built here, suggests a potential way to fill this gap.

In extending the haemoglobin analogy, we have sidestepped the issue of energy expenditure. This is not relevant for haemoglobin, but it can hardly be avoided in considering eukaryotic gene regulation, where reorganisation of chromatin and nucleosomes requires energy-dissipating motor proteins and post-translational modifications driven by chemical potential differences are found on all components of the regulatory machinery (*Wong and Gunawardena, 2020*). What impact such energy expenditure has on ensemble functional integration is a very interesting question. In a separate study that was stimulated by the present paper, we have confirmed that, if a conformational ensemble is maintained in steady state away from thermodynamic equilibrium, then it can exhibit greater functional capabilities than at equilibrium. We hope to report on these findings

subsequently. The results presented here offer a rigorous starting point for thinking about how regulatory ensembles integrate binding information at thermodynamic equilibrium. If, indeed, regulatory energy expenditure is essential for gene expression function, as studies increasingly suggest (*Park et al., 2019a*; *Grah et al., 2020*; *Wolff et al., 2021*), new methods, both theoretical and experimental, will be required to understand its functional significance.

# Materials and methods

## The linear framework

### Background and references

The graphs described in the main text, like those in *Figure 4*, are 'equilibrium graphs', which are convenient for describing systems at thermodynamic equilibrium. Equilibrium graphs are derived from linear framework graphs. The distinction between them is that the latter specifies a dynamics, while the former specifies an equilibrium steady state. We first explain the latter and then describe the former. Throughout this section we will use 'graph' to mean 'linear framework graph' and 'equilibrium graph' to mean the kind of graph used in the main text.

The linear framework was introduced in *Gunawardena, 2012*, developed in *Mirzaev and Gunawardena, 2013*, *Mirzaev and Bortz, 2015*, applied to various biological problems in *Ahsendorf et al., 2014*, *Dasgupta et al., 2014*, *Estrada et al., 2016*, *Wong et al., 2018a*, *Wong et al., 2018b*, *Yordanov and Stelling, 2018*, *Biddle et al., 2019*, *Yordanov and Stelling, 2020* and reviewed in *Gunawardena, 2014*, *Wong and Gunawardena, 2020*. Technical details and proofs of the ideas described here can be found in *Gunawardena, 2012*, *Mirzaev and Gunawardena, 2013*, as well as in the Supplementary Information of *Estrada et al., 2016*, *Wong et al., 2018b*, *Biddle et al., 2019*.

The framework uses finite, directed graphs with labelled edges and no self-loops to analyse biochemical systems under timescale separation. In a typical timescale separation, the vertices represent 'fast' components or states, which are assumed to reach steady state; the edges represent reactions or transitions; and the edge labels represent rates with dimensions of $(\text{time})^{-1}$. The labels may include contributions from 'slow' components, which are not represented by vertices but which interact with them, such as binding ligands in the case of allostery.

## Linear framework graphs and dynamics

Graphs will always be connected, so that they cannot be separated into sub-graphs between which there are no edges. The set of vertices of a graph $G$ will be denoted by $\nu(G)$. For a general graph, the vertices will be indexed by numbers $1, \cdots, N \in \nu(G)$ and vertex 1 will be taken to be the reference vertex. Particular kinds of graphs, such as the allostery graphs discussed in the paper, may use a different indexing. An edge from vertex $i$ to vertex $j$ will be denoted $i \rightarrow j$ and the label on that edge by $\ell(i \rightarrow j)$. A subscript, as in $i \rightarrow_G j$, may be used to specify which graph is under discussion. When discussing graphs, we used the word 'structure' to refer to properties that depend on vertices and edges only, ignoring the labels.

A graph gives rise to a dynamical system by assuming that each edge is a chemical reaction under mass-action kinetics with the label as the rate constant. Since each edge has only a single source vertex, the corresponding dynamics is linear and can be represented by a linear differential equation in matrix form:

$$\frac{du}{dt} = \mathcal{L}(G)u. \tag{28}$$

Here, $G$ is the graph, $u$ is a vector of component concentrations and $\mathcal{L}(G)$ is the Laplacian matrix of $G$. Since material is only moved between vertices, there is a conservation law, $\sum_i u_i(t) = u_{tot}$. By setting $u_{tot} = 1$, $u$ can be treated as a vector of probabilities. In such a stochastic setting, *Equation 28* is the master equation (Kolmogorov forward equation) of the underlying Markov process. This is a general representation: given any well-behaved Markov process on a finite state space, there is a graph, whose vertices are the states, for which *Equation 28* is the master equation.

The linear dynamics in *Equation 28* gives the linear framework its name and is common to all applications. The treatment of the external components, which appear in the edge labels and which

introduce nonlinearities, depends on the application. For the case of allostery treated here, we make the same assumptions as in thermodynamics for the grand canonical ensemble, with each ligand being present in a reservoir from which binding and unbinding to graph vertices does not change its free concentration. In this case, the edge labels are effectively constant. The same assumptions are implicitly used in other studies of allostery.

## Steady states and thermodynamic equilibrium

The dynamics in *Equation 28* always tends to a steady state, at which $du/dt = 0$, and, under the fundamental timescale separation, it is assumed to have reached a steady state. If the graph is strongly connected, it has a unique steady state up to a scalar multiple, so that $\dim \ker \mathcal{L}(G) = 1$. Strong connectivity means that, given any two distinct vertices, $i$ and $j$, there is a path of directed edges from $i$ to $j$, $i = i_1 \to i_2 \to \cdots \to i_{k-1} \to i_k = j$. Under strong connectivity, a representative steady state for the dynamics, $\rho(G) \in \ker \mathcal{L}(G)$, may be calculated in terms of the edge labels by the Matrix Tree Theorem. We omit the corresponding expression as it is not needed here, but it can be found in any of the references given above. This expression holds whether or not the steady state is one of thermodynamic equilibrium. However, at thermodynamic equilibrium, the description of the steady state simplifies considerably because detailed balance holds. This means that the graph is reversible, so that, if $i \to j$, then also $j \to i$, and each pair of such edges is independently in flux balance, so that

$$\rho_i(G)\ell(i \to j) = \rho_j(G)\ell(j \to i). \tag{29}$$

This 'microscopic reversibility' is a fundamental property of thermodynamic equilibrium. Note that a reversible, connected graph is necessarily strongly connected.

Take any path of reversible edges from the reference vertex 1 to some vertex $i$, $1 = i_1 \rightleftharpoons i_2 \rightleftharpoons \cdots \rightleftharpoons i_{k-1} \to i_k = i$, and let $\mu_i(G)$ be the product of the label ratios along the path:

$$\mu_i(G) = \left(\frac{\ell(i_1 \to i_2)}{\ell(i_2 \to i_1)}\right) \times \cdots \times \left(\frac{\ell(i_{k-1} \to i_k)}{\ell(i_k \to i_{k-1})}\right). \tag{30}$$

It is straightforward to see from *Equation 29* that $\mu_i(G)$ does not depend on the chosen path and that $\rho_i(G) = \mu_i(G)\rho_1(G)$. The vector $\mu(G)$ is therefore a scalar multiple of $\rho(G)$ and so also a steady state for the dynamics. The detailed balance formula in *Equation 29* also holds for μ in place of ρ. At thermodynamic equilibrium, the only parameters needed to describe steady states are label ratios.

## Equilibrium graphs and independent parameters

This observation about label ratios leads to the concept of an equilibrium graph. Suppose that $G$ is a linear framework graph which can reach thermodynamic equilibrium and is therefore reversible (above). $G$ gives rise to an equilibrium graph, $\mathcal{E}(G)$, as follows. The vertices and edges of $\mathcal{E}(G)$ are the same as those of $G$, but the edge labels in $\mathcal{E}(G)$, which we will refer to as 'equilibrium edge labels' and denote $\ell_{\mathrm{eq}}(i \to j)$, are the label ratios in $G$. In other words,

$$\ell_{\mathrm{eq}}(i \to j) = \frac{\ell(i \to_G j)}{\ell(j \to_G i)}. \tag{31}$$

*Scheme 1* illustrates the relationship between the linear framework graph and the corresponding equilibrium graph. Note that the equilibrium edge labels of $\mathcal{E}(G)$ are non-dimensional and that $\ell_{\mathrm{eq}}(j \to i) = \ell_{\mathrm{eq}}(i \to j)^{-1}$. The equilibrium edge labels are the essential parameters for describing a state of thermodynamic equilibrium.

These parameters are not independent because *Equation 29* implies algebraic relationships among them. Indeed, *Equation 29* is equivalent to the following 'cycle condition', which we formulate for $\mathcal{E}(G)$: given any cycle of edges, $i_1 \to i_2 \to \cdots \to i_{k-1} \to i_1$, the product of the equilibrium edge labels along the cycle is always 1:

$$\ell_{\mathrm{eq}}(i_1 \to i_2) \times \cdots \times \ell_{\mathrm{eq}}(i_{k-1} \to i_1) = 1. \tag{32}$$

This cycle condition is equivalent to the detailed balance condition in *Equation 29* and either condition is equivalent to $G$ being at thermodynamic equilibrium.

There is a systematic procedure for choosing a set of equilibrium edge label parameters which are both independent, so that there are no algebraic relationships among them, and also complete, so that all other equilibrium edge labels can be algebraically calculated from them. Recall that a spanning tree of $G$ is a connected subgraph, $T$, which contains each vertex of $G$ (spanning) and which has no cycles when edge directions are ignored (tree). Any strongly connected graph has a spanning tree and the number of edges in such a tree is one less than the number of vertices in the graph. Since $G$ and $\mathcal{E}(G)$ have the same vertices and edges, they have identical spanning trees. The equilibrium edge labels $\ell_{\mathrm{eq}}(i \to_T j)$, taken over all edges $i \to j$ of $T$, form a complete and independent set of parameters at thermodynamic equilibrium. In particular, if $G$ has $N$ vertices, there are $N-1$ independent parameters at thermodynamic equilibrium.

In the main text, we defined an equilibrium allostery graph, $A$ (*Figure 4*), without specifying a corresponding linear framework graph, $G$, for which $\mathcal{E}(G) = A$. Because label ratios are used in an equilibrium graph, there is no unique linear framework graph corresponding to it. However, some choice of transition rates, $\ell(i \to_G j)$ and $\ell(j \to_G i)$, can always be made such that their ratio is $\ell_{\mathrm{eq}}(i \to_{\mathcal{E}(G)} j)$. Hence, some linear framework graph $G$ can always be defined such that $\mathcal{E}(G) = A$. In some of the constructions below, we will work with the linear framework graph, $G$, rather than with the equilibrium graph $A$ and will then show that the construction does not depend on the choice of $G$.

## Steady-state probabilities and equilibrium statistical mechanics

The steady-state probability of vertex $i$, $\mathrm{Pr}_i(G)$, can be calculated from the steady state of the dynamics by normalising, so that

$$\mathrm{Pr}_i(G) = \frac{\rho_i(G)}{\rho_1(G) + \cdots + \rho_N(G)} \qquad \text{or} \qquad \mathrm{Pr}_i(G) = \frac{\mu_i(G)}{\mu_1(G) + \cdots + \mu_N(G)}, \tag{33}$$

where the first formula holds for any strongly connected graph and the second formula also holds if the graph is at thermodynamic equilibrium. In the latter case, *Equation 29* holds and $\mu(G)$ can be defined by *Equation 30*. The second formula in *Equation 33* corresponds to *Equation 3*. If the graph is at thermodynamic equilibrium, the equilibrium edge labels may be interpreted thermodynamically, as illustrated in *Figure 3* and discussed in the main text (*Equation 1*):

$$\ell_{\mathrm{eq}}(i \to j) = \exp\left(\frac{\Delta\Phi}{k_B T}\right). \tag{34}$$

If *Equation 34* is used to expand the second formula in *Equation 33*, it gives the specification of equilibrium statistical mechanics for the grand canonical ensemble, with the denominator being the partition function.

It will be helpful to let $\Pi(G)$ and $\Psi(G)$ denote the corresponding denominators in *Equation 33*, so that $\Pi(G) = \rho_1(G) + \cdots + \rho_N(G)$ for any strongly connected graph and $\Psi(G) = \mu_1(G) + \cdots + \mu_N(G)$ for a graph which is at thermodynamic equilibrium. We will refer to $\Pi(G)$ and $\Psi(G)$ as partition functions. It follows from *Equation 33* that

$$\mathrm{Pr}_i(G)\Pi(G) = \rho_i(G) \qquad \text{or} \qquad \mathrm{Pr}_i(G)\Psi(G) = \mu_i(G), \tag{35}$$

depending on the context.

## The allostery graph
### Structure and labels

An allostery graph, $A$, is an equilibrium graph which describes the interplay between conformational change and ligand binding, as illustrated in *Figure 4*. Its vertices are indexed by $(c_k, S)$, where $c_k$ specifies a conformation with $1 \leq k \leq N$ and $S \subseteq \{1, \cdots, n\}$ specifies a subset of sites bound by a ligand whose concentration is $x$. There is no difficulty in allowing multiple ligands and overlapping binding sites, but to keep the formalism simple, we describe here the case of a single ligand and distinct binding sites.

Recall from the main text that $A$ has vertical subgraphs, $A^{c_k}$, consisting of vertices $(c_k, R)$ for all binding subsets, $R$, together with all edges between them, with the vertices indexed by binding subsets, $R$, and with $R = \emptyset$ being the reference vertex. $A$ has horizontal subgraphs, $A_S$, consisting of

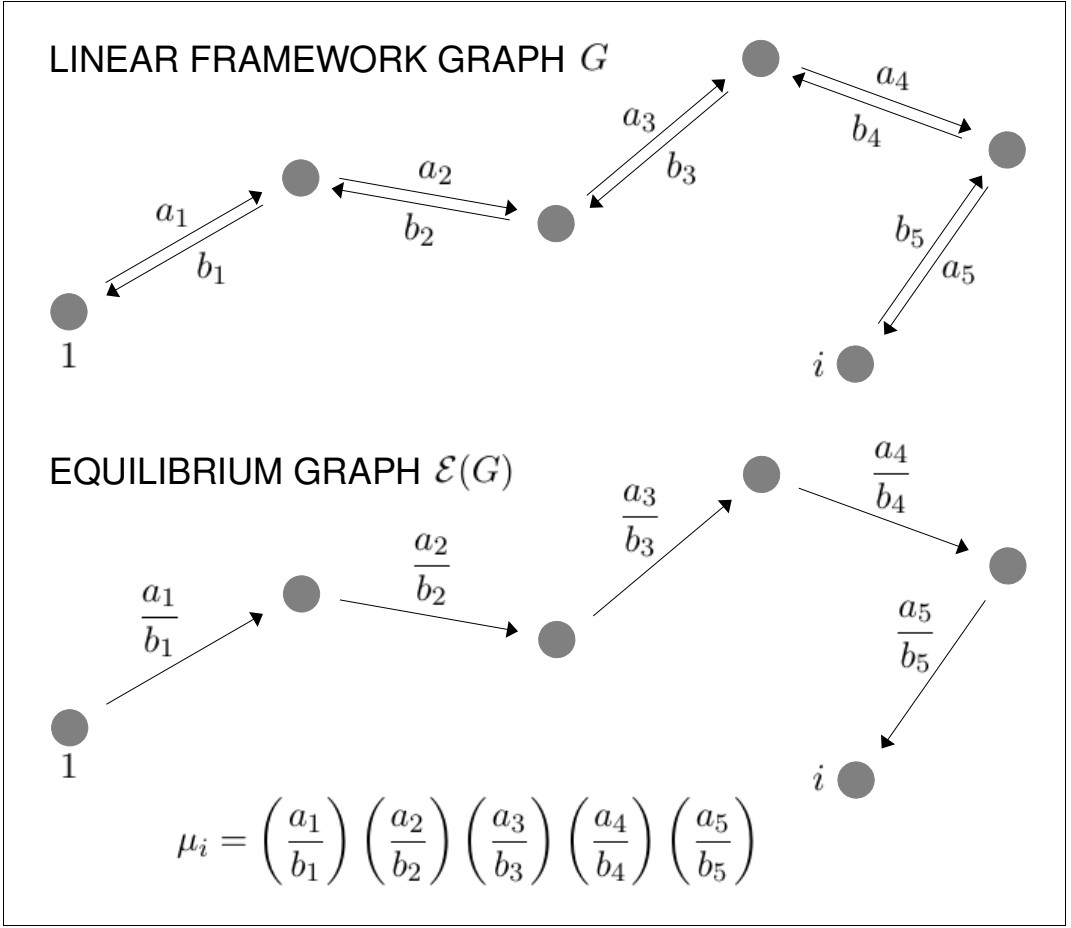

**Scheme 1.** Graphs and equilibrium calculations. At top, a path of reversible edges in a linear framework graph, $G$, from the reference vertex, 1, to a vertex $i$, with the edge labels shown. Below is the same path in the corresponding equilibrium graph, $\mathcal{E}(G)$, showing the equilibrium labels, as given by **Equation 31**. The formula for the quantity $\mu_i$, as specified in **Equation 30**, is shown at bottom. For $\mu$ to be well defined, $G$ must satisfy detailed balance (**Equation 29**) or, equivalently, the cycle condition (**Equation 32**) must hold in $\mathcal{E}(G)$. Equilibrium probabilities are calculated from $\mu$ using **Equation 33**.

vertices $(c_i, S)$ for all conformations $c_i$, together with all edges between them, with the vertices labelled by conformations $c_i$, and with $c_1$ being the reference vertex. The product structure of $A$ is revealed by all vertical subgraphs having the same structure as each other and all horizontal subgraphs having the same structure as each other (**Figure 4**).

As for the labels, the vertical binding edges have equilibrium labels,

$$\ell_{\text{eq}}((c_k, S) \rightarrow_A (c_k, S \cup \{i\})) = x K_{c_k, i, S} \ (i \notin S), \tag{36}$$

where $x$ is the concentration of the ligand and $K_{c_k, i, S}$ is the association constant for binding to site $i$ when the ligand is already bound at the sites in $S$. The horizontal edges, which represent transitions between conformations, have equilibrium labels, $\ell_{\text{eq}}((c_k, S) \rightarrow_A (c_l, S))$, which are not individually annotated. However, it is only necessary to specify these equilibrium labels for a single horizontal subgraph, of which the subgraph of empty conformations, $A_{\emptyset}$, is particularly convenient. To see this, let us calculate the quantity $\mu_{(c_k, S)}(A)$ using **Equation 30**. Taking the reference vertex in $A$ to be $(c_1, \emptyset)$, we can always find a path to any given vertex $(c_k, S)$ of $A$ by first moving horizontally within $A_{\emptyset}$ from $(c_1, \emptyset)$ to $(c_k, \emptyset)$ and then moving vertically within $A^{c_k}$ from $(c_k, \emptyset)$ to $(c_k, S)$. According to **Equation 30**, the steady state is given by the product of the equilibrium labels along this path, so that

$$\mu_{(c_k, S)}(A) = \mu_{c_k}(A_{\emptyset}) \mu_S(A^{c_k}). \tag{37}$$

Now consider any horizontal edge in $A$, $(c_k, S) \to (c_l, S)$. Since $A$ is at thermodynamic equilibrium, it follows from *Equation 29*, using μ in place of ρ, and *Equation 37*, that

$$\ell_{\text{eq}}((c_k, S) \to_A (c_l, S)) = \frac{\mu_{(c_l, S)}(A)}{\mu_{(c_k, S)}(A)} = \left(\frac{\mu_{c_l}(A_\emptyset)}{\mu_{c_k}(A_\emptyset)}\right)\left(\frac{\mu_S(A^{c_l})}{\mu_S(A^{c_k})}\right).$$

Applying *Equation 29* to $A_\emptyset$, with μ in place of ρ, we see that

$$\ell_{\text{eq}}((c_k, \emptyset) \to_{A_\emptyset} (c_l, \emptyset)) = \frac{\mu_{c_l}(A_\emptyset)}{\mu_{c_k}(A_\emptyset)}.$$

Hence, it follows that

$$\ell_{\text{eq}}((c_k, S) \to_A (c_l, S)) = \ell_{\text{eq}}((c_k, \emptyset) \to_{A_\emptyset} (c_l, \emptyset))\left(\frac{\mu_S(A^{c_l})}{\mu_S(A^{c_k})}\right). \tag{38}$$

Accordingly, all the labels in $A$ are determined by the vertical labels in *Equation 36*, from which $\mu_S(A^{c_k})$ and $\mu_S(A^{c_l})$ are determined, and the horizontal labels in the subgraph of empty conformations, $A_\emptyset$. As can be seen from *Scheme 2*, *Equation 38* amounts to exploiting the equilibrium cycle condition in *Equation 32*.

### Independent parameters

We can choose any spanning tree in the horizontal subgraph of empty conformations, $A_\emptyset$. As explained above, the equilibrium labels on the edges of this tree define a complete set of $N - 1$ independent parameters for $A_\emptyset$. As for the vertical subgraphs, $A^{c_k}$, which all have the same structure, consider the subgraph of $A^{c_k}$ consisting of all edges, together with the corresponding source and target vertices, of the form, $(c_k, S) \to (c_k, S \cup \{i\})$, where $\emptyset \subseteq S \subset \{1, \cdots, n\}$ and $i$ is less than all the sites in $S$ ($i<S$). It is not difficult to see that this subgraph is a spanning tree of $A^{c_k}$ (*Estrada et al., 2016*, SI, §3.2). Accordingly, the association constants, $K_{c_k, i, S}$ from *Equation 36*, with $i<S$, form a complete set of independent parameters for $A^{c_k}$. Because of the product structure of $A$, adjoining the spanning trees in $A^{c_k}$, for each conformation $c_k$ with $1 \leq k \leq N$, to the spanning tree in $A_\emptyset$, yields a spanning tree in $A$. Hence, the independent parameters for $A^{c_k}$ together with the $N - 1$ independent parameters for $A_\emptyset$ are also collectively independent as parameters for $A$. It follows from the description of labels above that these parameters are also complete for $A$, so that any equilibrium label in $A$ can be expressed in terms of them.

### A general method of coarse graining

#### Coarse graining a linear framework graph and *Equation 17*

We will describe the coarse-graining procedure for an arbitrary reversible linear framework graph, $G$, and then explain how this can be adapted to an equilibrium graph, as described for the allostery graph $A$ in the main text.

We will say that a graph $G$ is *in-uniform* if, given any vertex $j \in \nu(G)$, then for all edges $i \to j$, $\ell(i \to j)$ does not depend on the source vertex $i$.

#### Lemma 1

Suppose that $G$ is reversible and in-uniform. Then, $G$ is at thermodynamic equilibrium and the vector θ given by $\theta_j = \ell(i \to j)$, which is well-defined by hypothesis, is a basis element in $\ker \mathcal{L}(G)$ and a steady state for the dynamics.

**Proof**: If $i_1 \rightleftharpoons i_2 \rightleftharpoons \cdots \rightleftharpoons i_{k-1} \rightleftharpoons i_k$ is any path of reversible edges in $G$, then the product of the label ratios along the path satisfies

$$\left(\frac{\ell(i_1 \to i_2)}{\ell(i_2 \to i_1)}\right)\left(\frac{\ell(i_2 \to i_3)}{\ell(i_3 \to i_2)}\right) \cdots \left(\frac{\ell(i_{k-2} \to i_{k-1})}{\ell(i_{k-1} \to i_{k-2})}\right)\left(\frac{\ell(i_{k-1} \to i_k)}{\ell(i_k \to i_{k-1})}\right) = \frac{\ell(i_{k-1} \to i_k)}{\ell(i_2 \to i_1)}, \tag{39}$$

because the intermediate terms cancel out by the in-uniform hypothesis. If the path is a cycle, so that $i_k = i_1$, then, again because of the in-uniform hypothesis, the right-hand side of *Equation 39* is 1. Hence, $G$ satisfies the cycle condition in *Equation 32* and is therefore at thermodynamic equilibrium.

For the last statement, assume that $i_1$ is the reference vertex 1 and that $i_k = j$, for any vertex $j$. Using **Equation 30**, we see that $\mu_j(G) = \theta_j/\theta_1$. Since $\theta_1$ is a scalar multiple, the last statement follows.

∎

Now let $G$ be an arbitrary reversible graph, which need not satisfy detailed balance. Let $G_1, \cdots, G_m$ be any partition of the vertices of $G$, so that $G_i \subseteq \nu(G)$, $G_1 \cup \cdots \cup G_m = \nu(G)$ and $G_i \cap G_j = \emptyset$ when $i \neq j$. Let $\mathcal{C}(G)$ be the labelled directed graph with $\nu(\mathcal{C}(G)) = \{1, \cdots, m\}$ and let $u \rightarrow_{\mathcal{C}(G)} v$ if, and only if, there exists $i \in G_u$ and $j \in G_v$ such that $i \rightarrow_G j$. Finally, let the edge labels of $\mathcal{C}(G)$ be given by

$$\ell(u \rightarrow_{\mathcal{C}(G)} v) = Q\left(\sum_{i \in G_v} \rho_i(G)\right). \tag{40}$$

The quantity $Q$ in **Equation 40** is chosen arbitrarily so that the dimension of $\ell(u \rightarrow v)$ is (time)$^{-1}$, as required for an edge label. This is necessary because, by the Matrix Tree Theorem, the dimension of $\rho_i(G)$ is (time)$^{1-N}$, where $N$ is the number of vertices in $G$. However, $Q$ plays no role in the analysis which follows because the coarse graining applies only to the steady state of $\mathcal{C}(G)$, not its transient dynamics, and, as we will see, $\mathcal{C}(G)$ is always at thermodynamic equilibrium, so that $Q$ disappears when equilibrium edge labels are considered.

Note that $\mathcal{C}(G)$ inherits reversibility from $G$ and that $\mathcal{C}(G)$ is in-uniform. Hence, by Lemma 1, $\mathcal{C}(G)$ is at thermodynamic equilibrium and

$$\lambda\mu_v(\mathcal{C}(G)) = Q\left(\sum_{i \in G_v} \rho_i(G)\right), \tag{41}$$

where $\lambda$ is a scalar that does not depend on $v \in \nu(\mathcal{C}(G))$. Since $G_1, \cdots, G_m$ is a partition of the vertices of $G$, it follows from **Equation 41** that

$$\lambda\Psi(\mathcal{C}(G)) = \lambda\left(\sum_{v \in \nu(\mathcal{C}(G))} \mu_v(\mathcal{C}(G))\right) = Q\left(\sum_{i \in \nu(G)} \rho_i(G)\right) = Q\Pi(G).$$

**Equations 35 and 41** then show that both $\lambda$ and $Q$ cancel in the ratio for the steady-state probabilities, so that

$$\Pr{}_v(\mathcal{C}(G)) = \sum_{i \in G_v} \Pr{}_i(G). \tag{42}$$

**Equation 42** is the coarse-graining equation, as given in **Equation 17**.

## Coarse graining an equilibrium graph

The coarse-graining procedure described above can be applied to any reversible graph, which need not be at thermodynamic equilibrium. However, the coarse graining described in the paper was for an equilibrium graph. It is not difficult to see that the construction above can be undertaken consistently for any equilibrium graph. It is helpful to first establish a more general observation. The choice of edge labels for $\mathcal{C}(G)$, as given in **Equation 40**, is not the only one for which **Equation 42** holds, as the appearance of the factor $Q$ indicates. However, the label ratios in $\mathcal{C}(G)$ are uniquely determined by the labels of $G$.

Suppose that $G$ is a reversible graph with a vertex partition $G_1, \cdots, G_m$, as above. $G$ need not be at thermodynamic equilibrium. Suppose that $C$ is a graph which is isomorphic to $\mathcal{C}(G)$ as a directed graph ('structurally isomorphic'), in the sense that it has identical vertices and edges but may have different edge labels. (Technically speaking, an 'isomorphism' allows for the vertices of $C$ to have an alternative indexing to those of $\mathcal{C}(G)$ as long as the two indexings can be inter-converted so as to preserve the edges. For simplicity of exposition, we assume that the indexing is, in fact, identical. No loss of generality arises from doing this.)

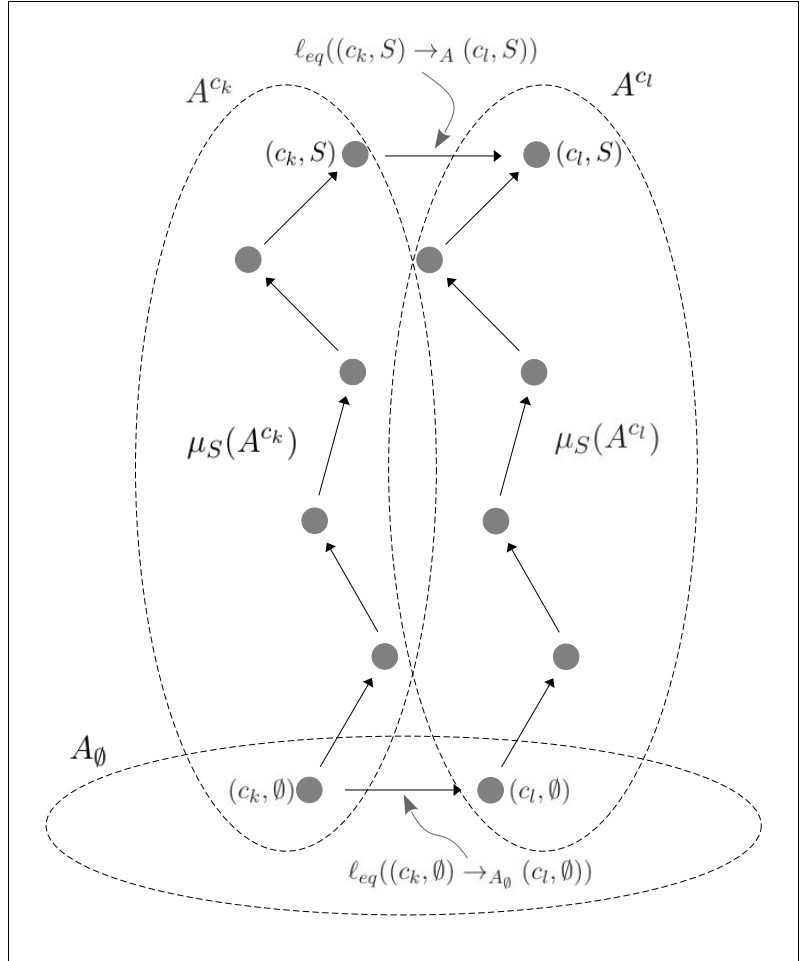

**Scheme 2.** Illustration of *Equation 38*. A hypothetical allostery graph shows how the label for the edge at the top, which links the vertical subgraphs at conformations $c_k$ and $c_l$ at the binding subset $S$, can be calculated from the quantities $\mu_S(A^{c_k})$ and $\mu_S(A^{c_l})$ and the edge label at the bottom. The $\mu_S$ quantities come from paths to the vertices in question from the respective reference vertices in the vertical subgraphs, as specified in *Equation 30* and *Scheme 1*. The edge label at the bottom comes from the horizontal subgraph of empty conformations, $A_\emptyset$. The vertical subgraphs $A^{c_k}$ and $A^{c_l}$ have the same structure and the paths are shown as the same in each subgraph, but they could be arbitrary paths because of the cycle condition at thermodynamic equilibrium (*Equation 32*). Once appropriate directions are taken, the two paths and the edges at the top and bottom constitute a large cycle in the allostery graph and *Equation 38* is simply a rewriting of *Equation 32* applied to this cycle.

## Lemma 2

Suppose that $C$ is at thermodynamic equilibrium and the coarse-graining equation (*Equation 42*) holds for $C$, so that $\Pr_u(C) = \sum_{i \in G_u} \Pr_i(G)$. If $u \rightleftharpoons_C v$ is any reversible edge, then its equilibrium label depends only on $G$,

$$\ell_{\text{eq}}(u \rightarrow_C v) = \frac{\sum_{i \in G_v} \rho_i(G)}{\sum_{i \in G_u} \rho_i(G)},$$

and $C$ and $\mathcal{C}(G)$ are isomorphic as equilibrium graphs, so that identical edges have identical equilibrium labels.

   **Proof**: It follows from *Equation 35* that $\Pr_i(G) = \rho_i(G)/\Pi(G)$ and, since $C$ is at thermodynamic equilibrium, $\Pr_u(C) = \mu_u(C)/\Psi(C)$. Using the coarse-graining equation for $\Pr_u(C)$, we see that

$$\mu_u(C) = \left( \sum_{i \in G_u} \rho_i(G) \right) \left( \frac{\Psi(C)}{\Pi(G)} \right). \tag{43}$$

Since $C$ is at thermodynamic equilibrium, *Equation 29*, with $\mu$ in place of $\rho$, implies that

$$\ell_{eq}(u \to_C v) = \frac{\mu_v(C)}{\mu_u(C)}.$$

Substituting with *Equation 43*, the partition functions cancel out to give the formula above. Since $\mathcal{C}(G)$ satisfies the same assumptions as $C$, it has the same equilibrium labels. Hence, $C$ and $\mathcal{C}(G)$ must be isomorphic as equilibrium graphs.

∎

## Corollary 1

Suppose that $A$ is an equilibrium graph and that $G$ is any graph for which $\mathcal{E}(G) = A$, as described above. If any coarse graining of $G$ is undertaken to yield the coarse-grained graph $\mathcal{C}(G)$, which must be at thermodynamic equilibrium, then

$$\ell_{eq}(u \to_{\mathcal{C}(G)} v) = \frac{\sum_{i \in A_v} \mu_i(A)}{\sum_{i \in A_u} \mu_i(A)}$$

and $\mathcal{E}(\mathcal{C}(G))$ depends only on $A$ and not on the choice of $G$.

**Proof**: $A$ acquires from $G$ the same coarse graining, with the partition $A_1, \cdots, A_m$ of $\nu(A)$, where $A_i = G_i \subseteq \{1, \cdots m\}$. By hypothesis, $G$ is at thermodynamic equilibrium, so that $\rho_i(G) = \lambda \mu_i(G)$ for some scalar multiple $\lambda$. Also, since $\mathcal{E}(G) = A$, $\mu_i(G) = \mu_i(A)$. Substituting in the formula in Lemma 2 yields the formula above. The equilibrium labels of $\mathcal{C}(G)$ therefore depend only on the equilibrium labels of $A$, as required.

∎

It follows from Corollary 1 that coarse graining can be carried out on an equilibrium graph, $A$, by choosing any graph $G$ for which $\mathcal{E}(G) = A$ and carrying out the coarse-graining procedure described above on $G$. This justifies the coarse-graining construction described in the main text.

## Coarse graining the allostery graph
### Proof of *Equation 18*

As described in the main text and *Figure 4*, the coarse-grained allostery graph, $A^\phi = \mathcal{C}(A)$, is defined using the partition of $A$ by its horizontal subgraphs, $A_S$, where $S$ runs through all binding subsets, $S \subseteq \{1, \cdots, n\}$. $A^\phi$ has the same structure of vertices and edges as any of the binding subgraphs, $A^{c_k}$, and is indexed in the same way by the binding subsets, $S$. *Scheme 3* shows an example, which illustrates the calculations undertaken in this section.

Consider the reversible edge in $A^\phi$, $S \rightleftharpoons S \cup \{i\}$, where $i \notin S$. This reversible edge effectively arises from the binding and unbinding of ligand at site $i$. According to *Equation 36*, its effective association constant, $K_{i,S}^\phi$, should satisfy

$$xK_{i,S}^\phi = \ell_{eq}(S \to_{A^\phi} S \cup \{i\}). \tag{44}$$

Since $A$ is at thermodynamic equilibrium, we can make use of the formula in Corollary 1 to rewrite this as

$$K_{i,S}^\phi = x^{-1} \left( \frac{\sum_{1 \leq k \leq N} \mu_{(c_k, S \cup \{i\})}(A)}{\sum_{1 \leq k \leq N} \mu_{(c_k, S)}(A)} \right).$$

*Equations 30 and 36* tell us that $\mu_{(c_k, S \cup \{i\})}(A) = xK_{c_k, i, S}\mu_{(c_k, S)}(A)$, so that, after rearranging,

$$K_{i,S}^{\phi} = \sum_{1 \leq k \leq N} K_{c_k,i,S} \left( \frac{\mu_{(c_k,S)}(A)}{\sum_{1 \leq k \leq N} \mu_{(c_k,S)}(A)} \right). \tag{45}$$

We can now appeal to *Equations 35 and 37* to rewrite the term in brackets on the right as

$$\frac{\mu_S(A^{c_k})\mu_{c_k}(A_\emptyset)}{\sum_{1 \leq k \leq N} \mu_S(A^{c_k})\mu_{c_k}(A_\emptyset)} = \frac{\mu_S(A^{c_k})\mathrm{Pr}_{c_k}(A_\emptyset)}{\sum_{1 \leq k \leq N} \mu_S(A^{c_k})\mathrm{Pr}_{c_k}(A_\emptyset)}. \tag{46}$$

At this point, it will be helpful to introduce the following notation. If $G$ is any equilibrium graph and $u : \nu(G) \to \mathbf{R}$ is any real-valued function defined on the vertices of $G$, let $\langle u \rangle$ denote the average of $u$ over the steady-state probability distribution of $G$,

$$\langle u \rangle = \sum_{i \in \nu(G)} u_i \mathrm{Pr}_i(G). \tag{47}$$

With this notation in hand, we can rewrite the denominator in *Equation 46* as $\langle \mu_S(A^{c_k}) \rangle$, where, from now on, averages will be taken over the steady-state probability distribution of the horizontal subgraph of empty conformations, $A_\emptyset$ (*Scheme 3*, bottom). Inserting this expression back into *Equation 45* and rearranging, we obtain a formula for the effective association constant as a ratio of averages,

$$K_{i,S}^{\phi} = \frac{\langle K_{c_k,i,S} \cdot \mu_S(A^{c_k}) \rangle}{\langle \mu_S(A^{c_k}) \rangle}, \tag{48}$$

which gives the first formula in *Equation 18*. The 'dot' in *Equation 48* signifies a product to make the formula easier to read. *Scheme 3* demonstrates this calculation. Recall from the main text that HOCs are defined by normalising to the empty binding subset, so that $\omega_{i,S}^{\phi} = K_{i,S}^{\phi}/K_{i,\emptyset}^{\phi}$. Furthermore, since the reference vertex of the vertical subgraphs, $A^{c_k}$, is taken to be the empty binding subset, $\mu_\emptyset(A^{c_k}) = 1$. It follows that the effective HOCs are given by

$$\omega_{i,S}^{\phi} = \frac{\langle K_{c_k,i,S} \cdot \mu_S(A^{c_k}) \rangle}{\langle K_{c_k,i,\emptyset} \rangle \cdot \langle \mu_S(A^{c_k}) \rangle}, \tag{49}$$

which gives the second formula in *Equation 18*.

## Elementary properties of effective HOCs

The main text describes three elementary properties of effective HOCs which follow from *Equation 49*. The only quantity in *Equation 49* which involves the ligand concentration, $x$, is $\mu_S(A^{c_k})$. It follows from *Equation 30* that this quantity is a monomial in $x$ of the form $ax^p$, where $a$ does not involve $x$ and $p = \#(S)$. In particular, $x^p$ does not depend on the conformation $c_k$. It follows that $x^p$ can be extracted from the averages in *Equation 49* and cancelled between the numerator and denominator. Hence, $\omega_{i,S}^{\phi}$ is independent of $x$. If $S = \emptyset$, then $\mu_S(A^{c_k}) = 1$ for all $1 \leq k \leq N$ and it follows from *Equation 49* that $\omega_{i,\emptyset}^{\phi} = 1$. Finally, if there is only one conformation $c_1$, the averages in *Equation 49* collapse and $\mu_S(A^{c_1})$ cancels above and below, so that $\omega_{i,S}^{\phi} = \omega_{c_1,i,S}$, as required.

## Generalised MWC formula

The original MWC formula calculates the binding curve, or fractional saturation, of the two-conformation model as a function of ligand concentration $x$ (*Monod et al., 1965*). Here, we do the same for an arbitrary allostery graph, $A$. Let $s = \#(S)$. The fractional saturation of $A$ is given by the average binding,

$$\sum_{1 \leq k \leq N} \sum_{S \subseteq \{1, \cdots, n\}} s \, \mathrm{Pr}_{(c_k,S)}(A),$$

normalised to the number of binding sites, $n$. By the coarse-graining formula in *Equation 42*, we can rewrite the fractional saturation as

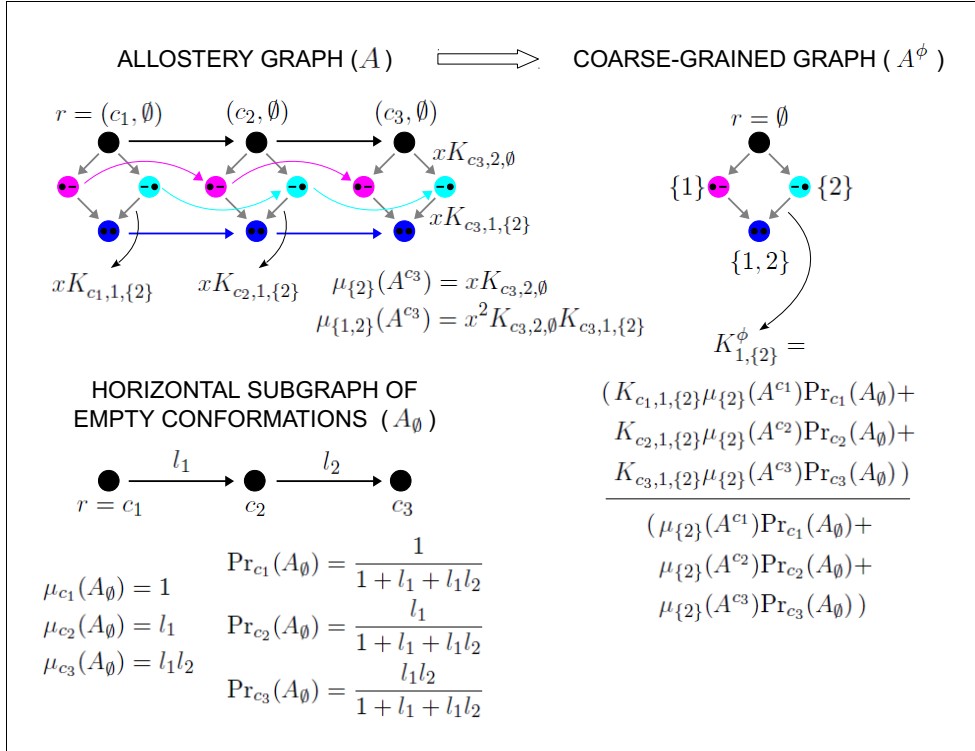

**Scheme 3.** Coarse graining and effective association constants. At top left is an example allostery graph, with binding of a single ligand to $n = 2$ sites for $N = 3$ conformations. Vertices indicate a bound site with a solid black dot and an unbound site with a black dash and binding subsets are colour coded: both sites unbound, black; only site 1 bound, magenta; only site 2 bound, cyan; both sites bound, blue. Some vertices are annotated and some edge labels are shown, with $x$ denoting ligand concentration. Note that the allostery graph has been oriented with its reference vertex at the top, in contrast to the graphs in the main text figures, in order to accommodate the formulas. Example calculations of $\mu_S$ based on *Equation 30* are shown for the vertical subgraph $A^{c_3}$. At bottom is the horizontal subgraph $A_\emptyset$ along with the calculation of its steady-state probability distribution in terms of the equilibrium labels, $l_1, l_2$. and the quantities $\mu_{c_k}$. At top right is the coarse-grained allostery graph, $A^\phi$, with vertices colour coded as for the binding subsets of the allostery graph. *Equation 48* for the effective association constants is illustrated below $A^\phi$.

$$\frac{1}{n}\left(\sum_{S\subseteq\{1,\cdots,n\}} s\,\mathrm{Pr}_S(A^\phi)\right). \tag{50}$$

The probability, $\mathrm{Pr}_S(A^\phi)$, can be calculated using *Equation 33*, which requires the quantities $\mu_S(A^\phi)$. These can in turn be calculated by the path formula in *Equation 30*. We can choose the path in $A^\phi$ to use the independent parameters introduced above. Let $S = \{i_1, \cdots, i_s\}$, where $i_1 < \cdots < i_s$. Making use of *Equation 44*, we see that

$$\mu_S(A^\phi) = K^\phi_{i_1,\{i_2,\cdots,i_s\}} K^\phi_{i_2,\{i_3,\cdots,i_s\}} \cdots K^\phi_{i_{s-1},\{i_s\}} K^\phi_{i_s,\emptyset} x^s. \tag{51}$$

*Equation 51* can be rewritten in terms of the non-dimensional effective HOCs, but it is simpler for our purposes to use instead the effective association constants, $K^\phi_{i,S}$. The dependence on $x$ in *Equation 51* shows that average binding is given by the logarithmic derivative of the partition function, $\Psi(A^\phi)$, so the fractional saturation can be written as

$$\frac{1}{n}\left(\sum_{S\subseteq\{1,\cdots,n\}} s\,\mathrm{Pr}_S(A^\phi)\right) = \frac{1}{n}\left(\frac{x}{\Psi(A^\phi)}\right)\left(\frac{d\Psi(A^\phi)}{dx}\right). \tag{52}$$

With this in mind, **Equation 51** shows that the partition function can be written as a polynomial in $x$,

$$\Psi(A^\phi) = \sum_{S\subseteq\{1,\cdots,n\}} \mu_S(A^\phi) = \sum_{0\le s\le n}\left(\sum_{1\le i_1<\cdots<i_s\le n} K^\phi_{i_1,\{i_2,\cdots,i_s\}}K^\phi_{i_2,\{i_3,\cdots,i_s\}}\cdots K^\phi_{i_{s-1},\{i_s\}}K^\phi_{i_s,\emptyset}\right)x^s.$$

Finally, the $K^\phi_{i,S}$ can be determined as averages over the horizontal subgraph of empty conformations using **Equation 48**. In this way, the fractional saturation in **Equation 52** is ultimately determined by the independent parameters of $A$, giving rise thereby to a generalised MWC formula that is valid for any allostery graph. We explain below how the classical MWC formula is recovered using this procedure.

## Effective HOCs for MWC-like models
### Proof of **Equation 19** and related work
Let $A$ be an allostery graph with ligand binding to $n$ sites which are independent and identical in each conformation. Because of independence, $\omega_{c_k,i,S} = 1$, so that $K_{c_k,i,S} = K_{c_k,i,\emptyset}$ does not depend on $S$; because the sites are identical, $K_{c_i,i,S}$ does not depend on $i$. Hence, we may write $K_{c_k,i,S} = K_{c_k}$ and the labels on the binding edges of the vertical subgraph $A^{c_k}$ are all given by $K_{c_k}$. It follows from **Equation 30** that $\mu_S(A^{c_k}) = (K_{c_k})^s$, where $s = \#(S)$. **Equation 49** then tells us that $\omega^\phi_{i,S}$ also depends only on $s$, so that we can write it as $\omega^\phi_s$, and **Equation 49** simplifies to

$$\omega^\phi_s = \frac{\langle (K_{c_k})^{s+1}\rangle}{\langle K_{c_k}\rangle\langle (K_{c_k})^s\rangle}, \tag{53}$$

which gives **Equation 19**.

If we consider the effective association constant instead of the effective HOC, then, with the same assumptions as above, **Equation 48** tells us that

$$K^\phi_s = \frac{\langle (K_{c_k})^{s+1}\rangle}{\langle (K_{c_k})^s\rangle}.$$

Suppose that only two conformations, $R$ and $T$, are present. Let $\ell_{\mathrm{eq}}(c_R \to c_T) = L$ and write $K_{c_T}$ and $K_{c_R}$ as $K_T$ and $K_R$, respectively. Then, for any random variable on conformations, $X_{c_k}$, the average is given by $\langle X_{c_k}\rangle = (X_{c_R} + X_{c_T}L)/(1+L)$. Hence,

$$K^\phi_s = \frac{K_R^{s+1} + K_T^{s+1}L}{K_R^s + K_T^s L}, \tag{54}$$

which is the formula for the $(s + 1)$-th 'intrinsic binding constant' given by **Gruber and Horovitz, 2018**, Equation (2.10). In their analysis, the word 'intrinsic' corresponds to our 'effective'.

We can use **Equation 54** to work out what the generalised MWC formula derived above yields for the classical MWC model. Substituting **Equation 54** in **Equation 51**, the intermediate terms in the product cancel out to leave,

$$\mu_S(A^\phi) = (K_R^s + K_T^s L)x^s,$$

in which the right-hand side depends only on $s = \#(S)$. Collecting together subsets of the same size, the partition function of $A^\phi$ may be written as

$$\Psi(A^\phi) = \sum_{0\le s\le n} {}^ns(K_R^s + K_T^s L)x^s = (1 + xK_R)^n + L(1 + xK_T)^n.$$

It then follows from **Equation 52** that the fractional saturation is given by

$$\frac{1}{n}\left(\frac{x}{\Psi(A^\phi)}\right)\left(\frac{d\Psi(A^\phi)}{dx}\right) = \frac{xK_R(1+xK_R)^{n-1} + xK_TL(1+xK_T)^{n-1}}{(1+xK_R)^n + L(1+xK_T)^n}.$$

If we set $\alpha = xK_R$ and $c\alpha = xK_T$, this gives, for the fractional saturation,

$$\frac{\alpha(1+\alpha)^{n-1} + c\alpha L(1+c\alpha)^{n-1}}{(1+\alpha)^n + L(1+c\alpha)^n}, \tag{55}$$

which recovers the classical MWC formula in the notation of *Monod et al., 1965*, *Equation 2*.

## Proof of *Equation 20*

The following result is unlikely not to be known in other contexts but we have not been able to find mention of it.

## Lemma 3

Suppose that $X$ is a positive random variable, $X > 0$, over a finite probability distribution. If $s \geq 1$, the following moment inequality holds,

$$\langle X^{s+1} \rangle \langle X^{s-1} \rangle \geq \langle X^s \rangle^2,$$

with equality if, and only if, $X$ is constant over the distribution.

**Proof**: Suppose that the states of the probability space are indexed by $1 \leq i \leq m$ and that $p_i$ denotes the probability of state $i$. Then,

$$\langle X^s \rangle = \sum_i X_i^s p_i. \tag{56}$$

The quantity $\alpha_s = \langle X^{s+1} \rangle \langle X^{s-1} \rangle - \langle X^s \rangle^2$ can then be written as

$$\alpha_s = \left(\sum_i X_i^{s+1} p_i\right)\left(\sum_i X_i^{s-1} p_i\right) - \left(\sum_i X_i^s p_i\right)^2.$$

Collecting together terms in $p_i p_j$, we can rewrite this as

$$\alpha_s = \sum_{1 \leq i \leq m}\left(\sum_{i < j \leq m}(X_i^{s+1}X_j^{s-1} + X_i^{s-1}X_j^{s+1} - 2X_i^s X_j^s)p_i p_j\right). \tag{57}$$

Note that the terms corresponding to $i = j$ yield $(X_i^{s+1}X_i^{s-1} - X_i^s X_i^s)p_i^2 = 0$ and so do not contribute to *Equation 57*. Choose any pair $1 \leq i \leq m$ and $i < j \leq m$ and let $X_j = \mu X_i$. Then, the coefficient of $p_i p_j$ in *Equation 57* becomes

$$X_i^{s+1}X_i^{s-1}\mu^{s-1} + X_i^{s-1}X_i^{s+1}\mu^{s+1} - 2X_i^s X_i^s \mu^s = (X_i^s)^2 \mu^{s-1}(1 - 2\mu + \mu^2).$$

Now, $1 - 2\mu + \mu^2 = (\mu - 1)^2 \geq 0$ for $\mu \in \mathbf{R}$, with equality if, and only if, $\mu = 1$. Since $X > 0$ by hypothesis, $\mu > 0$, so the coefficient of $p_i p_j$ is positive unless $\mu = 1$. Hence, $\alpha_s > 0$ unless $X_i = X_j$ whenever $1 \leq i \leq m$ and $i < j \leq m$, which means that $X$ is constant over the distribution. Of course, if $X$ is constant, then clearly $\alpha_s = 0$ for all $s \geq 1$. The result follows.

∎

## Corollary 2

If $A$ is an MWC-like allostery graph, its effective HOCs satisfy

$$1 \leq \omega_1^\phi \leq \omega_2^\phi \leq \cdots \leq \omega_{n-1}^\phi, \tag{58}$$

with equality at any stage if, and only if, $K_{c_k}$ is constant over $A_\emptyset$.

**Proof**: It follows from *Equation 53* that we can rewrite the effective HOCs recursively as

$$\omega_s^\phi = \omega_{s-1}^\phi \frac{\langle (K_{c_k})^{s+1} \rangle \langle (K_{c_k})^{s-1} \rangle}{\langle (K_{c_k})^s \rangle^2}. \tag{59}$$

Since $\omega_0^\phi = 1$, the result follows by recursively applying Lemma 3 to $X = K_{c_k} > 0$. **Equation 58** gives **Equation 20**.

∎

## Negative effective cooperativity

We consider an allostery graph $A$ with two conformations and two sites, in which binding is independent but not identical, so that the association constants differ between sites. Let $K_{c_k,1,\emptyset} = K_{c_k,1}$ and $K_{c_k,2,\emptyset} = K_{c_k,2}$, for $k = 1, 2$. Since the sites are independent, $\omega_{c_k,1,\{2\}} = 1$, so that $K_{c_k,1,\{2\}} = K_{c_k,1}$, for $k = 1, 2$. It follows from **Equation 30**—see also Scheme 1—that

$$\mu_{\{1\}}(A^{c_k}) = xK_{c_k,1} \ \text{ and } \ \mu_{\{2\}}(A^{c_k}) = xK_{c_k,2} \ \text{ for } k = 1, 2.$$

Let $\lambda$ be the single equilibrium label in the horizontal subgraph of empty conformations,

$$\lambda = \ell_{\text{eq}}(c_1 \to_{A_\emptyset} c_2) = \ell_{\text{eq}}((c_1, \emptyset) \to_A (c_2, \emptyset)).$$

It follows from **Equations 30 and 33**—see also the similar calculation in Scheme 3—that $\text{Pr}_{c_1}(A_\emptyset) = 1/(1+\lambda)$ and $\text{Pr}_{c_2}(A_\emptyset) = \lambda/(1+\lambda)$. We know from **Equation 49** that

$$\omega_{1,\{2\}}^\phi = \frac{\langle K_{c_k,1,\{2\}} \cdot \mu_{\{2\}}(A^{c_k}) \rangle}{\langle K_{c_k,1,\emptyset} \rangle \cdot \langle \mu_{\{2\}}(A^{c_k}) \rangle},$$

and using the identifications above, we see that

$$\begin{aligned}
\langle K_{c_k,1,\{2\}} \cdot \mu_{\{2\}}(A^{c_k}) \rangle &= \frac{(K_{c_1,1}K_{c_1,2} + \lambda K_{c_2,1}K_{c_2,2})x}{1+\lambda} \\
\langle \mu_{\{2\}}(A^{c_k}) \rangle &= \frac{(K_{c_1,2} + \lambda K_{c_2,2})x}{1+\lambda} \\
\langle K_{c_k,1,\emptyset} \rangle &= \frac{K_{c_1,1} + \lambda K_{c_2,1}}{1+\lambda}.
\end{aligned}$$

Substituting and simplifying, we find that

$$\begin{aligned}
\omega_{1,\{2\}}^\phi &= \frac{(K_{c_1,2}K_{c_1,1} + \lambda K_{c_2,2}K_{c_2,1}) \cdot (1+\lambda)}{(K_{c_1,2} + \lambda K_{c_2,2}) \cdot (K_{c_1,1} + \lambda K_{c_2,1})} \\
&= \frac{K_{c_1,1}K_{c_1,2} + \lambda(K_{c_1,1}K_{c_1,2} + K_{c_2,1}K_{c_2,2}) + \lambda^2 K_{c_2,1}K_{c_2,2}}{K_{c_1,1}K_{c_1,2} + \lambda(K_{c_1,1}K_{c_2,2} + K_{c_2,1}K_{c_1,2}) + \lambda^2 K_{c_2,1}K_{c_2,2}}.
\end{aligned}$$

The first and last terms are the same in the numerator and denominator, so it follows that $\omega_{1,\{2\}}^\phi < 1$ if, and only if,

$$\frac{K_{c_1,1}K_{c_1,2} + K_{c_2,1}K_{c_2,2}}{K_{c_1,1}K_{c_2,2} + K_{c_2,1}K_{c_1,2}} < 1,$$

which is to say

$$K_{c_1,1}K_{c_1,2} + K_{c_2,1}K_{c_2,2} - (K_{c_1,1}K_{c_2,2} + K_{c_2,1}K_{c_1,2}) < 0.$$

The left-hand side factors to give

$$(K_{c_1,1} - K_{c_2,1})(K_{c_1,2} - K_{c_2,2}) < 0.$$

We see that negative cooperativity arises if, and only if, the sites have opposite patterns of association constants in the two conformations.

## Flexibility of allostery

### The integrative flexibility theorem

We provide here a complete version of the proof that was sketched in the main text, showing rigorously how the approximation is handled. Some preliminary notation is needed. Recall that if $X$ is a finite set—typically, a subset of $\{1, \cdots, n\}$—then $\#(X)$ will denote the number of elements in $X$. If $X$ and $Y$ are sets, then $X \backslash Y$ will denote the complement of $Y$ in $X$, $X \backslash Y = \{i \in X, i \notin Y\}$. To control the approximation, we will use the 'little o' notation: $\mathcal{o}_u(1)$ will stand for any quantity which depends on $u$ and for which $\mathcal{o}_u(1) \to 0$ as $u \to 0$. For instance, $Au + Bu^2$ is $\mathcal{o}_u(1)$ but $(Au + Bu^2)/u$ is $\mathcal{o}_u(1)$ if, and only if, $A = 0$. This notation allows concise expression of complicated expressions which vanish in the limit as $u \to 0$. Note that $f(u) \to A$ as $u \to 0$ if, and only if, $f(u) = A + \mathcal{o}_u(1)$, which is a useful trick for simplifying $f$.

### Theorem 1

Suppose $n \geq 1$ and choose $2^n - 1$ arbitrary positive numbers

$$\beta_i > 0 \quad (1 \leq i \leq n) \quad \text{and} \quad \alpha_{i,S} > 0 \quad (\emptyset \neq S \subseteq \{1, \cdots, n\}, i < S).$$

Given any $\varepsilon > 0$ and $\delta > 0$, there exists an allosteric conformational ensemble, which has no intrinsic HOC in any conformation, such that

$$K_{i,\emptyset}^\phi = \beta_i + \mathcal{o}_\varepsilon(1) \quad \text{and} \quad \omega_{i,S}^\phi = \alpha_{i,S} + \mathcal{o}_\varepsilon(1) + \mathcal{o}_\delta(1)$$

for all corresponding values of $i$ and $S$.

**Proof**: Recall from the main text that we use an allostery graph $A$ whose conformations are indexed by subsets $T \subseteq \{1, \cdots, n\}$ and denoted $c_T$, as illustrated in *Figure 6*. The reference vertex of $A$ is $r = (c_\emptyset, \emptyset)$. For the horizontal subgraph of empty conformations, $A_\emptyset$, let $\lambda_T = \mu_{c_T}(A_\emptyset)$. It follows from *Equation 30*, using μ in place of ρ, that the $\lambda_T$ determine the equilibrium labels of $A_\emptyset$. Keeping in mind that $\lambda_\emptyset = 1$, the $\lambda_T$ form a set of $2^n - 1$ independent parameters for $A_\emptyset$, as explained above. The steady-state probabilities are then given by $\mathrm{Pr}_{c_T}(A_\emptyset) = \lambda_T / (\sum_{\emptyset \subseteq X \subseteq \{1, \cdots, n\}} \lambda_X)$ (*Equation 35*).

Let $\kappa_1, \cdots, \kappa_n > 0$ be positive quantities whose values we will subsequently choose. We assume that all intrinsic HOCs are one and, for any binding microstate $S \subseteq \{1, \cdots, n\}$, we set

$$K_{c_T,i,S} = K_{c_T,i,\emptyset} = \begin{cases} \kappa_i & \text{if } i \in T \\ \varepsilon \kappa_i & \text{if } i \notin T \end{cases} \tag{60}$$

If $c_T$ is a conformation and $S \subseteq \{1, \cdots, n\}$ is a binding microstate, it follows from *Equation 60* that

$$\mu_S(A^{c_T}) = \left( \prod_{i \in S} \kappa_i x \right) \varepsilon^{\#(S \backslash T)} = \begin{cases} \left( \prod_{i \in S} \kappa_i \right) x^{\#(S)} & \text{if } S \subseteq T \\ \mathcal{o}_\varepsilon(1) x^{\#(S)} & \text{otherwise.} \end{cases} \tag{61}$$

After coarse graining, we can calculate effective association constants and effective HOCs using the formulas in *Equations 48 and 49*. Let $S$ be a binding microstate and $i \notin S$. Using *Equation 48* and *Equations 60 and 61*,

$$K_{i,S}^\phi = \kappa_i \left( \frac{\sum_{S \cup \{i\} \subseteq T} \lambda_T + \mathcal{o}_\varepsilon(1)}{\sum_{S \subseteq T} \lambda_T + \mathcal{o}_\varepsilon(1)} \right).$$

Letting $\varepsilon \to 0$, we can use the trick described above to rewrite this as

$$K_{i,S}^\phi = \kappa_i \left( \frac{\sum_{S \cup \{i\} \subseteq T} \lambda_T}{\sum_{S \subseteq T} \lambda_T} \right) + \mathcal{o}_\varepsilon(1). \tag{62}$$

*Equation 62* is the more rigorous version of *Equation 22*. It follows from *Equation 62*, using the same trick to reorganise the terms which are $\mathcal{o}_\varepsilon(1)$, that the effective HOCs are

$$\omega_{i,S}^\phi = \frac{K_{i,S}^\phi}{K_{i,\emptyset}^\phi} = \frac{\left( \sum_{\emptyset \subseteq T} \lambda_T \right) \left( \sum_{S \cup \{i\} \subseteq T} \lambda_T \right)}{\left( \sum_{\{i\} \subseteq T} \lambda_T \right) \left( \sum_{S \subseteq T} \lambda_T \right)} + \mathcal{o}_\varepsilon(1). \tag{63}$$

*Equation 63* is the more rigorous version of *Equation 23*. We see that the effective HOCs are independent of the quantities $\kappa_i$ and depend only on the parameters, $\lambda_T$, of the horizontal subgraph $A_\emptyset$.

We can now specify the $\lambda_T$. If $T = \{i_1, \cdots, i_k\}$, where $i_1 < i_2 < \cdots < i_k$, we set

$$\lambda_T = \alpha_{i_1,\{i_2,\cdots,i_k\}} \alpha_{i_2,\{i_3,\cdots,i_k\}} \cdots \alpha_{i_{k-1},\{i_k\}} \delta^k, \tag{64}$$

where each of the $\alpha$ quantities is given by hypothesis. Note that the exponent of $\delta$ depends only on the size of $T$ and not on which elements $T$ contains. *Equation 64* is illustrated in *Figure 6*.

It follows from *Equation 64* that, given any $X \subseteq \{1, \cdots, n\}$,

$$\sum_{X \subseteq T} \lambda_T = \lambda_X (1 + \mathcal{O}_\delta(1)).$$

Using this, we see that the main term in *Equation 63* has the form

$$(1 + \mathcal{O}_\delta(1)) \cdot \frac{\lambda_{S \cup \{i\}}(1 + \mathcal{O}_\delta(1))}{\delta(1 + \mathcal{O}_\delta(1))\lambda_S(1 + \mathcal{O}_\delta(1))}. \tag{65}$$

It follows from *Equation 64* that, when $i < S$, $\lambda_{S \cup \{i\}} = \alpha_{i,S}\lambda_S\delta$, so using the trick above for reorganising the $\mathcal{O}_\delta(1)$ terms, we can rewrite *Equation 65* as $\alpha_{i,S} + \mathcal{O}_\delta(1)$. Substituting back into *Equation 63*, we see that, when $i < S$,

$$\omega_{i,S}^\phi = \alpha_{i,S} + \mathcal{O}_\varepsilon(1) + \mathcal{O}_\delta(1). \tag{66}$$

*Equation 66* is the more rigorous version of *Equation 26*.

With the choice of $\lambda_T$ given by *Equation 64*, we can return to *Equation 62* with $S = \emptyset$ and define

$$\kappa_i = \beta_i \left( \frac{\sum_{\{i\} \subseteq T} \lambda_T}{\sum_{\emptyset \subseteq T} \lambda_T} \right)^{-1}.$$

Substituting back into *Equation 62* with $S = \emptyset$, we see that

$$K_{i,\emptyset}^\phi = \beta_i + \mathcal{O}_\varepsilon(1). \tag{67}$$

*Equation 67* is the more rigorous version of *Equation 27*. The result follows from *Equations 66, 67*.

∎

## Construction of *Figure 8*

We implemented in a Mathematica notebook the proof strategy in Theorem 1 for any number $n$ of sites. The notebook takes as input parameters the $\beta_i$ and the $\alpha_{i,S}$ for $i < S$ in the statement of the theorem, along with specified values for the quantities $\varepsilon$ and $\delta$. It produces as output the effective bare association constants, $K_{i,\emptyset}^\phi$, and effective HOCs $\omega_{i,S}^\phi$ for $i < S$, as given by Theorem 1. The values of $\epsilon$ and $\delta$ can then be adjusted so that the calculated $K_{i,\emptyset}^\phi$ and $\omega_{i,S}^\phi$ are as close as required to the $\beta_i$ and $\alpha_{i,S}$. The notebook is available on request.

*Figure 8* shows the results from using this notebook on three examples, chosen by hand to illustrate different patterns of effective bare association constants and effective HOCs. The actual numerical values are listed below.

The colour names used here refer to the colour code for the three examples in *Figure 8*. The maximum error was calculated as the larger of $\max_i \left| \frac{\beta_i - K_{i,\emptyset}^\phi}{\beta_i} \right|$ and $\max_{i,S} \left| \frac{\alpha_{i,S} - \omega_{i,S}^\phi}{\alpha_{i,S}} \right|$. The quantities $\delta$ and $\varepsilon$ were adjusted to make the maximum error less than 0.01.

The binding curves for each example (*Figure 7B*) show the dependence on concentration of average binding to site $i$ (coloured curves), which can be written in terms of the coarse-grained graph, $A^\phi$, in the form

$$\sum_{S \subseteq \{1,\cdots,n\}} \chi_i(S)\,\mathrm{Pr}_S(A^{\phi})\,.$$

Here, $\chi_i(S)$ is the indicator function for $i$ being in $S$,

$$\chi_i(S) = \begin{cases} 1 & \text{if } i \in S \\ 0 & \text{if } i \notin S. \end{cases}$$

Since the size of $S$, which was denoted by $s$ above, is given by $s = \sum_{1 \le i \le n} \chi_i(S)$, we see from **Equation 50** that the fractional saturation (**Figure 7B**, black curves) is the sum of the average bindings over all sites, normalised to the number of sites, $n$.

| | Maroon | | Orange | | Red | |
|---|---|---|---|---|---|---|
| | $\delta = 10^{-7}, \varepsilon = 10^{-12}$ | | $\delta = 10^{-7}, \varepsilon = 10^{-14}$ | | $\delta = 10^{-7}, \varepsilon = 10^{-16}$ | |
| $i$ | $\beta_i$ | $K^{\phi}_{i,\emptyset}$ | $\beta_i$ | $K^{\phi}_{i,\emptyset}$ | $\beta_i$ | $K^{\phi}_{i,\emptyset}$ |
| 1 | 1.5777 | 1.5776 | 0.031353 | 0.031353 | 0.21257 | 0.21257 |
| 2 | 24.013 | 24.014 | 0.011104 | 0.011104 | 0.84301 | 0.84301 |
| 3 | 89.958 | 89.959 | 13.195 | 13.195 | 9.8514 | 9.8514 |
| 4 | 0.015685 | 0.015685 | 52.437 | 52.437 | 27.000 | 27.000 |
| $i, S$ | $\alpha_{i,S}$ | $\omega^{\phi}_{i,S}$ | $\alpha_{i,S}$ | $\omega^{\phi}_{i,S}$ | $\alpha_{i,S}$ | $\omega^{\phi}_{i,S}$ |
| 1, {2} | 0.084815 | 0.0848456 | 1.0801 | 1.0801 | 50.455 | 50.454 |
| 1, {3} | 3.7432 | 3.7432 | 34.768 | 34.768 | 0.016359 | 0.016401 |
| 1, {4} | 0.044245 | 0.044264 | 0.032668 | 0.032669 | 0.60018 | 0.60018 |
| 2, {3} | 30.240 | 30.239 | 4.0683 | 4.0683 | 7.2944 | 7.2944 |
| 2, {4} | 0.074064 | 0.074083 | 1.5098 | 1.5098 | 0.010809 | 0.010809 |
| 3, {4} | 9.2687 | 9.2685 | 0.025183 | 0.025184 | 0.012613 | 0.012613 |
| 1, {2,3} | 4.0933 | 4.0933 | 0.31238 | 0.31238 | 57.783 | 57.783 |
| 1, {2,4} | 15.687 | 15.683 | 0.70016 | 0.70016 | 0.025618 | 0.025623 |
| 1, {3,4} | 0.013335 | 0.013349 | 0.13042 | 0.13056 | 4.4450 | 4.4450 |
| 2, {3,4} | 0.082851 | 0.082892 | 2.5235 | 2.5235 | 0.13584 | 0.13584 |
| 1, {2,3,4} | 6.5843 | 6.5825 | 0.017404 | 0.017407 | 0.063587 | 0.063833 |
| Max. error | 0.00105 | | 0.00105 | | 0.00386 | |

## Allosteric ensembles for Hill functions
### Construction of *Figure 9*

As described in the main text, we considered an allosteric ensemble with four conformations and six ligand binding sites with no intrinsic cooperativity in any conformation. Accordingly, the bare association constants, $K_{c_k, i, \emptyset}$, constitute 6 free parameters for each conformation $c_k$, $k = 1, \cdots, 4$, giving 24 free parameters. A further 3 free parameters arise for the independent equilibrium labels of the horizontal subgraph of empty conformations, $A_{\emptyset}$, giving 27 free parameters in total. The association constants were restricted to lie in the range $[10^{-4}, 10^4]$ and the equilibrium labels in the range $[10^{-6}, 10^6]$. To compare the binding function, $f(u)$, to the Hill functions $\mathcal{H}_h(x)$, the concentration variable, $u$, was normalised to its half-maximal value, $u_{0.5}$, for which $f(u_{0.5}) = 0.5$ (**Estrada et al., 2016**). The normalised binding function, $g(x) = f(x u_{0.5})$, then satisfies $g(1) = 0.5$. We followed a two-step procedure to find binding functions which approximated Hill functions. The algorithm is publicly available on GitHub (github.com/rosamc/allostery-paper-2021; copy archived at swh:1:rev: 386b23961732962e8ac8390322c9c6e6dfc39168), and we describe it here in general terms. For step 1, we used the measures of position, $\gamma(g)$, and steepness, $\rho(g)$, of a normalised binding function,

$g(x)$, introduced previously (*Estrada et al., 2016*). The steepness of $g(x)$ is the maximum value of its derivative,

$$\rho(g) = \max_{x \geq 0} \frac{dg}{dx},$$

and the position of $g$ is the normalised concentration at which that maximum occurs,

$$\gamma(g) = z, \text{ such that } \left.\frac{dg}{dx}\right|_{x=z} = \rho(g).$$

The combination of these two measures provides an estimate of the shape of the binding function (*Estrada et al., 2016*). Starting with a seed for random number generation, we randomly sampled parameter values independently and logarithmically within the ranges specified above to find parameter sets for which $\gamma(g) \in [0.5, 1.2]$ and $\rho(g) \in [0.5, 1.3]$, which ensures that $g$ is not too far in position-steepness space from the Hill functions (*Estrada et al., 2016*, Supplementary Information, §6.1). This narrows down the search space substantially. Once such a parameter set has been found, step 2 of the procedure followed a Monte Carlo optimisation as follows. This algorithm was fine-tuned by hand, and full details are available with the source code on GitHub. The error between the selected binding function $g$ and the appropriate Hill function, $\mathcal{H}_h$, was measured as the average absolute difference between the functions at 1000 logarithmically spaced points between 0.0005 and 5,

$$\delta(g, \mathcal{H}_h) = \frac{\sum_{1 \leq j \leq 1000} |g(0.0005 u^j) - \mathcal{H}_h(0.0005 u^j)|}{1000},$$

where $u = 10^{0.0003003}$. Starting from the initial parameter set, $\theta_0$, as selected in the first step, we randomly chose each parameter with probability $p$ and, for each chosen parameter, we randomly picked a new value $v_1$ logarithmically in the range $[mv_0, Mv_0]$, where $v_0$ is the existing parameter value. If the chosen value fell outside the appropriate parameter range, we took $v_1$ to be the limit of the range. Having done this for each parameter to generate a new parameter set, $\theta_1$, we chose $\theta_1$ for the next step of the iteration if $\delta(g_{\theta_1}, \mathcal{H}_h) < \delta(g_{\theta_0}, \mathcal{H}_h)$ and, if not, we chose $\theta_1$ with probability $\beta$; otherwise, we retained $\theta_0$. The algorithm parameters $p$, $m$ and $M$ were adjusted so that $p$ decreased and the range $[m, M]$ narrowed as the error decreased. Iterations were continued to an upper limit of $5 \times 10^6$, or until a parameter set was found for which $\delta(g_\theta, \mathcal{H}_h) < 0.0002$. Step 1 and iterations of step 2 were undertaken with $\beta = 0.25, 0.5$ and 0.75 for each of 290 initial seeds for random number generation, and the examples shown in *Figure 9* were selected from among those with the least error. For Hill coefficient $h = 4$, we had to relax the error bound slightly and the two examples shown in *Figure 9* satisfy $0.0003 < \delta(g_\theta, \mathcal{H}_h) < 0.0004$.

## Acknowledgements

We are indebted to Hernan Garcia and an anonymous reviewer for questions and suggestions which helped to improve this paper. JWB and JG were supported by US National Science Foundation (NSF) Award #1462629. RMC was supported by US National Institutes of Health award #GM122928 and EMBO Fellowship ALTF683-2019. FW was supported by the James S McDonnell Foundation and NSF Graduate Research Fellowship #DGE1144152.

## Additional information

### Funding

| Funder | Grant reference number | Author |
|---|---|---|
| National Science Foundation | 1462629 | John W Biddle<br>Jeremy Gunawardena |
| National Institutes of Health | GM122928 | Rosa Martinez-Corral |
| European Molecular Biology Organization | ALTF683-2019 | Rosa Martinez-Corral |

| National Science Foundation | DGE1144152 | Felix Wong |
| James S. McDonnell Foundation | | Felix Wong |

The funders had no role in study design, data collection and interpretation, or the decision to submit the work for publication.

### Author contributions

John W Biddle, Rosa Martinez-Corral, Felix Wong, Conceptualization, Formal analysis, Writing - review and editing; Jeremy Gunawardena, Conceptualization, Formal analysis, Supervision, Methodology, Writing - original draft, Writing - review and editing

### Author ORCIDs

John W Biddle ![ORCID] https://orcid.org/0000-0002-9887-064X
Rosa Martinez-Corral ![ORCID] http://orcid.org/0000-0003-3600-3601
Felix Wong ![ORCID] https://orcid.org/0000-0002-2309-8835
Jeremy Gunawardena ![ORCID] https://orcid.org/0000-0002-7280-1152

### Decision letter and Author response

Decision letter https://doi.org/10.7554/eLife.65498.sa1
Author response https://doi.org/10.7554/eLife.65498.sa2

# Additional files

### Supplementary files

• Transparent reporting form

### Data availability

No data has been generated or acquired for this study, which is purely theoretical.

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
