## [Decision Letter]

**Acceptance summary:**

This paper extends classical models of molecular cooperativity to higher order cooperativity, where the binding of ligand by a protein is affected by other already bound ligands. The work quantifies effective higher order cooperativity between 3 or more ligands that interact indirectly by biasing the underlying (equilibrium) molecular ensemble. The work should be of broad interest to protein scientists since it suggests a new way of quantifying empirical observations of cooperativity.

**Decision letter after peer review:**

Thank you for submitting your article "Allosteric conformational ensembles have unlimited capacity for integrating information" for consideration by *eLife*. Your article has been reviewed by 2 peer reviewers, and the evaluation has been overseen by Arvind Murugan as Reviewing Editor and Aleksandra Walczak as the Senior Editor. The following individual involved in review of your submission has agreed to reveal their identity: Hernan G Garcia (Reviewer #1).

Essential revisions:

The reviewers had mixed opinions, primarily with respect to clarity of the paper and presenting a clear relationship to prior work. In particular, reviewer #2 has concerns about the way cooperativity is quantified here, the benefits of this approach and its relationship to prior work. Below, I summarize a few areas where the paper must be improved prior to being acceptable for publication. Please also refer to the reviewer's detailed reports for constructive criticism that will make this paper more readable and impactful.

1. Flavor of results in the main paper: The work relies on significant mathematical work that is entirely confined to the appendices. The main paper is too superficial as a result and the reader should have more meat to sink their teeth into. See reviewer's comments for suggestions – e.g., some equations (or intuition behind equations) can be moved from the appendix to the main paper. I present one suggestion re: Figure 4 below. Feel free to address this important issue in other ways instead.

Figure 4 is the only figure that presents some sense of the results and is much too brief. Perhaps Figure 4 can be unpacked, possibly into an additional figure, offering intuition into the remarkable binding curves shown (e.g., with positive and negative cooperativity in different regimes). For example, you could show the kinetic network needed to get one or two of the most interesting binding curves shown in Figure 4. The current visualization in Figure 4 in terms of heatmaps is hard to interpret.

The mathematical content in Materials and methods needs to be better integrated with the argument in the main text. One way to do this would be to add notes in the Methods that point to concepts discussed in the main text. See reviewer comments re: the same.

2. Relationship to prior work: Your work seeks to do two distinct things: (a) demonstrate that equilibrium conformational ensembles can implement any pattern of HOCs, (b) introduce a new way to quantify higher order cooperativity that's distinct from binding curve shape.

As one of the reviewers points out, the presentation of (b), relationship to prior work and benefits of the new measure over prior work should be better clarified. See reviewer comments for more. Could you spell out an example or two where the binding curve is an unwieldy or misleading characterization of cooperativity while your HOC coefficient performs better?

3. Concrete biological example – theory can and should precede experiments. But the paper will have more impact if the authors can lay out how to use the framework here to perform or interpret experiments. Ideally this would be done with a concrete example of a protein or protein complex where these ideas might potentially have relevance, how what is known about its conformations predicts HOCs and binding curves, what experimental signatures one might look for and so on – even if there is currently no data.

See review comments for other suggestions.

*Reviewer #1:*

Often in biology, in phenomena ranging from the binding of oxygen to hemoglobin to the binding of transcription factors to DNA, it is observed that the binding of a second ligand to its substrate is more likely than the binding of the first ligand. This so-called cooperativity is usually associated with direct ligand-ligand interactions. However, an increasing body of theoretical work rooted on the Monod-Wymand-Changeux and Koshland-Némethy-Filmer models has shown that, if the substrate can adopt two conformations, cooperativity can arise in the absence of direct interactions between ligands.

Despite the widespread adoption of these models, they have presented limitations when confronted with real data. For example, quantitatively recapitulating gene expression input-output functions in eukaryotes often calls for more than the pairwise interactions that lead to classic cooperativity. Instead, in order to reconcile theory and experiment it is necessary to invoke higher-order cooperativity. Here, multiple bound ligands act in a collective fashion to influence the binding (or unbinding) of additional ligands.

Biddle et al. propose an intriguing theoretical model for realizing higher-order cooperativity between binding sites in a single substrate in the absence of energy dissipation, which means that they must adhere to the strict constraints of microscopic reversibility imposed by thermodynamic equilibrium. They demonstrate that, by extending previous models and allowing the substrate to fluctuate between multiple distinct conformational states, systems may achieve arbitrary higher-order cooperativitive (HOC) behaviors, even at thermodynamic equilibrium. Their graph-based method extends the idea of allosteric regulation to apply to systems with many distinct conformational degrees of freedom and, as such, should, in principle, provide a useful conceptual tool for interrogating the wide range of biological processes in which allostery is thought to play some role.

The paper is extremely well-written, with ample room for the introduction of concepts-including their historical background-and for the discussion. However, we worry that the difficulty of their mathematical notation, as well as their choice to relegate key details about both the derivation and the application of their method to the SI will limit the impact and pedagogical value of this creative and timely work.

Likewise, the considerable import of their finding that sufficiently complex allosteric systems can realize any regulatory logic that is achievable at thermodynamic equilibrium is somewhat obscured by the absence of a clear, detailed application to a concrete biological system. All the same, we view this work as an exciting step towards developing theoretical models that adequately attend to the richness and complexity of real biological systems.

Strengths:

– The paper offers a new framework for thinking about how complex allosteric systems with multiple distinct conformations function to integrate information from ligand binding.

– The authors show that allostery, when sufficiently complex, can provide a physical basis for the emergence of higher-order cooperativities of an arbitrary nature.

– The authors provide an intuitive method for coarse-graining systems with many conformations into a single, tractable ligand-binding graph, which can then be used to quantify higher-order cooperativities between binding sites. This method should prove a useful tool for navigating the complexities present in many real biological systems.

– The authors show that their framework is consistent with (and therefore subsumes) previously used MWC models.

Weaknesses:

– Due to the strong results and implications of the paper, the mathematical proofs in the Materials and methods section must be easy to follow and accessible to the reader. The abundance of indices and references back and forth from the main text make it difficult to follow and evaluate the author's claims throughout this work. The derivations of the authors' coarse-graining procedure and their expression for effective higher-order-cooperativity, as well as their proof that sufficiently complex allosteric systems can achieve any regulatory logic, are nowhere to be found in the main text. While it may not be practical to include these pieces in full, the authors often could at least provide qualitative intuition for the origins and implications of the expressions they present.

– The lemmas and proofs in the Materials and methods are stated mostly in the form of equations, with few explanation on how the proof connects to the concept explained in the main text.

– It is worth noting that the authors limit themselves to considering systems at thermodynamic equilibrium. This is perfectly understandable given the considerable scope of the work already undertaken, but it will be interesting to see what new behaviors might emerge from systems operating away from equilibrium in future work.

– Given that this paper considers only the equilibrium situation, it would be interesting to explicitly state the advantage of adopting the linear framework as opposed to a thermodynamic description in terms of, for example, Boltzmann weights.

– The absence of a thorough, well-illustrated application to a concrete biological system somewhat dampens the paper's impact.

– The authors use the phrase "information integration" multiple times throughout, but they never provide a precise definition of what they mean. Typically a treatment of information transmission would be expected to deal with noise, as well as mean behavior, but that is not done here. They need to clearly define this term early on. While the authors provide an example that does give some intuition in lines 126-136, it might be helpful to move this discussion earlier to provide more context for the rest of the discussion in the introduction.

– In line 41, the authors point out that previous studies investigating effective cooperative effects in MWC models do not "quantitatively determine" the effective cooperativity, but instead infer it indirectly from the shape of the binding curve. However, they do not tell us why this matters. What can we expect to gain by quantifying effective cooperativity directly?

– What is the benefit of having more than 2 conformations? Can the authors show, quantitatively, how performance scales with the number of conformations? The discussion in lines 340-344 provides some basis for this, but the point seems worthy of further discussion and illustration. Is there a graphical way to illustrate the space of achievable integrative behaviors, and how this expands with increasing N (for some given n)?

– This work would be significantly strengthened by including a concrete example that demonstrates both how the framework could be employed to analyze a biological system and what it tells us about how conformational flexibility impacts integrative behaviors. For instance, the authors could revisit their earlier work on the hunchback gene in fruit flies (Estrada et al., Cell, 2016; Park et al., *eLife*, 2019), and show how the space of achievable GRFs expands with the number of conformational degrees of freedom.

*Reviewer #2:*

In this paper, the authors argue correctly that quantification of higher-order coupling (HOC) is crucial for the understanding of biological systems at many different levels of description. I found the paper hard to read. This is due, in part, to the lack of connection with previous descriptions of HOC. The most basic description of pairwise coupling is usually through linkage analysis developed by Wyman. Such coupling is often described by cycles, e.g. a double-mutant cycle or a cycle that describes binding of some ligand X in the absence and presence of a second ligand Y. Pairwise coupling is usually considered to have a dimension of 2 (and not 1 as in the work here). A natural extension to HOC coupling is then done via higher-order dimensional constructs, e.g. triple-mutant boxes for the 3-way coupling between 3 residues (JMB 1990 Aug 5;214(3):613-7; PNAS 2004 Jan 6;101(1):111-6; Annu Rev Biophys. 2017 May 22;46:433-453). Consequently, a key question for me about the current work is the relationship between the previously used measure for HOC and the one described here.

Also, is there an advantage to using the measure proposed in the current work? It seems to me that the description here bypasses intermediate orders of coupling. In other words, nth order coupling is not described in terms of all the lower orders of coupling. Is that a good thing?

In addition, the authors ignore (lines 48-50) the existence of the Hill constant which provides a measure of cooperativity despite having some shortcomings and (line 83) the many previous papers about HOC as mentioned above.

Other comments:

1. Line 308 and elsewhere -it seems that statistical corrections for the binding constants were not introduced. This is OK if stated and not misinterpreted.

2. Line 321 – HOC usually diminishes with factorial decomposition. Why not here?

3. Lines 328, 401-402 – site-heterogeneity leads to apparent negative cooperativity but it is apparent since it can involve no coupling or 'communication' between sites. It should not, therefore, be presented as a possible source for HOC and is not true negative cooperativity.

4. Line 338 – I thought that intrinsic HOC can arise only when the sites are not identical so what am I missing unless it's the statistical factor.

5. Figure 4 – why can binding decrease with increasing substrate concentration?

6 Lines 385-392 – for hemoglobin affinity increases but cooperativity actually decreases at high substrate concentrations because most of the molecules are 'locked' in the R state. Is this captured by the current formalism?

7. Line 699 – fix typo: i to k; I don't understand Equation 15. If each term in the product is a ratio of the terms for forward and reverse directions so should the result on the rhs. Thermodynamically, a product of equilibrium constants is an equilibrium constant but the result on the rhs is not.

8. The analogy with TF binding is potentially problematic because of confusion between different levels of cooperativity. For example, IPTG binding to the lac repressor dimer occurs without cooperativity but 2 IPTG molecules need to be bound for transcription to occur. Hence, measuring transcription as a function of IPTG concentration appears to be very cooperative but the fraction bound as a function of IPTG concentration is not.

---

## [Author Response]

Essential revisions:The reviewers had mixed opinions, primarily with respect to clarity of the paper and presenting a clear relationship to prior work. In particular, reviewer #2 has concerns about the way cooperativity is quantified here, the benefits of this approach and its relationship to prior work. Below, I summarize a few areas where the paper must be improved prior to being acceptable for publication. Please also refer to the reviewer's detailed reports for constructive criticism that will make this paper more readable and impactful.1. Flavor of results in the main paper: The work relies on significant mathematical work that is entirely confined to the appendices. The main paper is too superficial as a result and the reader should have more meat to sink their teeth into. See reviewer's comments for suggestions – e.g., some equations (or intuition behind equations) can be moved from the appendix to the main paper. I present one suggestion re: Figure 4 below. Feel free to address this important issue in other ways instead.Figure 4 is the only figure that presents some sense of the results and is much too brief. Perhaps Figure 4 can be unpacked, possibly into an additional figure, offering intuition into the remarkable binding curves shown (e.g., with positive and negative cooperativity in different regimes). For example, you could show the kinetic network needed to get one or two of the most interesting binding curves shown in Figure 4. The current visualization in Figure 4 in terms of heatmaps is hard to interpret.The mathematical content in Materials and methods needs to be better integrated with the argument in the main text. One way to do this would be to add notes in the Methods that point to concepts discussed in the main text. See reviewer comments re: the same.

Our previous experience has been that most readers would prefer not to confront the mathematics and we had structured the paper accordingly. We apologise for this misjudgement and have taken the following steps to provide more "meat" in the main text.

– We have described the free-energy landscape in more detail, with a new Equation 1 and a new Figure 3.

– As a response to point 2 below, we have added a new section to the Results in which we explain in detail the mathematical relationship between higher-order cooperativity measures. We have introduced a new Figure 5 and new Equations 4 to 16, along with 3 other unnumbered displayed equations.

– We have explained in more detail the basis of coarse graining and the further details provided in the Materials and methods (lines 443-50).

– We have included the essential details of the proof of the flexibility theorem in the main text. This material includes the new Equations 21 to 27, along with 3 other un-numbered displayed equations, as well as the new Figure 6, which is enhanced from what was previously Scheme 2 in the Material and methods. We still provide a fully rigorous and concise proof in the Materials and methods.

– We have broken up the old Figure 4 into two new figures (Figures 7 and 8), as requested, and included a new depiction of the allostery graph in Figure 8A.

2. Relationship to prior work: Your work seeks to do two distinct things: (a) demonstrate that equilibrium conformational ensembles can implement any pattern of HOCs, (b) introduce a new way to quantify higher order cooperativity that's distinct from binding curve shape.As one of the reviewers points out, the presentation of (b), relationship to prior work and benefits of the new measure over prior work should be better clarified. See reviewer comments for more. Could you spell out an example or two where the binding curve is an unwieldy or misleading characterization of cooperativity while your HOC coefficient performs better?This is an important point and we apologise for our unfamiliarity with the prior work described by Reviewer #2. We have now pointed out this prior work in the Introduction (lines 98-106) and included a new section of the Results entitled Relationships between higher-order measures (pages 15-21) in which we carefully explain the relationship between our HOCs and the two forms of higher-order couplings introduced in previous work. We present general formulas for the couplings described in both Horovitz and Fersht 1992 (Equations 6 and 7) and Horovitz and Fersht 1990 (Equation 11). The latter formula seems to be new, to our knowledge. We further give new general formulas for calculating both measures from our HOCs (Equations 8 and 14), from which we deduce rigorously that the two measures introduced in Horovitz and Fersht 1990, 1992 are, in fact, the same (Equation 15). We were surprised not to find a clear statement of this equality in the literature. We presume that it must be well known to those in the field and to be tacitly assumed. We note that it would not be easy to formulate a rigorous statement of this equality in the absence of a general definition for the higher-order couplings introduced in Horovitz and Fersht 1990. We have now provided such a definition in Equation 11. We hope, therefore, that this new section will be of some value and that it provides a full answer to the Reviewer's question as to "*the relationship between the previously used measure for HOC and the one described here*". As to the benefits of our HOCs, we make comparisons between all the measures in the penultimate paragraph of the new section. We feel that each measure is suitable for a different purpose and we explain why our HOCs are well suited to the problems studied in the present paper.3. Concrete biological example – theory can and should precede experiments. But the paper will have more impact if the authors can lay out how to use the framework here to perform or interpret experiments. Ideally this would be done with a concrete example of a protein or protein complex where these ideas might potentially have relevance, how what is known about its conformations predicts HOCs and binding curves, what experimental signatures one might look for and so on – even if there is currently no data.

We had included an extensive discussion of the implications of our results for gene regulation, based on the "haemoglobin analogy", as depicted in the old Figure 5 (now Figure 10), and we remarked on the kinds of experiments that would be needed to test this conceptual picture (lines 803-11). We feel this does illustrate the significance of our findings but acknowledge that this material is Discussion rather than Results. Accordingly, we have included a new final section of the Results entitled **Allosteric ensembles for Hill functions** (pages 32-5) and a new figure (Figure 9) to show that allosteric ensembles can be found whose binding functions closely approximate Hill functions.

See review comments for other suggestions.Reviewer #1:[…] – Given that this paper considers only the equilibrium situation, it would be interesting to explicitly state the advantage of adopting the linear framework as opposed to a thermodynamic description in terms of, for example, Boltzmann weights.

We thank the Reviewer for this suggestion. We have now explained the advantage of the graphbased linear framework at the point where we discuss equilibrium statistical mechanics (lines 266-74). We have also noted there the central role that linear framework graphs play in the subsequent new section in which we examine the relationship between higher-order measures.

– The authors use the phrase "information integration" multiple times throughout, but they never provide a precise definition of what they mean. Typically a treatment of information transmission would be expected to deal with noise, as well as mean behavior, but that is not done here. They need to clearly define this term early on. While the authors provide an example that does give some intuition in lines 126-136, it might be helpful to move this discussion earlier to provide more context for the rest of the discussion in the introduction.

We apologise for not being clear about what we mean by "integration". We were not thinking of it in terms of information theory, as the Reviewer suggests, but, rather, as the process by which the occurrence of ligand binding influences downstream function. We have now stated this in the second sentence of the text (lines 3-7).

– In line 41, the authors point out that previous studies investigating effective cooperative effects in MWC models do not "quantitatively determine" the effective cooperativity, but instead infer it indirectly from the shape of the binding curve. However, they do not tell us why this matters. What can we expect to gain by quantifying effective cooperativity directly?

Briefly, we gain access to the free-energy landscape, which cannot be acquired from aggregated measures such as the shape of the binding curve. To introduce this point, we have now added a sentence at lines 31-32 to explain how association constants or cooperativites are another way of describing free energies. We have then explained more carefully on lines 53-62 the significance of effective cooperativities for describing the free-energy landscape.

– What is the benefit of having more than 2 conformations? Can the authors show, quantitatively, how performance scales with the number of conformations? The discussion in lines 340-344 provides some basis for this, but the point seems worthy of further discussion and illustration. Is there a graphical way to illustrate the space of achievable integrative behaviors, and how this expands with increasing N (for some given n)?

We fully agree with the Reviewer that these are interesting questions but we fear that answering them amounts to writing another paper. As the Reviewer notes, we have explained why more conformations are mathematically essential to achieve flexibility (lines 520-21) and we have proved that, with enough conformations, complete flexibility can be achieved (**Integrative flexibility of ensembles** and Theorem 1 in the Materials and methods). We also note, in the new final section of the results, that the number of conformations may play a role in the flexibility with which Hill functions can be approximated (lines 649-56). However, as we point out, the impact of the number of conformations is a delicate question because of the potential interplay between numbers of sites, numbers of conformations and parametric ranges. To go further and to work out how the number of conformations influences function requires substantial further work. We feel this is more appropriate to a follow-up study.

– This work would be significantly strengthened by including a concrete example that demonstrates both how the framework could be employed to analyze a biological system and what it tells us about how conformational flexibility impacts integrative behaviors. For instance, the authors could revisit their earlier work on the hunchback gene in fruit flies (Estrada et al., Cell, 2016; Park et al., eLife, 2019), and show how the space of achievable GRFs expands with the number of conformational degrees of freedom.

Our thanks to the Reviewer for this suggestion. We have now included a new final section of the Results entitled **Allosteric ensembles for Hill functions** (pages 32-5) along with a new Figure 9 in which we show how the Hill functions, which provide fits to experimental data on hunchback, can be recovered from an allosteric ensemble.

Reviewer #2:In this paper, the authors argue correctly that quantification of higher-order coupling (HOC) is crucial for the understanding of biological systems at many different levels of description. I found the paper hard to read. This is due, in part, to the lack of connection with previous descriptions of HOC. The most basic description of pairwise coupling is usually through linkage analysis developed by Wyman. Such coupling is often described by cycles, e.g. a double-mutant cycle or a cycle that describes binding of some ligand X in the absence and presence of a second ligand Y. Pairwise coupling is usually considered to have a dimension of 2 (and not 1 as in the work here). A natural extension to HOC coupling is then done via higher-order dimensional constructs, e.g. triple-mutant boxes for the 3-way coupling between 3 residues (JMB 1990 Aug 5;214(3):613-7; PNAS 2004 Jan 6;101(1):111-6; Annu Rev Biophys. 2017 May 22;46:433-453). Consequently, a key question for me about the current work is the relationship between the previously used measure for HOC and the one described here.Also, is there an advantage to using the measure proposed in the current work? It seems to me that the description here bypasses intermediate orders of coupling. In other words, nth order coupling is not described in terms of all the lower orders of coupling. Is that a good thing?In addition, the authors ignore (lines 48-50) the existence of the Hill constant which provides a measure of cooperativity despite having some shortcomings and (line 83) the many previous papers about HOC as mentioned above.

We are grateful to the Reviewer for pointing out the previous work on higher-order measures and apologise for having overlooked it. We have addressed this important matter in detail and discussed the advantages of the new measure, as fully described in point **2** above. We have now cited in a new paragraph of the Introduction (lines 98-106) all the references provided by the Reviewer as well as Horovitz and Fersht 1992, which we have discussed further in the Results (see point **2**), Jain and Ranganathan 2004, Sadovsky and Yifrach 2007 and Carter et al. 2017. We hope these revisions go some way towards placing the paper in the context of previous work.

Pairwise coupling is usually considered to have a dimension of 2 (and not 1 as in the work here).

We agree that this is so for the customary higher-order couplings and we have used the new Equation 13 to point out this difference (lines 386-8). We note that the situation is more complicated when there is a non-trivial "offset", which arises in the new treatment of higher-order couplings which we have provided (Equation 11). The offset increases the order of the corresponding HOC, as can be seen from Equations 13 or 14.

It seems to me that the description here bypasses intermediate orders of coupling. In other words, nth order coupling is not described in terms of all the lower orders of coupling. Is that a good thing?

Indeed, the Reviewer is correct in saying that our HOCs are not hierarchical. Whether that is a good thing or not depends, presumably, on what kinds of problems one is trying to address. We believe that HOCs are well suited to describe integration of binding information and specifically to understand how such integration arises "effectively" from a conformational ensemble through coarse graining. This is one of the main contributions of our paper, for which a hierarchical measure of coupling would have been substantially harder to work with. Furthermore, as we show in Equations 8 and 14, our HOCs can precisely describe the hierarchical "intermediate orders of coupling" which are present in the higher-order measures introduced in Horovitz and Fersht 1990 and 1992. With Equations 8 and 14 now available, there is no difficulty in calculating the effective higher-order couplings arising from any conformational ensemble, thereby recovering the "intermediate orders of coupling" in this generalised setting.

In addition, the authors ignore (lines 48-50) the existence of the Hill constant which provides a measure of cooperativity despite having some shortcomings.

We have now mentioned the Hill coefficient (lines 53-9) and explained more carefully why aggregated measures of this kind provide only limited information about the underlying free energies. This point is reiterated in the last section of the Results (lines 640-5) and in the new Figure 9.

Other comments:1. Line 308 and elsewhere -it seems that statistical corrections for the binding constants were not introduced. This is OK if stated and not misinterpreted.

The Reviewer is correct that we do not use statistical factors. They are required when binding states are represented by the number of bound sites. We avoid this problem by accounting for each site which is bound in the subset of bound sites. At the specific point to which Reviewer refers, now Equation 19, we show that HOCs depend only on the number of bound sites. Statistical factors do not appear to be necessary for the discussion that follows.

2. Line 321 – HOC usually diminishes with factorial decomposition. Why not here?

We are not sure what the Reviewer means by "factorial decomposition". However, our finding that cooperativity increases with order for the MWC-like ensemble (Equation 20) was for our definition of HOC. It is conceivable that this is not the case for the measures introduced in Horovitz and Fersht 1990, 1992. Indeed, Equations 8 and 14, which show how higher-order couplings are calculated from HOCs, involve a ratio of HOCs. Hence, it would be possible, in principle, for these other measures to diminish with order, as the Reviewer suggests, even though our HOCs do not. However, we have not investigated this matter further.

3. Lines 328, 401-402 – site-heterogeneity leads to apparent negative cooperativity but it is apparent since it can involve no coupling or 'communication' between sites. It should not, therefore, be presented as a possible source for HOC and is not true negative cooperativity.

We have been careful to make the distinction which the Reviewer draws between cooperativity at the level of a single molecule, and "effective" cooperativity, at the level of an ensemble. We distinguish throughout the paper between the "intrinsic" cooperativity within a given conformation and the "effective" cooperativity arising from the ensemble. We prefer "effective" to either "apparent" or "false" cooperativity. We do not present the heterogeneity of sites as a source of negative cooperativity, only of negative effective cooperativity (line 400 in the original paper; line 699-700 in the revision). We feel this is a reasonable way to maintain the distinction which the Reviewer makes.

4. Line 338 – I thought that intrinsic HOC can arise only when the sites are not identical so what am I missing unless it's the statistical factor.

There seems to be some confusion here. We define "intrinsic" HOC to be the cooperativity between sites in a single conformation (Equation 2). We define sites to be "identical" if they have the same association constants for binding (line 477-8). It is possible for sites to be identical and still have intrinsic HOCs but, in the passage in question, we impose the requirement that all intrinsic HOCs are one, so that the sites are independent. This means that any effective cooperativity which arises in the ensemble cannot be attributed to intrisinc cooperativity arising from an individual conformation.

5. Figure 4 – why can binding decrease with increasing substrate concentration?

Average total binding, or fractional saturation, cannot increase with increasing substrate, no matter what cooperativities are present. That is a consequence of thermodynamics. However, average binding at an individual site can increase or decrease depending on the pattern of cooperativities, as shown in Figure 7B.

6 Lines 385-392 – for hemoglobin affinity increases but cooperativity actually decreases at high substrate concentrations because most of the molecules are 'locked' in the R state. Is this captured by the current formalism?

We do not know which measure of cooperativity the Reviewer has in mind here. However, if the implication is that some measure of cooperativity becomes concentration dependent, then none of the measures discussed in the paper have that property. They are all independent of concentration. Accordingly, the current formalism would not capture the behaviour described by the Reviewer, although it seems like an interesting question to explore further.

7. Line 699 – fix typo: i to k; I don't understand Equation 15. If each term in the product is a ratio of the terms for forward and reverse directions so should the result on the rhs. Thermodynamically, a product of equilibrium constants is an equilibrium constant but the result on the rhs is not.

Corrected. Thank you! The old Equation 15 (new Equation 39) is for a linear framework graph. In our treatment in this section, the only requirement for an edge label is that it is a rate, with units of (time)^-1^, and no thermodynamic terms, such as ligand concentrations, are specified within the labels. Accordingly, the ratios in Equation 39 are all non-dimensional, so no inconsistency arises between the left-hand and right-hand sides.

8. The analogy with TF binding is potentially problematic because of confusion between different levels of cooperativity. For example, IPTG binding to the lac repressor dimer occurs without cooperativity but 2 IPTG molecules need to be bound for transcription to occur. Hence, measuring transcription as a function of IPTG concentration appears to be very cooperative but the fraction bound as a function of IPTG concentration is not.

Indeed, we agree that cooperativity depends crucially on which input is being considered: if the input is the TF, that gives a very different result than if the input is IPTG. We do not see this as problematic but, rather, as a potential source of confusion if the input is not clearly specified. To address the Reviewer's concern, we have made sure to say "input pattern of TFs" throughout the Discussion.